



# New technique for high-precision, simultaneous measurements of CH4, N2O and CO2 concentrations, isotopic and elemental ratios of N2, O2 and Ar, and total air content in ice cores by wet extraction

Ikumi Oyabu[1], Kenji Kawamura[1,2,3], Kyotaro Kitamura[1], Remi Dallmayr[4], Akihiro Kitamura[5], Chikako Sawada[6], Jeffrey P. Severinghaus[7], Ross Beaudette[7], Satoshi Sugawara[8], Shigeyuki Ishidoya[9], Dorthe Dahl-Jensen[10,11], Kumiko Goto-Azuma[1,2], Shuji Aoki[12], Takakiyo Nakazawa[12]

1 National Institute of Polar Research, Tokyo 190-8518, Japan
2 Department of Polar Science, The Graduate University of Advanced Studies (SOKENDAI), Tokyo 190-8518, Japan
3 Japan Agency for Marine Science and Technology (JAMSTEC), Yokosuka 237-0061, Japan
4 Alfred Wegener Institute, Am Alten Hafen 26, Bremerhaven 27568, Germany
5 Labosoltech LLC, Tokyo 190-0003, Japan
6 Atmosphere and Ocean Research Institute, University of Tokyo, Tokyo 277-0882, Japan
7 Scripps Institution of Oceanography, University of California San Diego, La Jolla, CA 92093, USA
8 Miyagi University of Education, Sendai 980-0845, Japan
9 National Institute of Advanced Industrial and Technology (AIST), Tsukuba 305-8569, Japan
10 University of Copenhagen, Tagensvej 16, Copenhagen 2100, Denmark
11 University of Manitoba, 66 Chancellors Circle, Winnipeg, Manitoba, R3T 2N2, Canada
12 Tohoku University, Sendai 980-8577, Japan

Correspondence to: Ikumi Oyabu (oyabu.ikumi@nipr.ac.jp)

**Abstract.** Air in polar ice cores provides various information on past climatic and atmospheric changes. We developed a new method combining wet extraction, gas chromatography and mass spectrometry, for high-precision, simultaneous measurements of eight air components ($CH_4$, $N_2O$ and $CO_2$ concentrations, $\delta^{15}N$, $\delta^{18}O$, $\delta O_2/N_2$, $\delta Ar/N_2$ and total air content) from an ice core sample of ~60 g. The ice sample is evacuated for ~2 hours and melted under vacuum, and the released air is continuously transferred into a sample tube at 10 K within 10 minutes. The air is homogenized in the sample tube overnight at room temperature, and split into two aliquots for mass spectrometric and gas chromatographic measurements. Cares are taken to minimize contamination of greenhouse gases with long evacuation time, consumption of oxygen during sample storage by passivation treatment on sample tubes, and fractionation of isotopic ratios with long homogenization time for splitting. Precisions are assessed by analysing standard gases with artificial ice, and by duplicate measurements of the Dome Fuji and NEEM ice cores. The overall reproducibility (one standard deviation) from duplicate ice-core analyses are 3.2 ppb, 2.2 ppb and 3.1 ppm for $CH_4$, $N_2O$ and $CO_2$ concentrations, 0.006, 0.010, 0.09 and 0.12 ‰ for $\delta^{15}N$, $\delta^{18}O$, $\delta O_2/N_2$ and $\delta Ar/N_2$, and 0.67 $ml_{STP}$ $kg^{-1}$ for total air content, respectively. Our new method successfully combines the high-precision, small-sample and multiple-species measurements, with a wide range of applications for ice-core paleoenvironmental studies.



## 1. Introduction

Measurements of gas components of polar ice cores have provided valuable information on past climatic, atmospheric and glaciological changes. For example, $CH_4$, $N_2O$ and $CO_2$ are important greenhouse gases with natural and anthropogenic variations. $CH_4$ concentration (defined as a dry air mole fraction in this paper) in deep ice cores is useful for detecting abrupt climate changes and to synchronize age scales of different ice cores (e.g., Blunier and Brook, 2001; Brook et al., 1996; WAIS Divide Project Members, 2015). $\delta^{15}N$ of $N_2$ and $\delta^{40}Ar$ of Ar provide information on past firn thickness and surface temperature

(Huber et al., 2006b; Kobashi et al., 2011; Kobashi et al., 2008; Orsi et al., 2014; Severinghaus and Brook, 1999; Severinghaus et al., 1998). $\delta O_2/N_2$ in some ice cores are proxies for local summer insolation and used to constrain age scales by orbital tuning (Bender, 2002; Kawamura et al., 2007). $\delta^{18}O$ of $O_2$ records the variations of terrestrial hydrological cycles and used for dating as well as detection of abrupt climate changes (Bazin et al., 2013; Extier et al., 2018; Landais et al., 2010; Seltzer et al., 2017; Severinghaus et al., 2009). Total air content (TAC) is affected by atmospheric pressure, temperature, and firn porosity at bubble

close-off (Martinerie et al., 1994; Martinerie et al., 1992) and used for reconstructing ice sheet surface elevation (NEEM community members, 2013) and orbital tuning (Bazin et al., 2013; Lipenkov et al., 2011; Raynaud et al., 2007).

Capabilities of both reducing the sample size and improving the analytical precision of ice core analyses are desired, especially for deep ice cores from low accumulation sites to acquire high-resolution data. For example, the inter-polar difference (IPD)

of $CH_4$ for the Holocene is ~30 – 50 ppb (Beck et al., 2018; Chappellaz et al., 1997; Mitchell et al., 2013), thus analytical uncertainty of a few ppb is required for reconstructing subtle changes in IPD. Uncertainty of < ~0.01 ‰ would be required for $\delta^{18}O$ of $O_2$ (after correcting gravitational fractionation by $\delta^{15}N$) to detect the changes for Heinrich events (Seltzer et al., 2017; Severinghaus et al., 2009). The smallest amplitude of the local summer insolation variation at the precession band is a few %, and the corresponding amplitude of $\delta O_2/N_2$ may be < 0.5 ‰.


High precisions with relatively small samples have already been achieved for some species; ± 2.8 ppb for $CH_4$ with ~60 g of ice by Oregon State University (OSU) (Mitchell et al., 2013), ± 1.5 ppb for $N_2O$ with ~20 g of ice by Seoul National University (Ryu et al., 2018), and 0.005 ‰ for $\delta^{15}N$ and 0.01 ‰ for $\delta^{18}O$ with ~15 g of ice by Scripps Institution of Oceanography (SIO) (Seltzer et al., 2017; Severinghaus et al., 2009). However, a total of ~100 g of ice and more than one laboratory are required

to measure all species. Multiple-species measurements combining gas chromatography (for greenhouse gases) and mass spectrometry (for major gas ratios) have been applied by Tohoku University (Kawamura, 2001; Kawamura et al., 2003, 2007), but with lower precisions than the above-mentioned values and larger samples (> 200 g).

Here, we present a new method developed at National Institute of Polar Research (NIPR) to measure eight air components

($\delta^{15}N$, $\delta^{18}O$, $\delta O_2/N_2$, $\delta Ar/N_2$, concentrations of $CH_4$, $N_2O$ and $CO_2$, and TAC) using a 60 g of ice with high precisions. This method has the technical advantages of reducing the sample size without sacrificing precision. It also has the advantage of





paleoclimatic studies that all the measured species can be compared without any age difference. The method is also desired for very old ice cores from the Antarctic interior, with the resolution for 1.5-million-year ice near bedrock is expected to be on the order of 10 kyr m$^{-1}$ (Parrenin et al., 2017).


This paper is structured as follows. Chapter 2 describes the air extraction from ice and the splitting of the extracted air for the analyses by respective instruments. Chapter 3 describes the measurements of the sample air with two gas chromatographs and a mass spectrometer. The system performance and precisions are evaluated by various tests with standard gases (Chapter 4) and comparisons of our data from ~100 ice-core samples (Dome Fuji and NEEM) with published records from other

laboratories.

## 2. Air extraction and split

In order to measure $CH_4$, $N_2O$ and $CO_2$ concentrations, isotopic and elemental ratios of $N_2$, $O_2$ and Ar, and TAC from one ice sample with high precision, we modified the wet extraction and measurement techniques at Tohoku University (Kawamura et al., 2003; Kawamura et al., 2007; Nakazawa et al., 1993a; Nakazawa et al., 1993b). Briefly, an ice sample of 50 – 70 g is

melted under vacuum, and the released air is immediately and cryogenically transferred into a sample tube at < 10 K (cooled with a closed cycle refrigerator) without refreezing the meltwater. It requires a relatively short time (< 10 minutes) for melting ice and transferring extracted air, minimizing contaminations due to degassing from the inner walls of the vessel and line. Also, the pressure over the meltwater is relatively low (~100 – 200 Pa), thus the dissolution of released air in the meltwater is minimal (Kawamura et al., 2003). In comparison, the melt-refreeze method typically requires several tens of minutes to

refreeze meltwater (e.g., Brook et al., 2005; Chappellaz et al., 1997; Flückiger et al., 1999), possibly elevates trace gas concentrations in the extracted air by degassing from the inner wall of the vessel, as well as alter the air composition by gas-dependent dissolution in meltwater (at several tens of hPa) and incomplete degassing during refreezing.

The extracted air is homogenized in the sample tube for one night and split into two aliquots for mass spectrometric (MS) and

gas chromatographic (GC) measurements. About 20 and 80 % of the sample are used for the MS and GC measurements, respectively.

### 2.1 Air extraction

A schematic diagram of our extraction system is shown in Fig. 1. The components of the extraction line (tubings, fittings, valves and vessels) are made of electropolished (EP) stainless steel except for traps, which are made of Pyrex glass. The traps

1 – 3 have Kovar glass-to-metal transition. It has six inlet ports for stainless-steel vessels, each containing an ice core sample. The vessels and traps 1 – 3 are connected to the line with metal face-seal fittings (Fujikin UJR®, 1/2") using nickel gaskets. Diaphragm metal-seal valves (Fujikin FUDDFM-71G-9.52) are used for all stop valves (V1 – V23). ISO-KF25 flanges are





attached to both ends of the trap 4 with two-component epoxy adhesive, and Viton o-rings are used for connecting the trap to the line. The vacuum is provided by a turbomolecular pump (Pfeiffer HiPace 80) backed by an oil rotary pump (Edwards).

The vessels are made of stainless steel pipe (65A) with Conflat flange (ICF114) with a volume of ~600 $cm_3$, and the sample tubes are made of 1/4" EP stainless-steel tube with a metal-seal valve (Fujikin FUDDFM-71G-6.35) with a volume of 6.6 $cm_3$.

After the construction of the extraction line (before actual use), we performed pre-treatment of inner surfaces of all the lines, vessels and sample tubes as follows. Pure $O_2$ (> 99.999 %) was humidified by bubbling through pure-water in a glass flask

sealed with silicone cap at room temperature, and flowed into the lines and vessels heated to 90 – 100 ˚C with heating tapes at a flow rate of ~20 – 50 ml $min_{-1}$ for two weeks to efficiently remove trace organic substances and hydrocarbons. After the treatment, the line and vessels were evacuated at 90 – 100 ˚C by a turbomolecular pump for one week. The same treatment is applied to the sample tubes for air extraction.

For routine air extraction, the sample tubes and extraction line are evacuated overnight to < $1.3 \times 10_{-4}$ Pa (measured at the head of the turbomolecular pump with an ionization gauge, P3). If no extraction is planned for two days or more (i.e., on weekends and holidays), the sample tubes and extraction line are filled with pure air (> 99.99995%) at ~500 Pa. On the day of air extraction from ice core samples, trap 4 is cooled to -196 ˚C by liquid nitrogen to evacuate further the sample tubes and extraction line (< $10_{-4}$ Pa). The vessels are brought out from an oven at 50 ˚C and cooled to room temperature in ~30 min, and

then brought to the cold room at -20 ˚C for further cooling. Ice core samples of ~90 – 150 g, typically 7- to 12-cm long, are cut out from bulk ice-core samples with a band saw in a cold room at -20 ˚C. The same band saw is used to trim all faces for rough decontamination, removing ~2 – 11 mm from the original surfaces. The inner ~50 – 70 g of ice is used for the air extraction, and the removed outer ice is stored for other measurements (e.g., for multiple analyses in case of measurement failures). The amount of ice may be reduced to ~35 g for all measurements with somewhat lower precisions, and to ~9 g if

only MS measurements are conducted (without sample splitting).

As $\delta O_2/N_2$ and $\delta Ar/N_2$ ratios of the Dome Fuji ice core become highly fractionated especially near the surface of ice samples due to diffusive gas loss during ice storage, the surface must sufficiently be removed to precisely measure the air composition (Bereiter et al., 2009; Ikeda-Fukazawa et al., 2005; Kawamura et al., 2007). The gas loss also affects $\delta_{18}O$ and $\delta_{40}Ar$ because

it is weakly mass-dependent (Severinghaus et al., 2009). The thickness of required surface removal depends on the storage period, storage temperature and the form of air in ice (bubbles or clathrate-hydrates). On the other hand, it is established that the gas-loss effect is not important for greenhouse gas concentrations because of their large molecular sizes, high natural variability and measurement uncertainties. The fractionation of $\delta O_2/N_2$ is the most problematic among the measured species, thus we tested different thicknesses of surface removal from the first Dome Fuji core stored at -50˚C for 20 years by focusing

on $\delta O_2/N_2$. The surface is firstly trimmed with a band saw and secondly by a ceramic knife. The shaving by knife also enables the visual inspection of the ice for any cracks. We found that more than 8 mm should be removed for clathrate-hydrate ice (>





~1400 m) (details are described in section 5.2). The cleaned ice sample is placed in a pre-cooled extraction vessel and sealed with a Conflat flange and copper gasket. The vessels are placed in a dewar vessel that accommodates a copper tube (o.d. = 78 mm, i.d. = 74 mm, height = 135 mm) and a eutectic refrigerant bag (~1000 g, pre-cooled to -50 ˚C), to keep the ice temperature
below -25 ˚C.

Up to six vessels, thus prepared, are brought to our laboratory room at room temperature. All valves on the extraction line are closed, and the closed-cycle refrigerator is turned on. Then, pure air is introduced from V16 to purge the manifold for vessels, and the vessels are connected to the line. The room air is evacuated from the vessels with the turbomolecular pump. After ~5
min when the pressure after two water traps (i.e., without water vapor) is below $10^{-2}$ Pa, the flanges and connections are leak-tested with a helium leak detector ($< 10^{-8}$ Pa L $s^{-1}$). Then, pure air is introduced into the vessels and pumped out four times to further remove room air from the vessels. All the vessels are then evacuated for 90 min through the evacuation line (Fig. 1, blue). Typically, four to five samples are simultaneously evacuated. The evacuation is made to remove residual room air from the vessels as well as to sublimate the ice surface for further cleaning of the sample. Then, the vacuum line is switched to
sample transfer line (Fig. 1, pink), which is the line for transferring sample air during the extraction, by closing V8, V9, V14 and V17, cooling the traps 2 and 3 to -80 ˚C with ethanol, and opening V22, V13, V10 and V8. The evacuation continues for another 30 min.

After the evacuation, all vessels, except for one for the first extraction, are isolated by closing V2 – V7. V19 and V21 are
opened, and the vessel and line are evacuated for another ~5 min. V22 is closed to stop the evacuation, and the valve of a sample tube is opened to establish the line for sample air transfer. Then, the ice sample is melted by immersing the vessel in a hot water bath (~90 ˚C) by a few mm from the bottom. The air released from the melting ice is continuously transferred into the sample tube at ~10 K, after passing through two water traps at −80 °C. The first trap has sufficient inner volume to condense a large amount of water vapor, and the second trap contains fine glass tubes for high trapping efficiency. Sample transfer is
monitored by a Baratron Gauge (MKS, full scale = 1333 Pa) (P1 in Fig. 1), which measures the sample air pressure in the line without water vapor. The maximum pressure during the transfer is ~100 – 200 Pa. The hot water bath is removed after the completion of ice melting. When the pressure decreases below the detection limit (0.1 Pa), the sample transfer is considered to be complete, and the valve of the tube is closed. Residual pressure in the transfer line is measured using a Convectron Gauge (Granville-Phillips, P2 in Fig. 1) (typically <0.2 mTorr). The time required for melting the ice sample is <~3 minutes, and the
total time for the air transfer takes ~10 min (including the ice melting). Finally, the valve of the vessel is closed, and the line is evacuated for ~2 min to decrease the pressure to $<10^{-4}$ Pa (P3).

The pressure of air released from the ice sample in the next vessel during the previous transfer is measured with the Baratron Gauge (P1) by closing V20, V21 and V22 and opening the valve of the vessel (typically <1.0 Pa for the second vessel and ~4
Pa for the fifth vessel). This ensures the absence of a leak at the flange and fitting of the vessel. Then, V22 and V21 are opened,



and the line and vessel are evacuated for ~5 min, the ice sample is melted, and the released air is transferred to the next sample tube. We repeat these procedures until all the extractions are completed.

After collecting the air from all prepared samples, the sample tubes are removed from the helium cycle cooler and laid in the laboratory room with ambient temperatures for one night. All the vessels and traps are also disconnected from the line, rinsed with pure water, and placed in ovens at 50˚C for drying. Another set of sample tubes and traps are connected to the line and evacuated with the turbomolecular pump (to <$10^{-2}$ Pa), and then the line is checked with a helium leak detector (< $10^{-8}$ Pa L $s^{-1}$). After the leak check, all the valves are opened, and the line is evacuated until the next extraction.

## 2.2 Splitting

A small aliquot of air is separated from the sample tube and transferred to a second tube using the "split line" (Fig. 2) for MS analysis. We employed the general design of the line for noble gas measurements developed at SIO (Bereiter et al., 2018). The split line is made of electropolished stainless steel except for a U-shaped cold trap, which is made of Pyrex glass and connected to the line with bored-through 3/8" UltraTorr® fittings (Swagelok). A diaphragm metal-seal valve (Fujikin FUDDFM-71G-6.35) is placed next to the sample tube (V1) for splitting, and stainless-steel bellows valves (Swagelok SS-8BW or SS-4H) are used for other valves. The vacuum is provided by a turbomolecular pump (Pfeiffer HiPace80) backed by a dry scroll pump. The same pre-treatment with humidified $O_2$, as applied to the extraction line, is employed for the split line. To minimize the depletion of the $\delta O_2/N_2$ ratio during the sample storage in the tube, a passivation treatment (GoldEP-White®, Nissho Astec Co. Ltd.) is employed, which forms a passive layer of oxidized chromium on the stainless-steel surface (hereafter, this type of tube is called GEPW).

The experimental procedures are as follows. The split line is filled with pure air (> 99.99995%) at ~500 Pa when not in use (e.g., during nights and weekends), and evacuated for more than 30 minutes before the splitting. The GEPW tubes are evacuated overnight. The sample tube containing ice-core air is connected to the split line with a 1/4" UJR® fitting using a silver-plated nickel gasket. The GEPW tube is inserted in a helium cycle cooler at <10 K and connected to an adapter with a VCR® fitting, which is then connected to the split line with a VCO® fitting (Swagelok). The whole line is evacuated for ~30 minutes, during which the pressure decreases to < $3 \times 10^{-5}$ Pa (measured at the head of a turbomolecular pump with an ionization gauge). After a leak check, V1 is closed, and the valve on the sample tube is opened to expand the sample air into the small volume (1.4 $cm^3$) between the valves. The time required for equilibration of the air composition in the small volume with the sample tube is > 20 minutes. During this waiting time, the sample air may be fractionated if the temperature gradient exists between the tube and the small volume. To minimize such fractionation, the sample tube and small volume are covered with a sheet of bubble wrap so that air conditioners on the laboratory ceiling do not directly blow against the splitting part. The expanded air is then split by closing the valve of the sample tube. The air in the split volume is transferred to the GEPW tube for 5 minutes, after passing through the cold trap at −196 °C. The sample transfer is monitored by measuring the pressure of



the line with a Baratoron Gauge (MKS, full scale = 1333 Pa) (P1 in Fig. 2). The GEPW tube is lowered by a few centimeters

into the helium cycle cooler when the pressure becomes below 1.0 Pa to improve the trapping efficiency of the gas by exposing fresh metal surface. The air transfer is complete in 5 minutes, and the valve of the GEPW tube is closed. The residual pressure is measured using a Convectron Gauge (Granville-Phillips) (P2 in Fig. 2), and the GEPW tube is disconnected from the line. The GEPW tube is warmed to room temperature and allowed to homogenize the sample air for at least 3 hours before the MS analysis. Sometimes the samples are measured on the following day of the splitting. The air remained in the original sample

tube is used for measuring $CH_4$, $CO_2$ and $N_2O$ concentrations as well as total air content.

## 3. Measurements of extracted air

### 3.1 CH4, N2O and CO2 concentrations

#### 3.1.1 Gas chromatography

After taking the aliquot of air for the mass spectrometer analysis, the air left in the original sample tube was measured for the

concentrations of $CH_4$, $CO_2$ and $N_2O$ with two gas chromatographs (Agilent 7890A) (Fig. 3). The settings of the GCs are summarized in Table 1. Briefly, $CH_4$ and $CO_2$ are measured with one GC (GC1) equipped with two Frame Ionized Detectors (FIDs) ($CO_2$ is converted to $CH_4$ by nickel catalyst), and $N_2O$ is measured with another GC (GC2) equipped with an Electron Capture Detector (ECD). We employ capillary columns to obtain high separation and narrow peaks. $CH_4$ and $CO_2$ are separated with a GS-Carbon PLOT (Agilent) capillary column (L = 30 m, i.d. = 0.53 mm, film thickness = 3 μm), and $N_2O$ is separated

with a HP-PLOT Q (Agilent) column (L = 30 m, i.d. = 0.53 mm, film thickness = 40 μm). We use $N_2$ (> 99.99995%, Taiyo Nissan corp., Japan) for carrier and make-up gases, and $H_2$ (> 99.99995%, Taiyo Nissan corp., Japan) for FID for GC1. Hydrocarbon-free air for FID is generated by a zero-air generator (PEAK Scientific, ZA015A). For GC2, we use He carrier (> 99.99995%, Taiyo Nissan corp., Japan) for high separation, and the mixture of Ar and $CH_4$ (5 %) as makeup gas for high sensitivity. We use two gas purifiers in series (a "Mini Fine Purer" from Osaka Gas Liquid and a "Big Universal Trap" from

Agilent) for the carrier, makeup and $H_2$ gases to ensure their purity. Zero air is further purified with a Hydrocarbon/Moisture Trap (Agilent).

To measure a small amount of sample gas, we use small sample loops (1/16" o.d., 0.5 cm³, Valco) filled at sub-ambient pressure. Small sample loops are also effective in reducing baseline fluctuations when GC valves are switched. The dead

volumes of the inlet, fittings and tubing need to be minimized for filling the loops at sufficient pressure. We achieve the total volume (sample loops and dead volumes) of 3.3 ml by using 1/16" tubing (0.7 mm or 1.0 mm i.d.), a customized metal-seal fitting (VCR) with a small bore (1.5 mm i.d.) for the connection of the sample tube, and a customized bracket with small dead volume for the pressure transducer at the inlet (machined Valco cross fitting). This configuration allows us to fill the sample loops at 400 – 600 for the first injection and 300 – 500 hPa for the second injection, for typical ice-core measurements. The





third injection is necessary if the pressure for the first injection exceeds the range of calibration, or if a gas handling error occurs. To minimize the broadening of $CO_2$ peak by passing through the nickel catalyst (Agilent G3440-63002), we replaced the 1/4" (o.d.) tube for packing the catalyst with a 1/8" tube. To minimize adsorption/desorption of trace gases on the inner walls of the tubing, we employ VICI® electroformed Ni tubing or mirror-polished stainless-steel tubing (LaboSolTec). Typical chromatograms are shown in Fig. 4.


A standard gas measurement at atmospheric pressure is conducted as follows. V8 is set to position 1, V1 is set to OFF, V3 is set to OFF, and V7 is opened to allow the standard gas to flow through the two sample loops at 100 ml min-1 for 1.0 min using a mass flow controller (HORIBA STEC, SEC-E40). V7 is closed to stop the gas flow, V3 is set to ON to disconnect the two GCs, and the GC measurements are initiated by switching V1 and V4 to let the carrier gases to flow through the sample loops.

In GC1, $CH_4$ separated by column 1 is detected with the front FID (retention time ~1.8 min). At 1.74 min, V2 is switched to lead $CO_2$ from column 1 to pass through Ni catalyst (to convert to $CH_4$) and then to back FID (retention time ~ 2.3 min). Finally, at 4.6 min, V1 and V2 are switched to the original positions. In GC2, immediately after $N_2O$ passes through column 2 (at 1.5 min), V4 is switched to back-flush column 2 to vent $H_2O$ out from the GC during the run. It is important to prevent the accumulation of $H_2O$ in the columns, which may cause unstable baseline during later measurements. $N_2O$ is further

separated in column 3, and V5 is switched at 1.95 min to lead $N_2O$ to ECD (air eluting before $N_2O$ is vented to the atmosphere). Finally, at 4.89 min, V5 is switched to the original position. After the run, the sample loops are evacuated to < 0.25 hPa (P1) (Paroscientific Digiquartz® Series 2000, absolute 0.16 MPa full-scale).

For a standard gas measurement at sub-ambient pressure, the sample loops are first evacuated for ~30 min by closing V11,

switching V12 and V13 (to connect the sample loops and Dry Pump 1), and turning V3 ON. Then, the standard gas is allowed to flow through Sample Loop 1 by turning V8 to position 1. The flow rate is 100 ml min-1 for 1 min, and then 17 ml min-1 for 1 min. V3 is turned OFF to isolate the pump and start filling the sample loops. When the pressure (P1) reaches a prescribed value, the flow is stopped by closing V7. Then, 15 sec is allowed to stabilize the pressure and the temperature of the sample loops, and the GC measurements are initiated by switching V1 and subsequently V4 (1 sec after V1) to introduce the carrier

gases to the sample loops. The measurement procedures of the GCs are the same as above.

The routine GC calibration and measurement procedures are as follows. We use three standard gases to cover the ranges of greenhouse gas concentrations in the samples (details of our working standard gases are described in the next section and summarized in Table 2). On each day of the measurement, the standard gases in the lines (1/8" tubes) connecting the cylinders

and GC inlet are first pumped out for 5 min with a dry pump, and the standard gases are freshly introduced from the cylinders into the lines. The rest of the standard gas handling and measurements are automated with a custom-made software (with LabVIEW). First, the overall stability of the GC system is assessed by measuring the three standard gases three times at atmospheric pressure. For each standard gas, the peak areas of three consecutive measurements must agree within 1 % to





proceed. The linearities of the detector responses are also checked by comparing the middle standard gas concentrations

calculated from linear interpolation of the concentration-area relationships of high and low standard gases with the original

values. The typical differences are +1.1±1.7 ppb for $CH_4$, -0.3±0.2 ppm for $CO_2$, and +2.3±1.2 ppb for $N_2O$. Then, each of the

three standard gases is measured at three sub-ambient pressures (i.e., nine measurements in total) to construct calibration curves

for the ice-core measurements. Typical pressures are 300, 450 and 600 hPa to cover the pressure range for two injections of

sample air from ~60 g of ice.


After all the standard gas measurements, a sample tube is connected to the GC inlet by VCR, V8 is set to position 4, V10 is

switched ON to evacuate the inlet with a turbomolecular pump for ~20 sec, and the VCR connection is leak-checked with P2

(JTEKT® PMS-5M-2 pressure transducer). The inlet and two sample loops are then evacuated via V10 and V3, respectively,

for 15 min to < 0.25 hPa, V10 is closed, the sample gas is expanded into the inlet by opening the stop valve on the sample

tube, and the controlling software is started. Three seconds later, the two sample loops are connected and isolated from the

vacuum line (V3 OFF), and the sample air is expanded into the sample loops (V8 position 3). The rest of the GC measurement

sequences are the same as the standard gas measurements. After the measurement of a sample, the sample loops are evacuated

by switching V3 for ~2 min (P1 < 0.25 hPa), and the second measurement is initiated automatically. After the end of the second

measurement, the sample tube is replaced with the next one.


After measuring all samples, the standard gases are measured again at the sub-ambient pressures to account for the drifts of

GC signals during the sample measurements. The areas of the standard gases before and after the sample measurements were

linearly interpolated to the time of the sample measurements for calculating the sample concentrations, assuming that the drift

is linear with time.


The concentration of greenhouse gas in the sample air is determined as follows. First, the peak areas of the three standard gases

at the sample pressure (in the sample loop) are calculated by:

$$A_{St,n,P} = a_n P^2 + b_n P + c_n,$$  (1)

where $A_{St,n,P}$ is peak area of standard gas $n$ (= 1 to 3) calculated for the sample pressure ($P$), and $a_n$, $b_n$ and $c_n$ are coefficients

obtained by second-order polynomial fit to the peak area versus pressure from the calibration measurements. Then, the

greenhouse gas concentration in the sample is obtained by:

$$C = dA^2 + eA + f,$$  (2)

where $C$ is concentration, $A$ is sample peak area, and $d$, $e$ and $f$ are coefficients obtained by the second-order polynomial fit to

the standard gas concentrations versus $A_{St,n,P}$ ($n$ = 1 to 3). Each sample air is measured at least twice, and the mean values are

used.



### 3.1.2 Standard gases

The $CH_4$, $N_2O$ and $CO_2$ concentrations are determined against Tohoku University (TU) scales, which are based on gravimetrically prepared primary standard gases (Aoki et al., 1992; Tanaka et al., 1983). Uncertainties of the TU primary standards are $< \pm 0.2$, 0.2 and 0.03% for $CH_4$, $N_2O$ and $CO_2$ concentrations, respectively (Aoki et al. (1992) for $CH_4$, Ishijima

et al. (2001) for $N_2O$ and Tanaka et al. (1987) for $CO_2$). The ice-core standard gases are calibrated using the primary standard gases manufactured in 2008 for $CH_4$ (300.1 – 2799.1 ppb) and $CO_2$ (200.13 – 449.72 ppm), and those made in 1991 for $N_2O$ (100.0 – 400.1 ppb). Working standard gases at NIPR contain $CH_4$, $N_2O$ and $CO_2$ in purified air in 47-L aluminum cylinders (Taiyo Nissan corp., Japan), whose concentrations were calibrated at Tohoku University using their working standard gases named '2007-Ice-Work' (with 25 measurements for each cylinder). We have five working standard gases (named STD 1 – 5)

with different concentrations covering from preindustrial Holocene to glacial maxima (Table 2). Two additional cylinders (STD-A and B) are prepared and calibrated against the NIPR working standard gases at NIPR and used for various tests. Uncertainty of $CH_4$, $N_2O$ and $CO_2$ concentrations of the NIPR working standards are $< \pm 0.6$ ppb, $\pm 0.3$ ppb and $\pm 0.02$ ppm, respectively (one standard error of the mean).

For modern atmospheric concentration levels, the TU scales are in agreement with the NOAA/WMO scales within ~2 ppb for $CH_4$, ~0.3 ppm for $CO_2$, and ~0.5 ppb for $N_2O$ as reported in WMO/IAEA Round Robin Comparison Experiment (Dlugokencky, 2005; Tsuboi et al., 2017) (https://www.esrl.noaa.gov/gmd/ccgg/wmorr/index.html). However, at lower concentration levels, inter-calibration between TU and NOAA scales have not been conducted. For $CH_4$ and $N_2O$, we discuss the consistency of calibration scales by comparing our ice-core data with those from other laboratories (see section 5.1).

### 3.2 Mass spectrometry for isotopic and elemental ratios of $N_2$, $O_2$ and Ar

Isotopic and elemental ratios ($\delta^{15}N/^{14}N$ of $N_2$, $\delta^{18}O/^{16}O$ of $O_2$, $\delta O_2/N_2$ and $\delta Ar/N_2$) are analysed on a dual inlet mass spectrometer (Thermo Fisher Scientific, Delta V) with 9 Faraday cups and amplifiers to simultaneously collect ion beams of molecular masses 28, 29, 32, 33, 34, 36, 38, 40 and 44. The registers, typical beam intensities and other MS settings are given in Table 3. While we collect raw data from all of these cups, we do not use the signals of masses 33, 36 and 38 in this paper

because high precisions for isotopic ratios with these masses (including appropriate corrections for interference and nonlinearity in the mass spectrometer) are not established. To achieve high precision, we control the temperature around the mass spectrometer (especially around the inlet) by isolating the mass spectrometer from the room-air temperature fluctuation with plastic sheets and introducing temperature-controlled air generated by an air conditioner (Orion PAP03B) into the booth. Two large (~45 cm diameter) and a small (~20 cm diameter) fans in the booth vigorously mix the air to maintain the

temperature around the inlet at $25.7 \pm 0.3$ ˚C all year round.



### 3.2.1 Measurement procedures

Our reference gas is commercially available purified air (>99.9999 %, Taiyo Nissan Co.) in a 47-L cylinder filled in 3-liter electropolished stainless-steel containers (hereafter reference cans), each with two bellows-seal valves (Swagelok SS-4H) creating small pipette volume (1.3 cm$_3$) at the exit. The inner surface of the can is preconditioned by humidified $O_2$ at > 120

°C as for the extraction line. The reference can attached to the standard side is rarely disconnected.

Our mass spectrometry largely follows Severinghaus et al. (2009). Prior to the daily sample measurements, a reference can is connected to the sample port of Delta V using VCO® fitting (typically on the prior evening to stabilize the can's temperature), and the ports and pipets are evacuated. The MS valves leading to the reference cans are closed, and both bellows are evacuated

for 5 min. Then, the MS valves to the inlet ports are opened to check leak by an ion gauge, and all bellows and lines are further evacuated for 5 min. On both sides, the reference gas is introduced into the pipette of the can by closing the valve at the MS side and opening the other valve, and they are equilibrated for 10 min. The pipette volumes are disconnected from the cans, and the aliquots are expanded into the bellows and equilibrated for 10 min. Then, the bellows are isolated from the inlet and compressed to reach ~30 mbar. The initial pressure in the fully expanded bellows is ~28 mbar from a freshly filled can. The

reference can is replaced by a new one when the initial pressure decreases to ~18 mbar.

The reference gas from the sample port is measured against the standard side ("can versus can") for 4 blocks to check the standard deviations. Before each block, the acceleration voltage is optimized by centering the mass 40 peak, the background is measured after a 120-sec idle time, and the pressures are adjusted to 5000 ± 50 mV for mass 28 (3 × 10$_8$ Ω), automatically

with the ISODAT software. The idle time and integration time are 10 seconds and 16 seconds, respectively. Each block consists of 17 changeover cycles, but the first cycle is discarded and only the latter 16 values are used. After running the 4 blocks, two blocks are run by imbalancing the sample pressure by ± 10 % against the standard side for obtaining pressure imbalance sensitivity (see below).

The GEPW tube containing the sample air is connected to the sample port, and the sample port is evacuated during the previous measurement. The procedure of the sample measurement is the same as the "can vs. can" measurement, except that the sample expansion into the bellow is made in one step by simply opening the tube valve. We run two blocks for each sample to obtain a total of 32 cycles. Typical standard deviations in 1 block (16 cycles) are 0.013, 0.029, 0.010 and 0.017 ‰ for $\delta^{15}$N, $\delta^{18}$O, $\delta O_2/N_2$ and $\delta Ar/N_2$, respectively.

### 3.2.2 Pressure imbalance and chemical slope corrections

The ratios of ion currents of different masses are slightly sensitive to the pressure in the ion source, thus a correction is applied with an established procedure (Severinghaus et al., 2003). The pressure imbalance sensitivity (PIS) is a slope of δ values





against differences in beam intensity ($\Delta P = (I_{sa}/I_{st} - 1) \times 1000$ where $I$ is the mean beam intensity in one block), which is determined by measuring the reference gas at the sample side at three pressures (at $\Delta P = 0$, +100 and -100 ‰). The PIS is

measured every day and used for correcting the sample values measured on the same day by:

$$\delta_{pressure\ corrected} = \delta_{measured} - (PIS)\ \Delta P. \tag{3}$$

The PIS gradually changes over several weeks, and they shift after a filament replacement.

Relative ionization efficiencies of gas are also sensitive to variations in the mixing ratio of the gas in total air (the sensitivity

is called "chemical slope") (Severinghaus et al., 2003). The chemical slopes are determined from the measurements of the reference gas added by pure $O_2$ (+10, +20 and +30 % of original $O_2$ amount) for $\delta^{15}N$, and pure $N_2$ (+10, +20 and +30% of original $N_2$ amount) for $\delta^{18}O$. The correction is made by:

$$\delta^{15}N_{slope\ corrected} = \delta^{15}N_{pressure\ corrected} - [CS1] \times \delta O_2/N_{2measured} \tag{4}$$

$$\delta^{18}O_{slope\ corrected} = \delta^{18}O_{pressure\ corrected} - [CS2] \times \delta N_2/O_{2measured} \tag{5}$$


where CS1 and CS2 are chemical slopes for $\delta^{15}N$ and $\delta^{18}O$, respectively. The chemical slopes are fairly stable, thus are measured only a few times per year. The typical values of CS1 and CS2 are 0.0005 ‰/‰ and 0.0018 ‰/‰, respectively.

The final normalization against the modern atmosphere (the ultimate standard gas for ice cores by definition) requires through

investigation of the stability of reference gases and the atmospheric ratios, which we discuss in section 4.2.

### 3.3 Total air content

Total air content (TAC) is the amount of occluded air in a unit mass of ice ($ml_{STP}$ kg-1) (Martinerie et al., 1992). In our system, TAC is calculated from:

$$TAC = \frac{P}{1013.25} \cdot \frac{(V_a + V_c) \cdot (V_a + V_b)}{V_a} \cdot \frac{273.15}{T} \cdot \frac{1}{m}, \tag{6}$$

where $P$ is pressure in the sample loops upon first expansion, $T$ is air temperature near GC, $m$ is mass of ice sample (measured with an electronic balance, SARTORIUS CP4202S), and $V_a$, $V_b$ and $V_c$ are the volume of the sample tube, the volume of the pipette at the split line, and the combined volume of GC inlet and sample loops, respectively.

$V_a$ and $V_b$ were determined manometrically against a known volume (118.7 cm3) with a pressure gauge (Paroscientific

Digiquartz® Model 745-100A, absolute 0.69 MPa full-scale). The whole apparatus is first evacuated, the air is introduced from a cylinder into the glass flask at about atmospheric pressure. The air is expanded into the manifold, pipette, and tube, and the pressure measured at each step is used to calculate the volumes by the ideal gas law. The expansion and recording are repeated 10 times, and they are averaged.





Similarly, $V_c$ was determined manometrically for each sample tube (~6.6 cm3), which was attached to the GC inlet. N2 or air in a sample tube at a known pressure was expanded into the evacuated GC inlet and sample loops, and the pressure was recorded (valve on the sample tube is kept open). The expansions/evacuations were repeated a few times, and the gas in the sample tube was re-filled. The whole procedures were repeated a few times to obtain a total of 12 measurements for each tube.

The calibration of the volumes $V_a$, $V_b$ and $V_c$ must be made for individual sample tubes because the volumes in the valves and end connections are slightly different from each other ($V_a$, $V_b$ and $V_c$ are different by up to 0.8 %, 0.6% and 0.5%, respectively, between the tubes). Average $V_a$, $V_b$ and $V_c$ are 6.6 cm3, 1.4 cm3 and 3.4 cm3, respectively. The standard error of the mean for the 10 – 12 measurements of $V_a$, $V_b$ and $V_c$ are 0.04 %, 0.10 % and 0.04 %, respectively. By propagating these values and the uncertainties of temperature (assumed to be 1 K), pressure (assumed to be 16 Pa) and ice mass (assumed to be 0.1 g), 1σ

uncertainty of TAC is estimated to be 0.5 mlSTP kg-1.

## 4. Evaluation of system performance using standard gas and atmosphere

During air extraction, splitting and analyses, alteration of air composition may occur for various reasons, such as gas dissolution or chemical reaction in the meltwater, degassing from inner surfaces of vessel and line, and diffusive fractionations of isotopic ratios. Below, we evaluate the performance of the tubes, apparatus and instruments by various tests and controlled

measurements (mimicking ice-core analyses with standard gas and gas-free ice).

### 4.1 CH4, N2O and CO2 concentrations

### 4.1.1 Tube storing test

We evaluate the concentration changes during gas storage in the sample tubes and test tubes (used for injecting standard gas to the apparatus, with metal-seal valves at both ends), by filling standard gas from a cylinder (STD-A) into evacuated tubes,

and measuring the sample tubes on the following day and the test tubes on the same day. The changes in CH4, N2O and CO2 concentrations thus obtained are insignificant with respect to the measurement precisions for both the sample tubes (+0.8±2.1 ppb, +1.3±2.1 ppb and +0.1±1.1 ppm, respectively, with n = 25) and test tubes (+0.7±2.2 ppb, +0.9±1.0 ppb and −0.2±0.1 ppm, respectively, with n = 17) (Table 4). The excellent results of the storing tests are attributable to the passivation treatment of the tubes and the use of valves with clean inner surfaces (Fujikin metal diaphragm valves). We note from our earlier

experience that CH4 is produced by up to several ppb by opening and closing metal bellows valves (Swagelok SS-4H) if they become old (after several hundred operations), and that CO2 concentration increases by up to ~10 ppm if the passivation treatment is insufficient.





### 4.1.2 Standard gas transfer test

Gas-free ice for tests is made from ultra-pure water in a stainless-steel vessel (~1800 cm$^3$) sealed with a Conflat blank flange.
The water is boiled for 20 – 30 min with an open outlet port on the flange, and cooled to room temperature after closing the outlet, and then put in a freezer at -20 ˚C. The side of the vessel is surrounded by insulation so that the water is frozen from the bottom over a few days. The ice is removed from the vessel, and ice with visible cracks and bubbles are removed (more than half of the ice).

Standard gas from a cylinder is flushed through a pre-evacuated line and test tube (volume is 3 or 5 cm$^3$) at 50 ml min$^{-1}$ for 5 min, and sampled at atmospheric pressure after ceasing the flow by closing an upstream valve. The relatively low flow rate for flushing prevents thermal fractionation of the gas due to adiabatic expansion at the pressure regulator (important for isotopic analyses), and the pre-evacuation and >200 ml of total flow ensure clean sampling.

The test tube with standard gas and vessel with ~50 g of gas-free ice are attached to the extraction line, and the vessel is evacuated for 120 min. The ice is melted while the vessel is evacuated (for 15 min) to remove any air degassed from the melt. We also use ice-core melt instead of gas-free ice melt for the blank test. In this case, the vessel with ice-core melt is evacuated for 30 min after a sample extraction to pump out any residual air. Then, the standard gas from the test tube is slowly injected into the extraction system and transferred continuously over the gas-free water into a sample tube, maintaining the pressure
similar to that of ice-core extraction.

The standard gas thus transferred to the sample tubes are measured on the following day, after handling it with the split line (see below for the results of isotopic analyses). No significant changes in $CH_4$, $N_2O$ and $CO_2$ concentrations are observed with respect to the mean values of the test tubes' storing tests (+0.8±2.7 ppb -0.1±1.7 ppb and +1.3±0.7 ppm, Table 4). Based on
the above results, we apply no corrections for $CH_4$, $N_2O$ and $CO_2$ concentrations.

### 4.2 Isotopic and elemental ratios of $N_2$, $O_2$ and Ar

### 4.2.1 Storing test of GEPW tubes

To evaluate the possible effect of gas storage in the GEPW tubes for one day, the reference gas was transferred to the tubes using the split line and measured the following day, and the results were compared with those measured on the same day. They
are identical within the measurement uncertainties for all ratios (-0.002 ± 0.003 ‰ for $\delta^{15}N$, -0.002 ± 0.010 ‰ for $\delta^{18}O$, -0.012 ± 0.056 ‰ for $\delta O_2/N_2$, -0.013 ± 0.042 ‰ for $\delta Ar/N_2$, with n=14), thus no corrections are applied for the storage duration in the GEPW tubes.



### 4.2.2 Standard gas transfer test

The standard gas filled in a test tube was transferred to a sample tube with the extraction line and gas-free water (the same
experiment as in section 4.1.2), and an aliquot was taken with the split line and transferred to a GEPW tube as the ice-core
analyses (Fig. 5). On the other hand, the same standard gas filled in the same test tube was attached directly to the split line,
and its aliquot was transferred to a GEPW tube, skipping the extraction line and overnight storage (Fig. 5). Comparison of the
measured values from these two experiments gives the changes in the isotopic and elemental ratios during the ice-core air
extraction and overnight storage, denoted as $\Delta\delta_{extraction}$. The values of $\Delta\delta_{extraction}$ are -0.005 ± 0.001, -0.003 ± 0.002 and -0.102
± 0.011 for $\delta^{15}N$, $\delta^{18}O$ and $\delta Ar/N_2$, respectively (errors are standard error, n > 100). The signs of changes are negative for all
gases, which may suggest a slightly less effective transfer of heavier isotopes with our extraction line. Based on these results,
we employ the above values for $\delta^{15}N$ and $\delta Ar/N_2$ as constant corrections, and no correction for $\delta^{18}O$, for the ice-core data. We
also conducted similar tests with ~1 ml$_{STP}$ sample size using a small test tube, in which whole gas was transferred to a GEPW
tube without splitting. The changes are not significantly different from those of the larger sample sizes (-0.002 ± 0.002, +0.004
± 0.006 and -0.113 ± 0.048 ‰ for $\delta^{15}N$, $\delta^{18}O$ and $\delta Ar/N_2$, respectively (errors are standard error with n = 9)).

Relatively large decrease and dependence on the sample size are found for $\delta O_2/N_2$ in the above tests (-0.193 ± 0.015, -0.293 ±
0.029 and -0.482 ± 0.048 ‰ for 5, 3 and 1 cm$_3$, respectively). We interpret the result as $O_2$ consumption by the inner walls of
the extraction line and sample tubes, whose magnitude (number of $O_2$ molecules consumed) might be only weakly dependent
on the sample size. We use an exponential fit to the above data (Fig. 6),

$$\Delta\delta_{extraction, O2/N2} = 0.55362 \exp(-0.31606 \times V) - 0.078802 \text{ (‰)},\qquad(7)$$

where $V$ is the sample size of air in ml$_{STP}$ for correcting the ice core data.

### 4.2.3 Long-term stability of standard gas and atmosphere, and normalization of sample ratios

For normalization of the ice-core data and monitoring its long-term stability, standard gas in a cylinder (STD-A) and
atmosphere (sampled outside the NIPR building) have been regularly measured against the reference can.

The atmospheric sampling and measurement procedures follow Headly (2008) and Orsi (2013). Briefly, the atmosphere is
collected in a 1.5 L glass flask with a metal piston pump (Senior Aerospace, MB-158), aspirated air intake and two water traps.
The flow rates of the sampling and aspiration lines are 4 and 15 L min$_{-1}$, respectively, with a flushing time of > 10 min before
sampling. In the laboratory, the flask air is expanded into three volumes in series (~4, ~1.5 and ~2 ml) and allowed for 30 min
to equilibrate, and the air in the middle volume is transferred to a GEPW tube. The STD-A is filled in a test tube, and an aliquot
of it is transferred to a GEPW tube using the split line (see section 4.1.2).



The $\delta_{15}$N, $\delta_{18}$O, $\delta O_2/N_2$, and $\delta Ar/N_2$ of STD-A and atmosphere for 2016 to 2019 are shown in Fig. 5 and Fig. 7, and the values

of the STD-A against the atmosphere is summarised in Table 5. Shifts in the ratios are commonly seen when the ion source filament or reference can is renewed (as indicated by vertical lines in the figure). However, there are no discernible trends and seasonal variations during the use of one reference can for all ratios except for $\delta O_2/N_2$. Typical standard deviations of a set of atmospheric measurements (~10 replicates) using two flasks within a few days are 0.003, 0.007, 0.020, and 0.034 ‰ for $\delta_{15}$N, $\delta_{18}$O, $\delta O_2/N_2$, and $\delta Ar/N_2$, respectively, while those of the STD-A measurements are 0.004, 0.008, 0.058, and 0.033 ‰,

respectively. Comparison of $\delta_{15}$N, $\delta_{18}$O and $\delta Ar/N_2$ between STD-A and atmosphere indicate slightly better reproducibilities for the atmospheric measurements, possibly due to fractionations during the filling of STD-A into the test tubes. Therefore, the atmosphere is the best choice for the normalization of those ratios.

General trends towards more positive values are seen for $\delta O_2/N_2$ (Fig. 5d and Fig. 7d), presumably because $O_2$ in the reference

can is gradually consumed by oxidation of organic matter on the inner wall. Moreover, atmospheric $\delta O_2/N_2$ in urban areas may show spikes and seasonal variations (Ishidoya and Murayama, 2014), which are much larger than our measurement precision. Indeed, $\delta O_2/N_2$ of STD-A only shows linear trends (due to the drift in the reference can), but the atmospheric $\delta O_2/N_2$ sometimes deviates from its linear trend by up to ~0.5 ‰ (e.g., in Dec. 2017, Jan. 2018, April 2018, Mar. 2019 and Dec. 2019). Thus, the use of a standard gas in the cylinder (STD-A) for normalization, rather than the atmosphere sampled at the time of calibrations,

is the better choice for precise $\delta O_2/N_2$ measurements. We here define our "modern air" for $\delta O_2/N_2$ as the annual average $\delta O_2/N_2$ in 2017 observed over Minamitorishima island (hereafter MTS air) (24°17'N, 153°59'E) observed by National Institute of Advanced Industrial Science and Technology (AIST) in cooperation with Japan Meteorological Agency (JMA) (Ishidoya, 2017), and determined $\delta O_2/N_2$ in STD-A against it (+2.595 ± 0.008 ‰). We note that AIST recently developed a gravimetric $\delta O_2/N_2$ calibration scale for precise and long-term atmospheric monitoring (Aoki et al., 2019).


The values of atmosphere or STD-A against the reference can within a few weeks immediately before and after the sample measurements are averaged and used for normalizations. The final corrected and normalized $\delta$ value of an ice core sample is:

$$\delta_{normalized} = \left[ \frac{(\delta_{slope\ corrected} - \Delta\delta_{extraction}) \cdot 10^{-3} + 1}{\delta_{ATM} \cdot 10^{-3} + 1} - 1 \right] \cdot 10^3 \ (\text{‰}) , \qquad (8)$$

where $\Delta\delta_{extraction}$ is the correction for air extraction and overnight storage (only for $\delta_{15}$N, $\delta O_2/N_2$ and $\delta Ar/N_2$, see section 4.2.2),

and $\delta_{ATM}$ is the atmospheric value corrected for PIS and chemical slope.

## 5. Ice core analyses and comparison with published records

We analysed the Dome Fuji (hereafter DF) ice core, Antarctica, and NEEM ice core, Greenland, and compared the results with other records to evaluate the overall reliabilities of our methods. The reproducibilities of ice-core measurements are also assessed using the pooled standard deviation of duplicates (measurements of two ice samples from the same depth,



Severinghaus et al. (2003)) for some depths. The number of samples and depths are as follows: 49 samples from 40 depths in 112.88 – 157.81 m (bubbly ice, 2.0 – 0.2 kyr BP), and 70 samples from 35 depths in 1245.00 – 1918.59 m (clathrate ice, 79 – 150 kyr BP) from the DF core, and 75 samples from 47 depths in 112.68 – 449.10 m (bubbly ice and above brittle zone, 2.0 – 0.2 kyr BP) from the NEEM core.

We employed the following age scales and synchronizations. For the preindustrial late Holocene (~0 to ~1800 year C.E.), the GICC05 chronology was used for the NEEM core as published by Rasmussen et al. (2013), and a WAIS Divide Core gas chronology (WDC05A) (Mitchell et al., 2013; Mitchell et al., 2011) was transferred to the DF core by $CH_4$ synchronization (the tie points are shown in Fig. 9 and Table 6). For the other ice cores for comparisons (including GISP2, WDC, Law Dome cores), we employed their own published time scales (Table A1).


The following data were rejected or not acquired due to experimental errors. The DF sample at 144.75 m lost $CH_4$, $CO_2$ and $N_2O$ concentrations and TAC because of a connection failure between the GC and computer. $\delta^{15}N$, $\delta^{18}O$, $\delta O_2/N_2$ and $\delta Ar/N_2$ from the DF core at 1521.06, 1540.56 and 1712.10 m were rejected because of the leaky valve on the GEPW tube. The NEEM sample at 217.15 m showed anomalous $CO_2$ and $N_2O$ concentrations as compared with another sample at the same depth (+83

ppm and +47 ppb, respectively); all GC data including $CH_4$ were rejected for this sample. The NEEM sample at 229.80 m and 438.83 m showed anomalously low $\delta^{15}N$ and $\delta^{18}O$ (half of the typical values, or lower), possibly due to gas handling error or leak. As all the anomalous NEEM data were acquired within 2 months after establishing the method, there would have been experimental errors that slipped from our attention.

### 5.1 $CH_4$, $N_2O$ and $CO_2$ concentrations

The pooled standard deviations of the $CH_4$, $N_2O$ and $CO_2$ concentrations are ± 3.2 ppb, ± 1.3 ppb and ± 3.2 ppm for the DF bubbly ice (number of pairs = 8), ± 3.3 ppb, ± 2.2 ppb and ± 3.1 ppm for the DF clathrate ice (n = 29), and ± 2.9 ppb, ± 3.0 ppb and ± 5.5 ppm for the NEEM bubbly ice (n = 25) (Table 7). The pooled standard deviations of $CH_4$ and $N_2O$ are similar to those reported from most precise measurements by other laboratories (± 2.8 ppb for $CH_4$ by OSU (Mitchell et al., 2013), and ± 1.5 ppb for $N_2O$ by Seoul National University (Ryu et al., 2018)).


Our new $CH_4$, $N_2O$ and $CO_2$ data from the DF core agree with the previous data from Tohoku University (Fig. 8) (Kawamura, 2001), indicating consistency of the TU concentration scales for ice core analyses over the past ~ 20 years. We compare our results for the preindustrial late Holocene (~0 to ~1800 C.E.) at ~50-year resolution with other ice core records from other groups on the NOAA concentration scales (Fig. 9). The DF $CH_4$ data agree well with those from the WAIS Divide core by

OSU (Mitchell et al., 2013) and Law Dome cores by CSIRO (Rubino et al., 2019), which are currently the best Antarctic records in terms of precision and resolution (see Fig. A1 in Appendix for comparison with other records). We note that multi-decadal variations are smoothed out in the DF core because of the slow bubble-trapping process. For Greenland, our NEEM





data show good agreement with the GISP2 data from OSU (Mitchell et al., 2013), including multidecadal to centennial-scale variations.


We note that unrealistically high $CH_4$ variabilities were found at two depths in the NEEM core (417.60 – 418.00 m and 361.05 – 361.35 m; gas ages are ~218 C.E. and ~532 C.E., respectively) (Fig. 10). The change of ~20 to 50 ppb between the neighbouring depths (only < 1 year apart in age) is impossible to be an atmospheric origin considering diffusive mixing in firn. The good agreements between the duplicate measurements for these depths exclude the possibility of analytical error. The $N_2O$

concentrations at the same depths are not significantly different from those in the neighbouring depths, suggesting that the $CH_4$ anomalies are not due to ice-sheet surface melt and associated gas dissolution, which should elevate both $CH_4$ and $N_2O$. In any case, we exclude these anomalously high concentrations from the calculation of pooled standard deviation and comparison with other ice core records. The $CH_4$ concentration of 723.7 ppb at 961 C.E. (the mean of four discrete measurements), in the middle of an increase over a century, is also higher than the GISP2 data by ~30 ppb. The four discrete values agree within 12

ppb (717.6, 724.9, 729.5 and 722.9 ppb), indicating again that the large deviation from the GISP2 data is not due to analytical error. The reason for this discrepancy is not clear, but it could be due to uncertainty in age synchronization between the cores, a reversal of the NEEM gas age by firn layering (the possibility that the bubbles were closed-off in the last stage of firn-ice transition, Rhodes et al. (2016)) or natural artefacts (e.g., gas dissolution by surface melt, or biological $CH_4$ production within the ice sheet).


$N_2O$ concentrations from both polar regions should agree with each other within the uncertainty of ice core analyses. Our data from the NEEM core agree with those from the Law Dome core by CSIRO within ~5 ppb without systematic bias (Rubino et al., 2019) (see Fig. A1 in Appendix for comparison with other records). The DF data also agree with the NEEM data within ~5 ppb. The sample at 1076 C.E. (129.16 m) of the DF core shows very high concentration (~20 ppb higher than the

neighbouring depths), which is unlikely to be due to experimental errors because the $CH_4$ and $CO_2$ concentrations of the same sample are not elevated. Anomalous $N_2O$ concentrations were also found in late Holocene DF samples in previous measurements (Kawamura, 2001), and they are possibly natural artefacts ($N_2O$ production in ice sheet) (Kawamura, 2001; Sowers, 2001).

The overall agreements of our $CH_4$ and $N_2O$ data with the other datasets suggest the reliability of our method and consistency of the TU scales at low concentrations with the NOAA scales. We note that our method does not apply experimental corrections for $CH_4$ and $N_2O$ concentrations. For reference, the OSU method (wet extraction with refreezing) applies solubility correction of 1 % (~3 – 8 ppb) and blank correction of 2.5 ppb for $CH_4$ (Mitchell et al., 2011), and the CSIRO method (dry extraction) applies blank corrections of 4.1 ppb and 1.8 ppb for $CH_4$ and $N_2O$, respectively (MacFarling Meure et al., 2006). The negligible

effect of gas dissolution in our method is explained by the immediate removal of the released air from the ice vessel, maintaining low pressure above the meltwater.



The $CO_2$ concentration is generally measured with dry extraction methods (e.g., (Ahn et al., 2009; Barnola et al., 1987; Monnin et al., 2001; Nakazawa et al., 1993a), because wet extraction method has a risk of contamination by acid-carbonate reaction and oxidation of organic materials in meltwater. While $CO_2$ production in the meltwater indeed occurs, its magnitude is up to 20 ppm, and the glacial-interglacial $CO_2$ variations are well captured in the DF record for the last 340 kyr BP (Kawamura et al., 2003). Our new wet-extraction $CO_2$ values of the DF core for the last 2000 years agree with the Law Dome (MacFarling Meure et al., 2006), EDML (Siegenthaler et al., 2005) and WAIS Divide (Ahn et al., 2012) ice cores mostly within 0 to +10 ppm, with several ~20-ppm deviations. The NEEM wet-extraction $CO_2$ data are higher than those of DF and other ice cores by ~10 – 30 ppm (maximum ~60 ppm), which is much larger than the pooled standard deviation (5.3 ppm for the NEEM dataset). The primary reason for the high $CO_2$ values in the NEEM ice core is high impurity contents in the core, which cause in-situ $CO_2$ production in the ice sheet (Anklin et al., 1995).

## 5.2 Elemental and isotopic compositions of $N_2$, $O_2$ and Ar

### 5.2.1 Gas-loss fractionation and surface removal

Previous studies have indicated that gases can slightly be lost from ice cores during storage, causing size- and mass-dependent fractionations in $\delta O_2/N_2$, $\delta^{18}O$ and $\delta Ar/N_2$ (Bender et al., 1995; Bereiter et al., 2009; Huber et al., 2006a; Ikeda-Fukazawa et al., 2005; Kawamura et al., 2007; Severinghaus et al., 2009). To examine whether those ratios as originally recorded in the ice sheet can be obtained from the DF core, which had been stored at -50 °C for ~20 years, we measured samples from the same depths but with different thickness of surface removal (for example, 8 and 5 mm) (Fig. 11). The outer ice were also measured and compared with the values from the inner ice.

First, we compare the data from the outer ice and inner ice to validate the magnitude of gas-loss induced fractionation. For both the bubbly ice and clathrate ice (note that bubble-clathrate transition zone is not investigated), all the measured samples show lower $\delta O_2/N_2$ and $\delta Ar/N_2$ in the outer ice than those in the inner ice (Fig. 12). The outer ice has more depleted $\delta O_2/N_2$ and $\delta Ar/N_2$ than the inner ice, and $\delta O_2/N_2$ is more depleted than $\delta Ar/N_2$, indicating significant size-dependent fractionation in the outer ice. Most samples show higher $\delta^{18}O$ in the outer ice than in the inner ice, while $\delta^{15}N$ from the outer and inner ice agree to each other, suggesting that significant mass-dependent fractionation also occurred for $O_2$, but not for $N_2$, in the outer ice.

Next, we examine the $\delta O_2/N_2$ data from the inner ice with different thicknesses of surface removal. Below 1380 m (pure clathrate ice), the $\delta O_2/N_2$ values from the inner ice with the outer removal of 5 mm are mostly lower than those from the adjacent pieces with 8-mm removal (Fig. 13), suggesting that gas loss affects the gas composition to more than 5 mm from the



surface. On the other hand, no significant differences are observed between $\delta O_2/N_2$ values from ice with the removal of more

than 8 mm (Fig. 14). The $\delta Ar/N_2$, $\delta^{15}N$ and $\delta^{18}O$ data from the inner ice with surface removal of 5 mm and 8 mm are not

different from each other, suggesting insignificant mass-dependent gas-loss fractionation in ice > 5 mm away from the surface.

From these results, we conclude that the removal of more than 8 mm is sufficient to obtain the gas composition as originally

trapped in the DF1 core.

### 5.2.2 Reproducibility and comparison with previous data

The pooled standard deviations for the DF clathrate hydrate ice with removal thickness of >8 mm are 0.006, 0.010, 0.089 and

0.115 ‰ for $\delta^{15}N$, $\delta^{18}O$, $\delta O_2/N_2$ and $\delta Ar/N_2$, respectively (Table 7). The reproducibility of $\delta O_2/N_2$ is one order of magnitude

better than those previously reported (Bender, 2002; Extier et al., 2018). The reproducibility for $\delta^{15}N$ and $\delta^{18}O$ are comparable

to but slightly worse than the most precise measurements by SIO (Seltzer et al., 2017; Severinghaus et al., 2009).

We compare our new DF data with previous data from Tohoku University (Fig. 15) (Kawamura, 2001; Kawamura et al., 2007).

The previous $\delta O_2/N_2$ data were significantly depleted due to gas loss during the sample storage at -25 °C (Fig. 15c, grey marks),

thus they were corrected for effect by assuming a linear relationship between the storage duration and $\delta O_2/N_2$ (Fig. 15c, red

marks). Our new $\delta O_2/N_2$ data agree with the gas-loss corrected old data, suggesting that $\delta O_2/N_2$ as originally trapped in the DF

core can be measured from the 20-year old samples. The new $\delta^{15}N$ and $\delta^{18}O$ data generally agree with those of Kawamura

(2001) and Kawamura et al. (2007) within the uncertainty of the old data, although the large uncertainties of the old datasets

do not permit precise comparisons.

The duplicate measurements of bubbly ice of the NEEM core (by removing more than 3 mm from the surface) produced pooled

standard deviations for $\delta^{15}N$ and $\delta^{18}O$ (0.006 and 0.008 ‰) similar to those for the DF clathrate ice (Table 7). This suggests

that the removal of 3 mm is sufficient for the bubbly ice at least for the isotopic ratios, possibly due to generally low pressure

of bubbly ice (because it is shallower) compared with clathrate ice. On the other hand, pooled standard deviations for $\delta O_2/N_2$

and $\delta Ar/N_2$ of the bubble ice are much larger than those for the DF clathrate ice (0.236 and 0.120 ‰ for the DF core, and 0.775

and 0.450 ‰ for the NEEM core, respectively), possibly related with the natural variability of pressure and composition of

individual air bubbles with different trapping histories (Ikeda-Fukazawa et al., 2001; Kobashi et al., 2015). The larger pooled

standard deviations for the NEEM core than those of the DF core possibly reflect natural difference within the ice sheets, or

artefacts (gas loss) during drilling, handling and storage at the NEEM site associated with the warmer environment than the

Dome Fuji drilling site. We also note that, due to the small number of duplicates for the bubbly ice, it is difficult at this stage

to assess whether there are small systematic lowering of $\delta O_2/N_2$ and $\delta Ar/N_2$ with the 3-mm removal.





## 5.3 Total air content


The pooled standard deviations for TAC are 0.66 and 0.67 $ml_{STP}$ $kg^{-1}$ for both DF bubbly ice and clathrate ice. The data from the clathrate ice agree with those from previous measurements using ~300 g of ice (Kawamura, 2001), while the data from the bubbly ice appear to be lower than the previous data, especially for the shallowest depths (Fig. 15). These results may be explained by the fact that TAC of bubbly ice is biased towards lower values due to the so-called "cut-bubble effect" (Martinerie

et al., 1990), in which bubbles intersecting the sample surfaces are cut and lose air. The cut-bubble effect is larger for samples with a smaller surface-to-volume ratio and samples from shallower depths.

## 6. Conclusions

We presented a new analytical technic for high-precision, simultaneous measurements of $CH_4$, $N_2O$ and $CO_2$ concentrations, isotopic and elemental ratios of $N_2$, $O_2$ and Ar, and total air content, from a single ice core sample with relatively small size

(50 – 70 g) by a wet extraction. The ice sample is melted under vacuum in 3 min, and the released air is continuously transferred and cryogenically trapped into a sample tube, with the total duration for extraction of about 10 minutes. The rapid and continuous transfer minimizes contaminations due to degassing from the inner walls of the apparatus, as well as dissolution of the sample air into the meltwater. The extracted air is homogenized in the sample tube for one night, and split into two aliquots for mass spectrometric measurement (~20 % of the sample) and gas chromatographic measurement (~80 % of the sample).


The system performance was evaluated by measuring the standard gas after treating it as the ice-core air extraction, by passing it through the extraction and split lines with gas-free water in the extraction vessel. We do not observe significant changes in the mean $CH_4$, $N_2O$ and $CO_2$ concentrations, possibly because of the long evacuation, rapid and continuous gas transfer at low pressure over meltwater, and passivation treatments of the extraction lines and sample tubes. Thus, we do not apply corrections

(e.g., so-called blank correction and solubility correction) for the greenhouse gas concentrations. For the mass spectrometry, we do not observe significant changes in $\delta_{18}O$, while we observe changes in $\delta_{15}N$, $\delta O_2/N_2$, $\delta Ar/N_2$. Moreover, the change in $\delta O_2/N_2$ is dependent on the sample size. Thus, we apply constant corrections for $\delta_{15}N$ and $\delta Ar/N_2$, and sample-size-dependent correction for $\delta O_2/N_2$.

Standard deviations of duplicate measurements for DF clathrate ice are 3.2 ppb, 2.2 ppb, and 3.1 ppm for $CH_4$, $N_2O$ and $CO_2$ concentrations, respectively, and 0.006, 0.010, 0.09 and 0.12 ‰ for $\delta_{15}N$, $\delta_{18}O$, $\delta O_2/N_2$ and $\delta Ar/N_2$, respectively. The $CH_4$ and $N_2O$ data from the DF and NEEM ice cores for the last 2,000 years agree well with those from the GISP2, WAIS Divide and Law Dome cores. We also demonstrate significant gas-loss induced depletion of $\delta O_2/N_2$ in the ice near the sample surface of the DF clathrate ice, which has been stored at -50 °C over ~20 years. The original $\delta O_2/N_2$, $\delta Ar/N_2$, $\delta_{15}N$ and $\delta_{18}O$ in the ice

sheet may still be obtained by removing the sample surface by > 8 mm.



Our new method will have many paleoclimatic applications, such as detecting subtle variations in greenhouse gas cycles (in particular $CH_4$ inter-polar difference and $N_2O$ variations), hydrological cycles ($\delta^{18}O$ of $O_2$), insolation signals for dating ($\delta O_2/N_2$ and $\delta Ar/N_2$), and local climatic and glaciological conditions ($\delta^{15}N$ and TAC) from deep ice cores with high temporal 675 resolution.

**Data availability**

Gas data of the Dome Fuji and NEEM ice cores are available at the NIPR ADS data repository (https://ads.nipr.ac.jp/dataset/A20200501-001; for inquiry: oyabu.ikumi@nipr.ac.jp).

**Author contribution**

K. Kawamura designed the wet extraction apparatus and mass spectrometer configuration, and developed the overall concept of the splitting and multiple-species analyses. K. Kawamura, SS and AK designed the GC configurations. JPS and RB developed the calibration protocols for atmospheric normalization of mass spectrometer measurements except for $\delta O_2/N_2$, and customized ISODAT scripts. RD developed GC controlling software with discussion with K. Kawamura. SI provided the $\delta O_2/N_2$ standard scale. DDJ provided the NEEM ice core samples. KGA provided funding for the mass spectrometer, and the 685 NEEM samples as the national representative. SA and TN provided the greenhouse gas concentration scales and funding for the wet extraction apparatus and cryostat. IO and K. Kitamura made the measurements. IO, K. Kitamura and K. Kawamura established the detailed procedures of ice core measurements and calibrations. CS contributed to the early stages of development. IO and K. Kawamura wrote the manuscript, and all authors contributed to the discussion.

**Competing interests**

The authors declare that they have no conflict of interest.

**Acknowledgments**

We thank Yumiko Miura for a part of the data reduction, Akito Tanaka, Satoko Nakanishi, Mariko Hayakawa, and Satomi Oda for the measurements and Shinji Morimoto for providing the Tohoku University scales and assisting our standard gas calibrations. We acknowledge the Dome Fuji drilling projects led by Okitsugu Watanabe, Yoshiyuki Fujii and Hideaki 695 Motoyama and participants in the Japanese Antarctic Research Expedition and Ice Core Consortium related to the Dome Fuji fieldworks and managements. NEEM is directed and organized by the Center of Ice and Climate at the Niels Bohr Institute and US NSF, Office of Polar Programs. It is supported by funding agencies and institutions in Belgium (FNRS-CFB and



FWO), Canada (NRCan/GSC), China (CAS), Denmark (FIST), France (IPEV, CNRS/INSU, CEA and ANR), Germany (AWI), Iceland (RannIs), Japan (NIPR), Korea (KOPRI), The Netherlands (NWO/ALW), Sweden (VR), Switzerland (SNF), United Kingdom (NERC), and the USA (US NSF, Office of Polar Programs). This study was supported by Japan Society for the Promotion of Science (JSPS) and Ministry of Education, Culture, Sports, Science and Technology-Japan (MEXT) KAKENHI Grant Numbers 17K12816, 17J00769 and 20H04327 to IO, 20H00639, 17H06316, 15KK0027, 26241011, 21671001 and 18749002 to K. Kawamura, 22221002 to KGA, 18K01129 to SI, 22310003 to TN, the GRENE Arctic Climate Change Research Project of the MEXT to SA, the Global Environment Research Coordination System from the Ministry of the Environment to SI, and Project Research KP305 and Senshin Project from National Institute of Polar Research.

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



# Figures

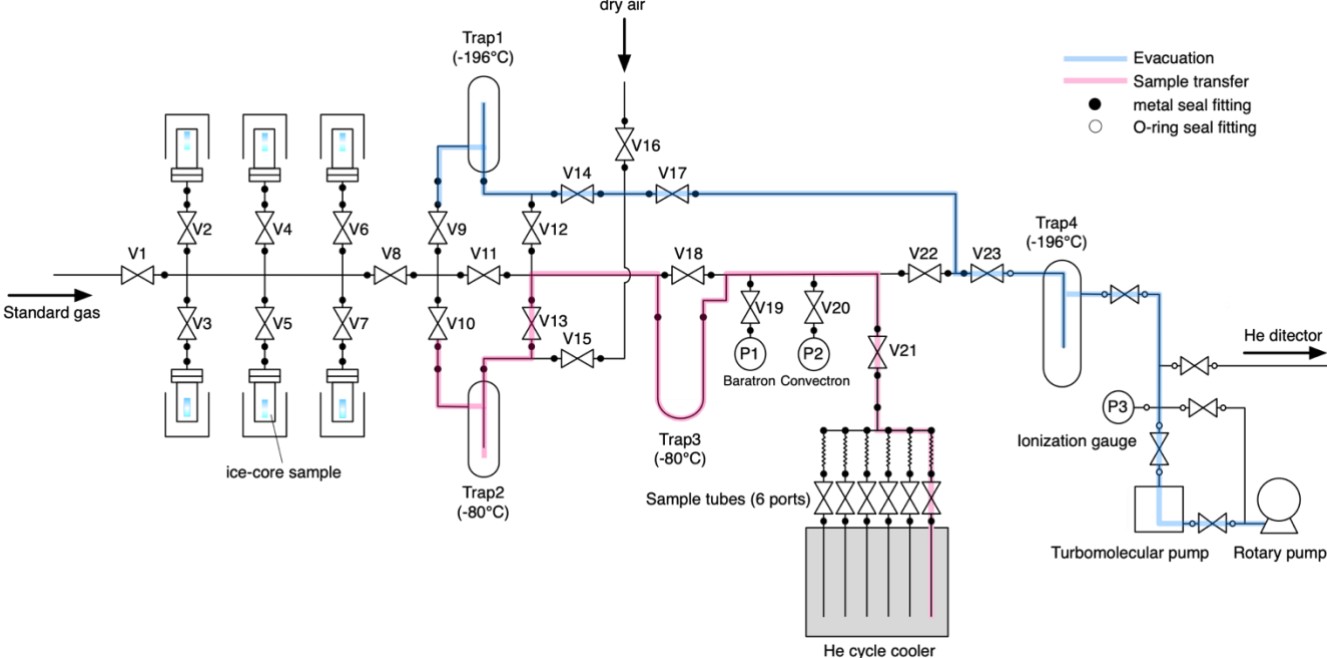

915            **Figure 1:** Schematic diagram of the wet extraction system.




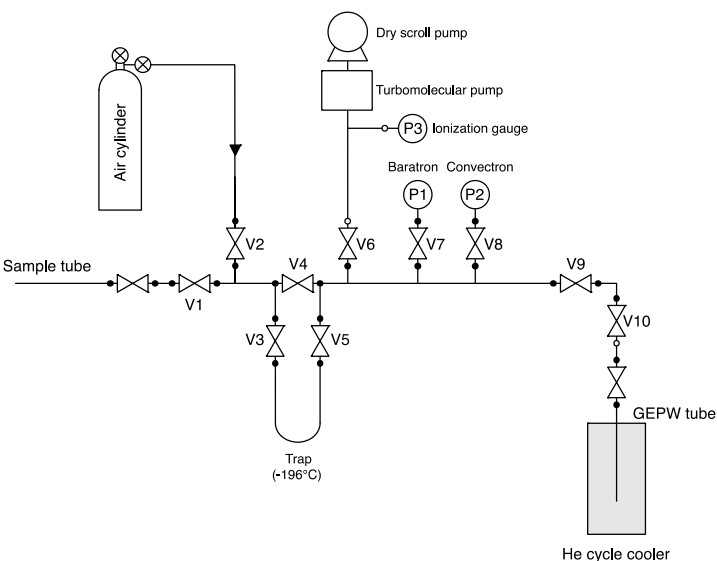

**Figure 2:** Schematic diagram of the split line.




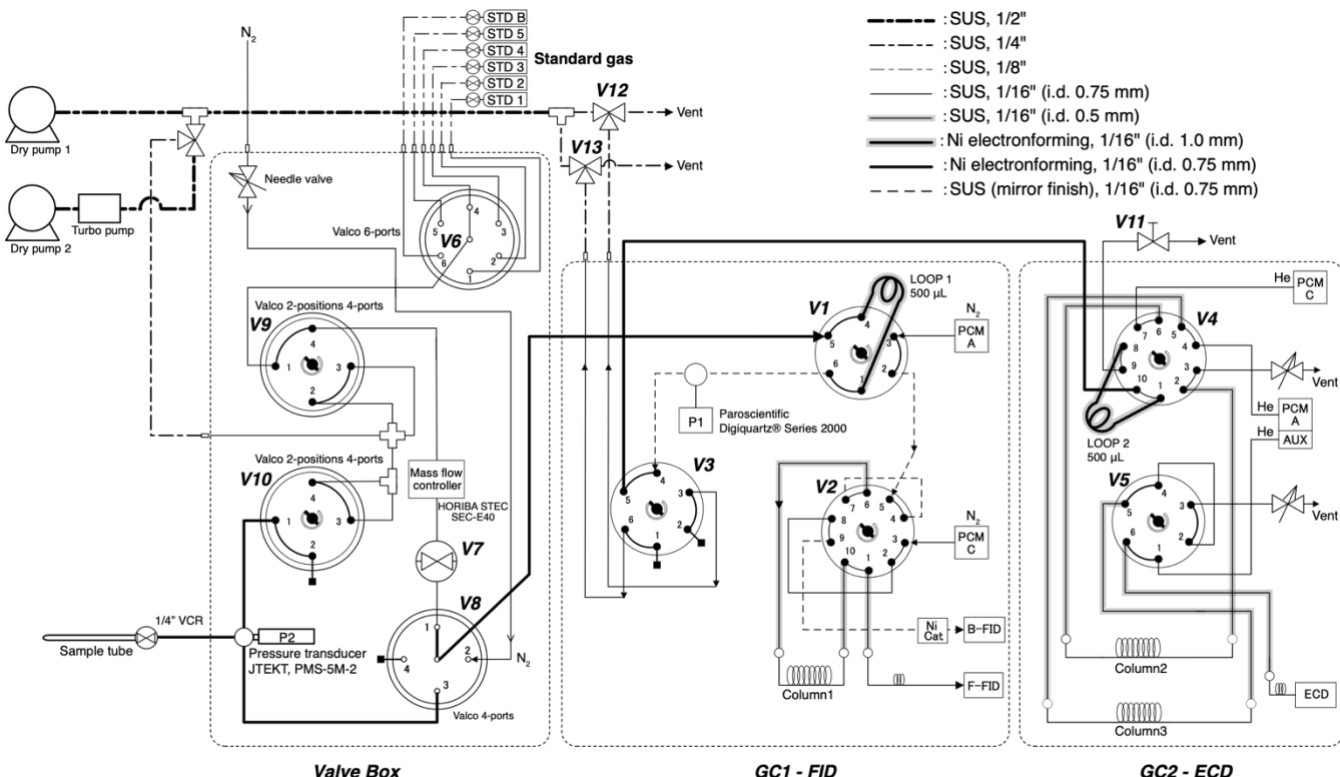

**Figure 3:** Schematic diagram of gas chromatographs and inlet. All two-position valves are in "OFF" positions.

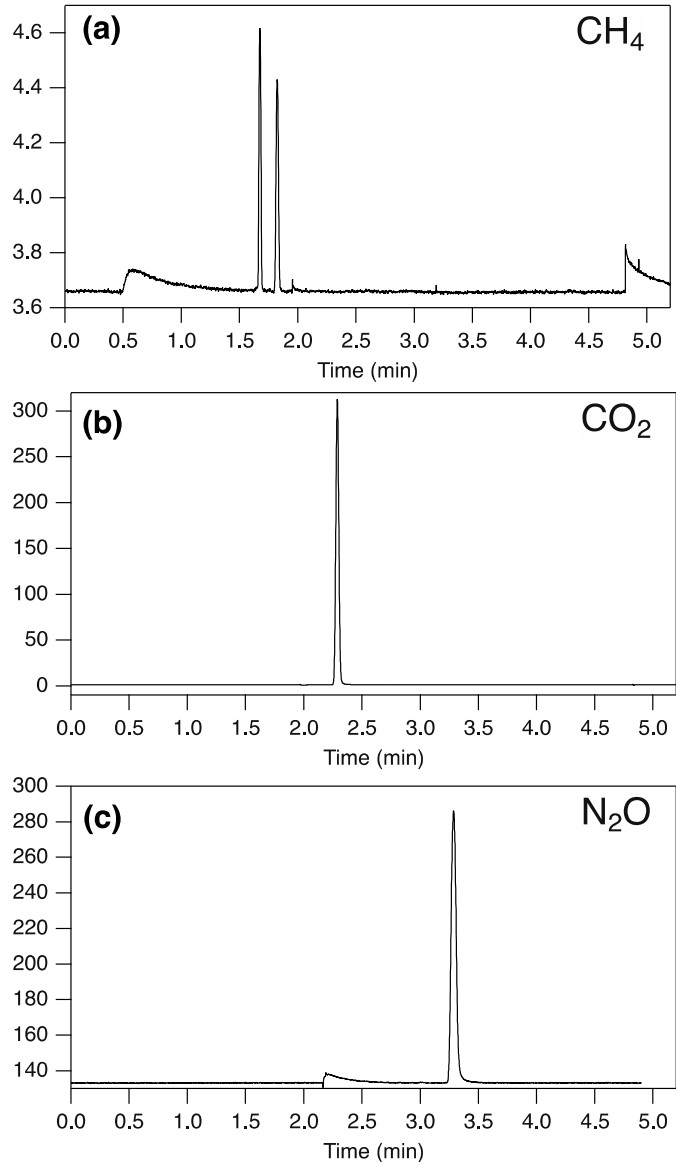


**Figure 4:** Typical chromatogram of (a) Front FID for CH4 (the largest peak is O2, and the second-largest peak is CH4), (b) Back FID for CO2, and (c) ECD for N2O.

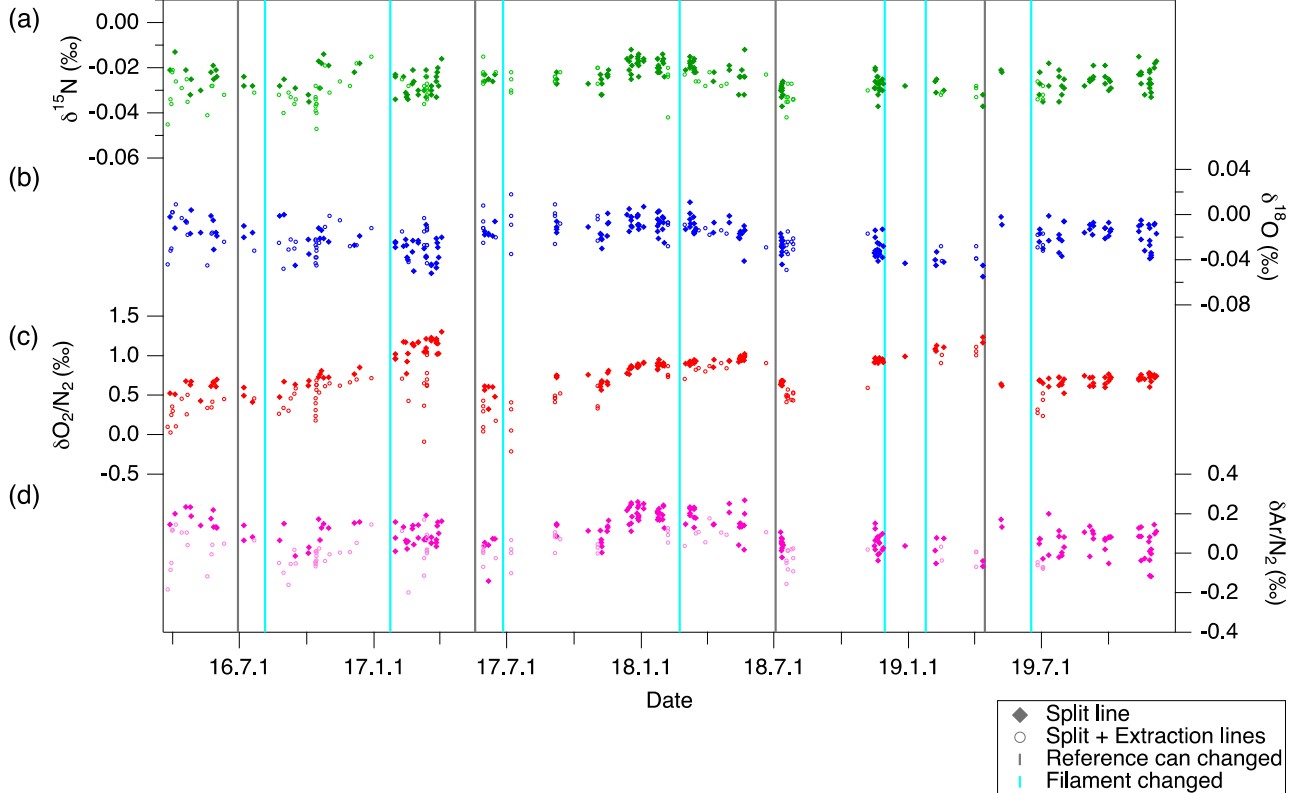


**Figure 5:** Standard gas (STDA) composition measured against reference gas for (a) $\delta^{15}N$, (b) $\delta^{18}O$, (c) $\delta O_2/N_2$, and (d) $\delta Ar/N_2$. Filled markers represent samples transferred only through the split line, and open markers represent samples transferred through both extraction and split lines. Vertical grey lines indicate the timing of replacements of reference can, and vertical light blue lines indicate the timing of filament replacements.






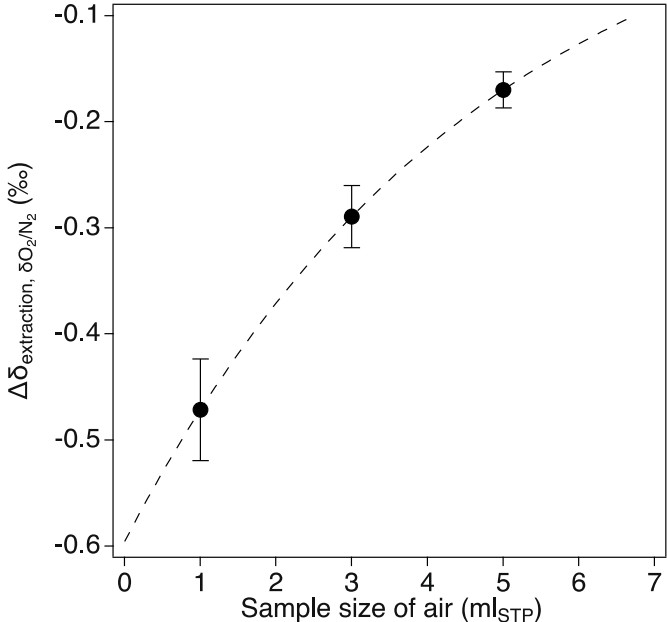

**Figure 6:** Change of $\delta O_2/N_2$ by wet extraction and overnight storage. Dashed line represents exponential fit to the data ( $\Delta\delta_{extraction,\ O2/N2}$ = -0.55362 exp(-0.31606 × $V$) + -0.078802).




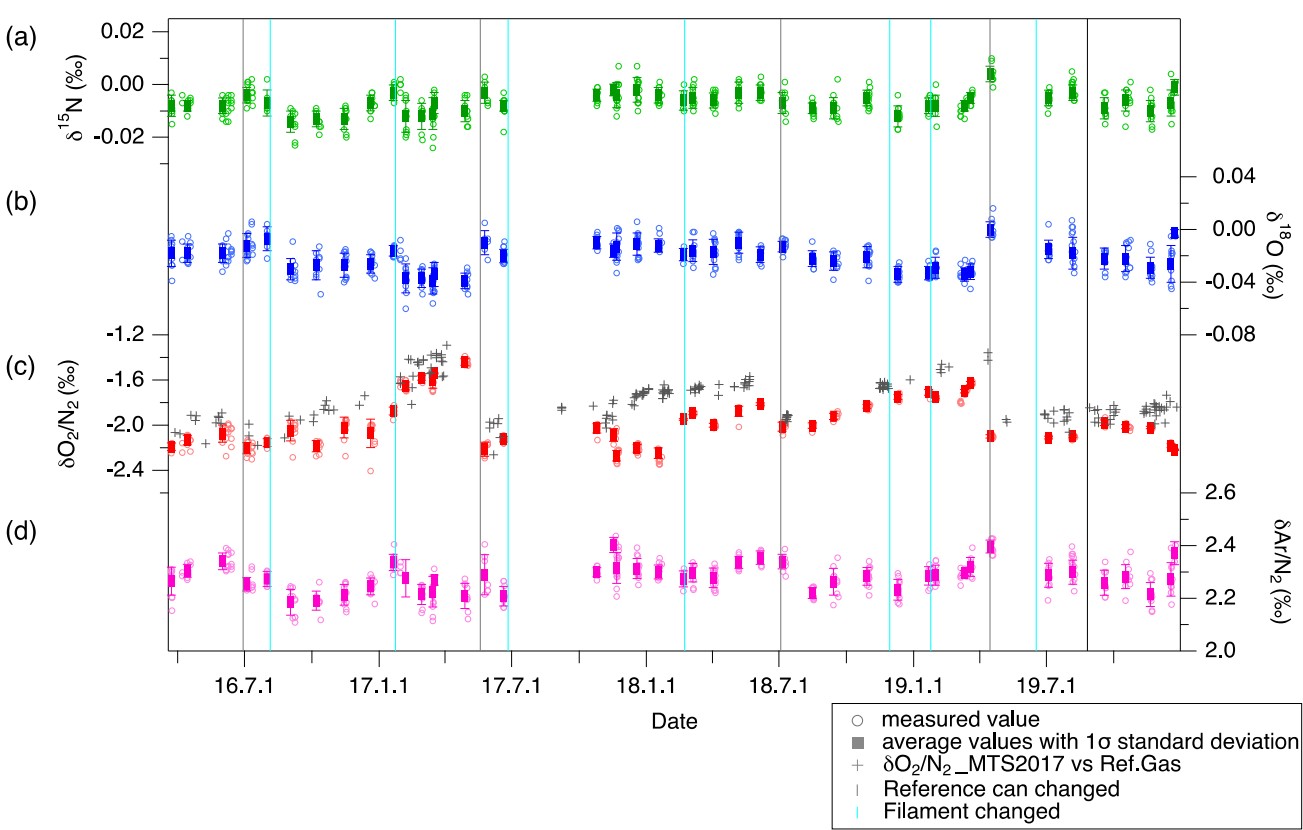

**Figure 7:** Atmospheric composition measured against reference gas for (a) $\delta^{15}N$, (b) $\delta^{18}O$, (c) $\delta O_2/N_2$, and (d) $\delta Ar/N_2$. Open markers represent individual data points, whereas filled markers represent the means of values measured within several days (error bars are one standard deviations). Grey plus (+) markers in (d) represent estimated $\delta O_2/N_2$ of MTS-2017 against reference gas through the measurements of STD-A against the reference gas, assuming that $\delta O_2/N_2$ of STD-A has not changed. Vertical grey lines indicate the timing of replacements of reference can, and vertical light blue lines indicate the timing of filament replacements.





**Figure 8:** CH₄, N₂O and CO₂ concentrations of the Dome Fuji ice core, and comparison with previous records from the same core (Kawamura, 2000 and Kawamura et al., 2007).


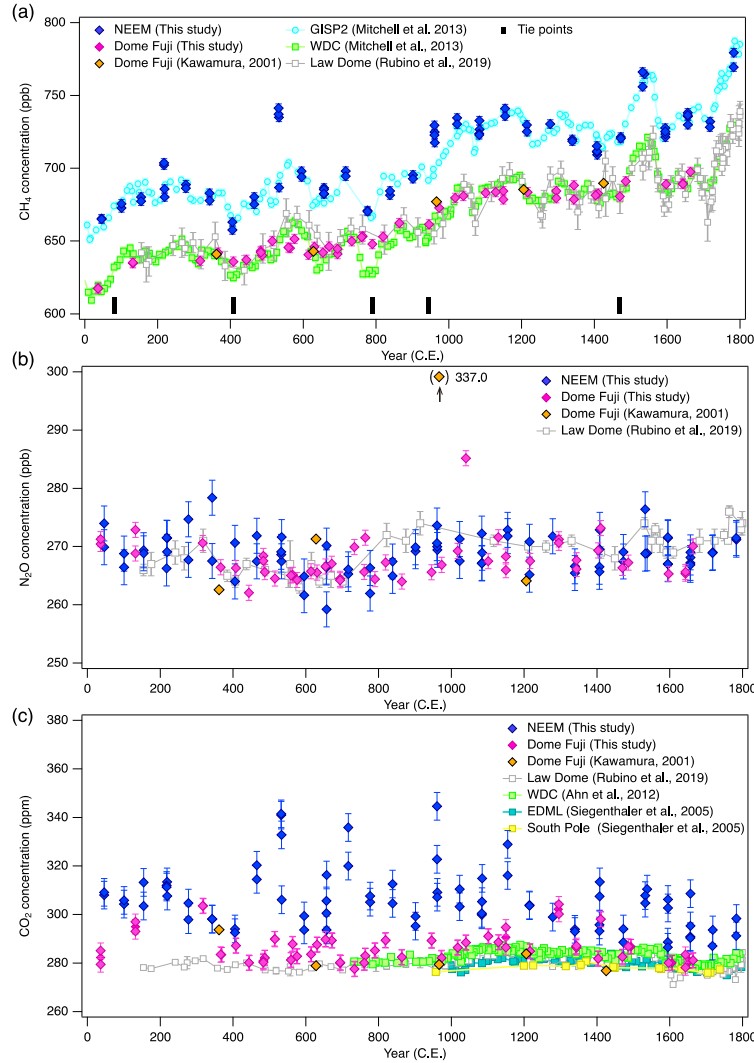

**Figure 9:** (a) CH₄, (b) N₂O and (c) CO₂ concentrations for 0 – 1800 C.E. from the DF and NEEM ice cores measured with our new method, and the comparison with published records from other groups (Ahn et al., 2012; Kawamura, 2001; Kawamura et al., 2007; Mitchell et al., 2013; Rubino et al., 2019; Siegenthaler et al., 2005). The DF data is placed on the WDC05A chronology by placing 5 tie points between the CH₄ variations of the DF and WD cores (thick tick marks at the bottom of (a)), and all the other data are placed on the respective (published) time scales. Details are summarized in Table A1.



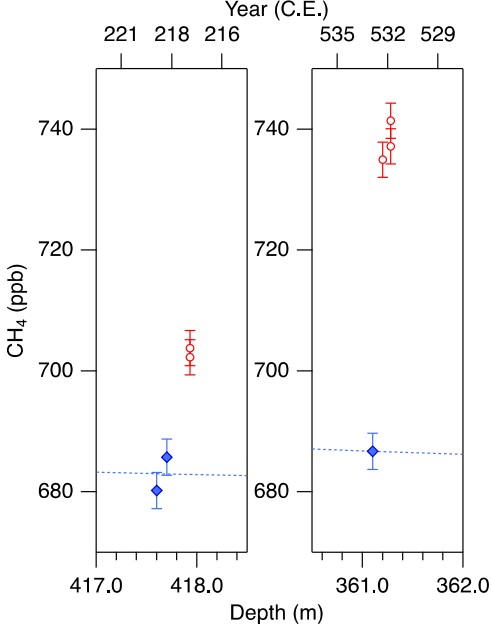


**Figure 10:** Detailed views of individual CH4 data for the abrupt (non-atmospheric) increases in the NEEM core at ~418 and 361 m. Data shown in blue agree with the GISP2 data, and those in red are unrealistically high (interpreted as natural artefacts). The dotted line (blue) connects the blue markers (mean of two data for 417 m and one data for 361 m) with their neighbouring data points.






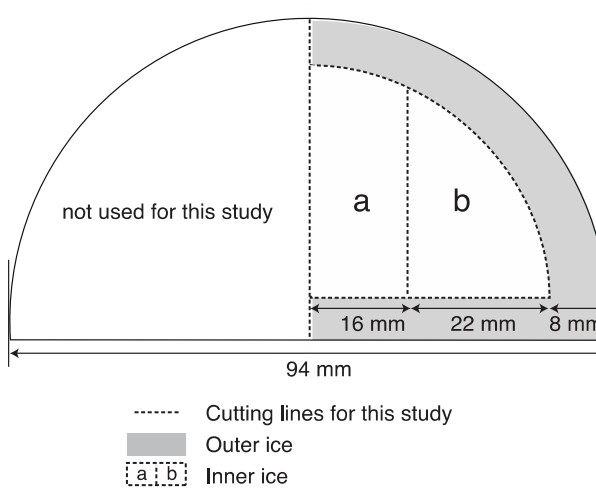

**Figure: 11:** Typical cross-sectional cutting plan of the DF core for duplicate measurements and outer-inner comparisons for mass spectrometer analyses. The sample length is ~12 cm. The original outer surface (black line) has been exposed to the atmosphere for ~20 years. Dotted lines indicate the boundaries between "a," "b," and "outer" pieces. For single (non-duplicate) measurements, "a" piece is not cut.



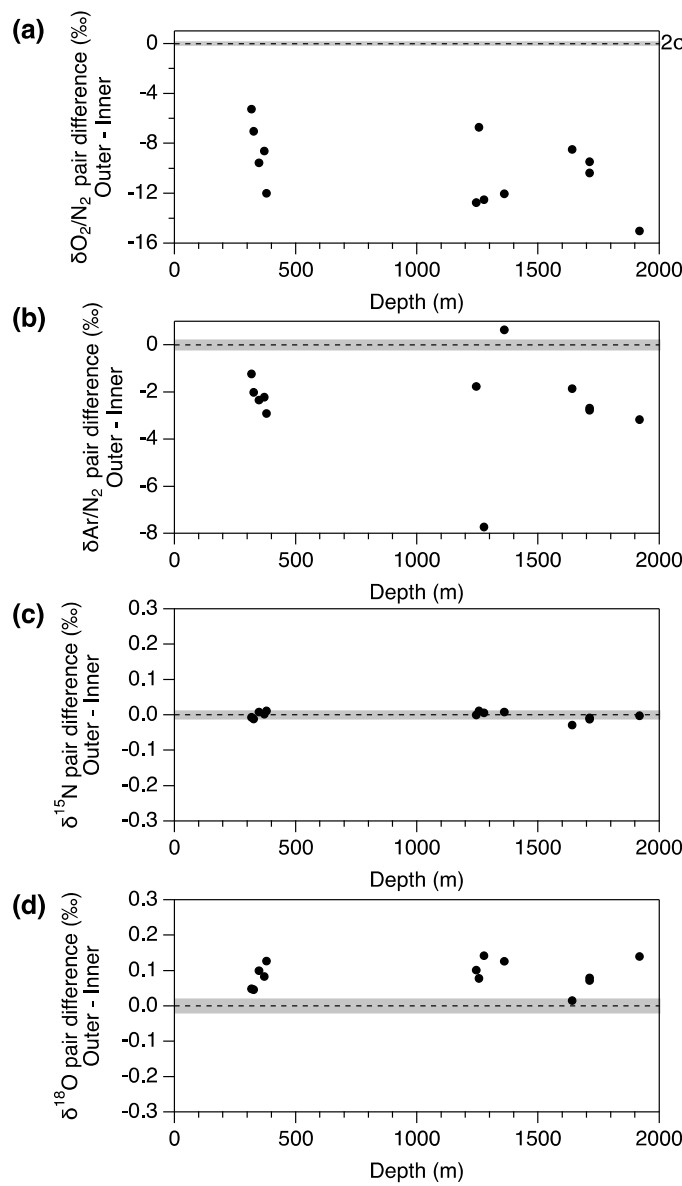

**Figure 12:** Difference between outer ice and inner ice for (a) $\delta O_2/N_2$, (b) $\delta Ar/N_2$, (c) $\delta^{15}N$, and (d) $\delta^{18}O$. Grey shadings indicate the estimated
$2\sigma$ uncertainty for clathrate ice (from pooled standard deviations of duplicates with > 8 mm of outer removal).



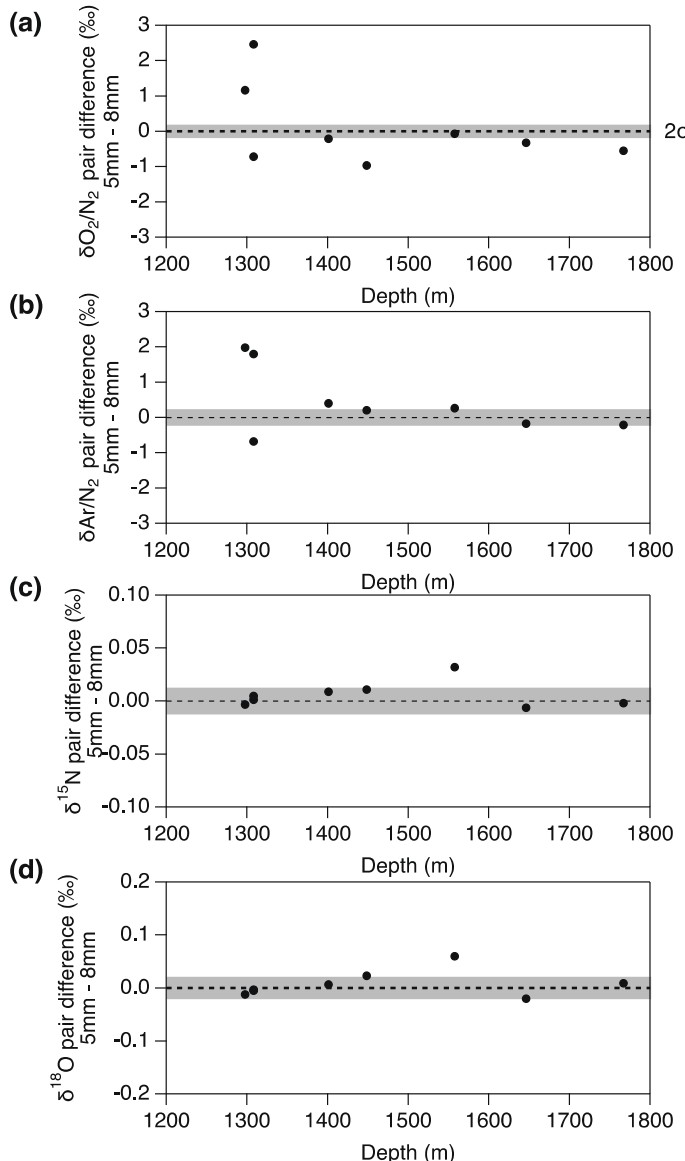

**Figure 13:** Pair difference between the two inner ice with the thickness of outer removal of 5 mm and 8 mm for (a) $\delta O_2/N_2$, (b) $\delta Ar/N_2$, (c) $\delta^{15}N$, and (d) $\delta^{18}O$. Grey shadings indicate the estimated $2\sigma$ uncertainty for clathrate ice (from pooled standard deviations of duplicates with > 8 mm of outer removal).



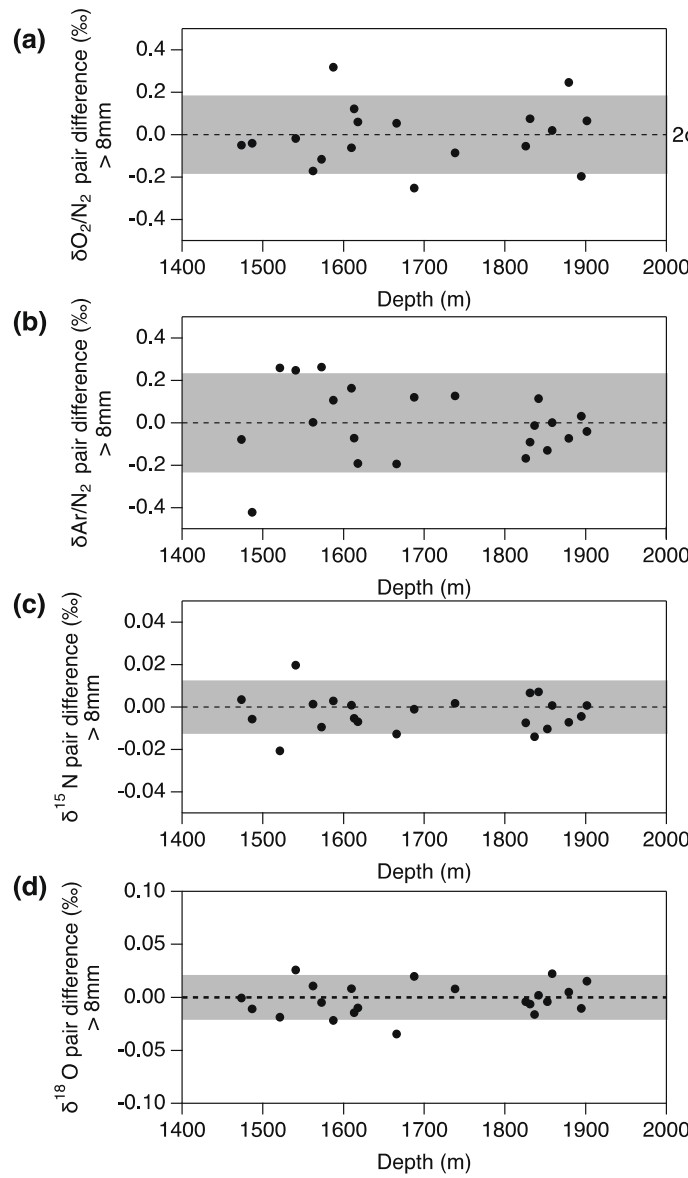

**Figure 14:** Pair difference between the two inner ice with the thickness of outer removal of > 8 mm for (a) $\delta O_2/N_2$, (b) $\delta Ar/N_2$, (c) $\delta^{15}N$, and (d) $\delta^{18}O$. Grey shadings indicate the estimated $2\sigma$ uncertainty for clathrate ice (from pooled standard deviations of duplicates with > 8 mm of outer removal).



**Figure15:** The DF records of mass spectrometer measurements and TAC from this study (filled markers), and comparison with previous records from the same core (crosses connected with dotted lines, Kawamura, 2000; Kawamura et al., 2007). (a) δ15N, (b) δ18O, (c) δO2/N2, (d) δAr/N2, and (e) TAC. For the previous δO2/N2 records in the right panel of (c), both raw data (black) and corrected data for gas-loss fractionation during core storage at -25 °C (red) are shown.



## Tables

Table 1:  Settings of gas chromatographs.

|  | GC1 | | GC2 |
|---|---|---|---|
|  | CH$_4$ | CO$_2$ | N$_2$O |
| Carrier gas | N$_2$, 10 ml min$^{-1}$ | | He, 7 ml min$^{-1}$ |
| Sample loop volume | 0.5 ml | | 0.5 ml |
| Oven temperature | 30 ˚C | | 30 ˚C |
| Column | GS-CarbonPLOT | | HP-PLOT/Q |
| Length | 30 m | | 30 m |
| Internal diameter | 0.53 mm | | 0.53 mm |
| Film thickness | 3 μm | | 40 μm |
| Ni-catalyst temperature | None | 400 ˚C | None |
| Detector | FID | FID | ECD |
| Temperature | 200 ˚C | 200 ˚C | 325 ˚C |
| H$_2$ flow rate | 35 ml min$^{-1}$ | 40 ml min$^{-1}$ | None |
| Air flow rate | 400 ml min$^{-1}$ | 400 ml min$^{-1}$ | None |
| Make-up gas | N$_2$, 20 ml min$^{-1}$ | N$_2$, 20 ml min$^{-1}$ | Ar+CH$_4$ (5%), 10 ml min$^{-1}$ |

Table: 2: Standard gases.

|  | STD 1 | STD 2 | STD 3 | STD 4 | STD 5 | STD A | STD B | scale |
|---|---|---|---|---|---|---|---|---|
| Cylinder ID | CQB09303 | CQB12403 | CQB09309 | CQB07938 | CQB09336 | CRC00059 | CRC00057 | |
| CH$_4$ [ppb] | 255.3 | 430.9 | 609.9 | 796.2 | 968.8 | 526.7 | 720.2 | TU-2008 |
| N$_2$O [ppb] | 186.2 | 222.9 | 259.7 | 298.1 | 317.9 | 241.3 | 273.2 | TU-2006 |
| CO$_2$ [ppm] | 169.79 | 209.25 | 249.64 | 289.37 | 330.71 | 229.04 | 269.21 | TU-2008 |





**Table 3: Collector configurations of the mass spectrometer.**

| Mass | Slit width (mm) | Resistor ($\Omega$) | Typical ion current (A) |
|---|---|---|---|
| 28 | 2.0 | $3 \times 10^8$ | $2 \times 10^{-8}$ |
| 29 | 3.8 | $3 \times 10^{10}$ | $1 \times 10^{-10}$ |
| 32 | 3.8 | $1 \times 10^9$ | $4 \times 10^{-9}$ |
| 33 | 1.4 | $1 \times 10^{12}$ | $3 \times 10^{-12}$ |
| 34 | 3.8 | $3 \times 10^{11}$ | $2 \times 10^{-11}$ |
| 36 | 2.0 | $1 \times 10^{12}$ | $1 \times 10^{-12}$ |
| 38 | 1.4 | $1 \times 10^{12}$ | $2 \times 10^{-13}$ |
| 40 | 2.0 | $3 \times 10^9$ | $3 \times 10^{-10}$ |
| 44 | 2.4 | $1 \times 10^{10}$ | $4 \times 10^{-12}$ |


**Table 4: Test results using a standard gas (STD-A) for greenhouse gases.**

| | | $CH_4$ (ppb) | $N_2O$ (ppb) | $CO_2$ (ppm) | n |
|---|---|---|---|---|---|
| Sample tubes | Average | +0.8 | +1.3 | +0.1 | 25 |
| (overnight storing) | Std. dev. | 2.1 | 2.1 | 1.1 | |
| | | | | | |
| Test tubes | Average | +0.7 | +0.9 | -0.2 | 17 |
| (immediate measurement) | Std. dev. | 2.2 | 1.0 | 0.1 | |
| | | | | | |
| Extraction line | Average | +1.5 | +0.8 | +1.1 | 9 |
| (mimicking ice core extraction) | Std. dev. | 1.6 | 1.3 | 0.7 | |

Values are differences from the calibrated concentrations of the STD-A cylinder (Table 3)


**Table 5: Composition of STD-A for mass spectrometer measurements.**

| | $\delta^{15}N$ (‰) | $\delta^{18}O$ (‰) | $\delta O_2/N_2$ (‰) | $\delta Ar/N_2$ (‰) | n |
|---|---|---|---|---|---|
| Mean | -0.018 | 0.002 | 2.595 | -2.171 | 214 (18 for $\delta O_2/N_2$) |
| Std. error | <0.001 | <0.001 | 0.008 | 0.006 | |

$\delta O_2/N_2$ is determined in Jul. 2019 against the 2017 annual mean $\delta O_2/N_2$ over Minamitorishima island provided by AIST. Other ratios are

determined against outside air sampled monthly at NIPR over Feb. 2016 – Dec. 2019.





**Table 6: Age control points for the DF core from CH$_4$ matching to the WDC core (0 – 1800 C.E.)**

| DF depth | WDC05A age | Approximate 1σ error [a] |
|---|---|---|
| (m) | (C.E.) | (year) |
| 119.23 | 1470 | 42 |
| 131.29 | 945 | 42 |
| 134.17 | 790 | 33 |
| 148.65 | 408 | 41 |
| 156.515 | 80 | 44 |

[a] estimated as the half of the mean age intervals from the tie point to the neighbouring CH$_4$ data points (uncertainty of WDC05A itself is also considered).


**Table 7: Pooled standard deviations for the NEEM and Dome Fuji ice cores.**

| | CH$_4$ | N$_2$O | CO$_2$ | TAC | Number | δ$^{15}$N | δ$^{18}$O | δO$_2$/N$_2$ | δAr/N$_2$ | Number |
|---|---|---|---|---|---|---|---|---|---|---|
| | (ppb) | (ppb) | (ppm) | (ml$_{STP}$ kg$^{-1}$) | of pairs | (‰) | (‰) | (‰) | (‰) | of pairs |
| NEEM (bubble) | 2.9 | 3.0 | 5.5 | | 25 | 0.006 | 0.008 | 0.775 | 0.450 | 23 |
| DF (bubble) | 3.2 | 1.3 | 3.2 | 0.66 | 8 | 0.009 | 0.018 | 0.236 | 0.120 | 8 |
| DF (clathrate) | 3.2 | 2.2 | 3.1 | 0.67 | 29 | 0.006 | 0.010 | 0.089 | 0.115 | 22 |




## Appendix A: Comparison with other available records

**Table A1: Ice cores and time scales shown in this study.**

|  | Ice core | Reference | Offset (ppb) | Time scale |
|---|---|---|---|---|
| CH4 | NEEM | This study |  | GICC05 (Rasmussen et al., 2013) |
|  | Dome Fuji | This study |  | WDC05A (Mitchell et al., 2013; Mitchell et al., 2011) |
|  | Dome Fuji | Kawamura (2001) |  | WDC05A (Mitchell et al., 2013; Mitchell et al., 2011) |
|  | GISP2 | Mitchell et al. (2013) |  | WDC05A (Mitchell et al., 2013; Mitchell et al., 2011) |
|  | GISP2 | Brook (2009) |  | WDC05A (Mitchell et al., 2013; Mitchell et al., 2011) |
|  | GRIP | Blunier et al. (1995); Chappellaz et al. (1997); Chappellaz et al. (1993) | +23.84[a] | GICC05 (Rasmussen et al., 2014; Seierstad et al., 2014) |
|  | NGRIP | Beck et al. (2018) | +6.15[a] | GICC05 (Rasmussen et al., 2014; Seierstad et al., 2014) |
|  | WAIS Divide | Mitchell et al. (2013) |  | WDC05A (Mitchell et al., 2013; Mitchell et al., 2011) |
|  | Law Dome | Rubino et al. (2019) |  | Rubino et al. (2019) |
|  | Dome C | Flückiger et al. (2002) | +10.38[a] | Beck et al. (2018) |
|  | EDML | Schilt et al. (2010) | +1.52[a] | Beck et al. (2018) |
|  | TALDAICE | Beck et al. (2018) | +6.15[a] | Beck et al. (2018) |
|  | TALDAICE | Bock et al. (2017) | +6.15[a] | Beck et al. (2018) |
|  | TALDAICE | Schilt et al. (2010) | +4.06[a] | Beck et al. (2018) |
|  | Siple Dome | Brook (2009) |  | Beck et al. (2018) |
| N2O | NEEM | This study |  | GICC05 (Rasmussen et al., 2013) |
|  | Dome Fuji | This study |  | WDC05A (Mitchell et al., 2013; Mitchell et al., 2011) |
|  | Dome Fuji | Kawamura (2001) |  | WDC05A (Mitchell et al., 2013; Mitchell et al., 2011) |
|  | EURO core | Flückiger et al. (1999) |  | Flückiger et al. (1999) |
|  | GRIP | Flückiger et al. (1999) |  | GICC05 (Rasmussen et al., 2014; Seierstad et al., 2014) |
|  | GISP2 | Sowers et al. (2003) |  | WDC05A (Mitchell et al., 2013; Mitchell et al., 2011) |
|  | Law Dome | Rubino et al. (2019) |  | Rubino et al. (2019) |
|  | H15 | Matchida et al. (1995) |  | Machida et al. (1995) |
|  | EDC | Schilt et al. (2010) |  | Beck et al. (2018) |
|  | TALDICE | Schilt et al. (2010) |  | Beck et al. (2018) |
|  | EDML | Schilt et al. (2010) |  | Beck et al. (2018) |
|  | NEEM | Prokopiou et al. (2018) |  | GICC05 (Rasmussen et al., 2013) |
| CO2 | NEEM | This study |  | GICC05 (Rasmussen et al., 2013) |
|  | Dome Fuji | This study |  | WDC05A (Mitchell et al., 2013; Mitchell et al., 2011) |
|  | Dome Fuji | Kawamura et al. (2007) |  | WDC05A (Mitchell et al., 2013; Mitchell et al., 2011) |
|  | Law Dome | Rubino et al. (2019) |  | Rubino et al. (2019) |
|  | WDC | Ahn et al. (2012) |  | Ahn et al. (2012) |
|  | EDML | Siegenthaler et al. (2005) |  | Siegenthaler et al. (2005) |
|  | South Pole | Siegenthaler et al. (2005) |  | Siegenthaler et al. (2005) |

[a] Offset correction was made to $CH_4$ concentrations by following Beck et al. (2018).



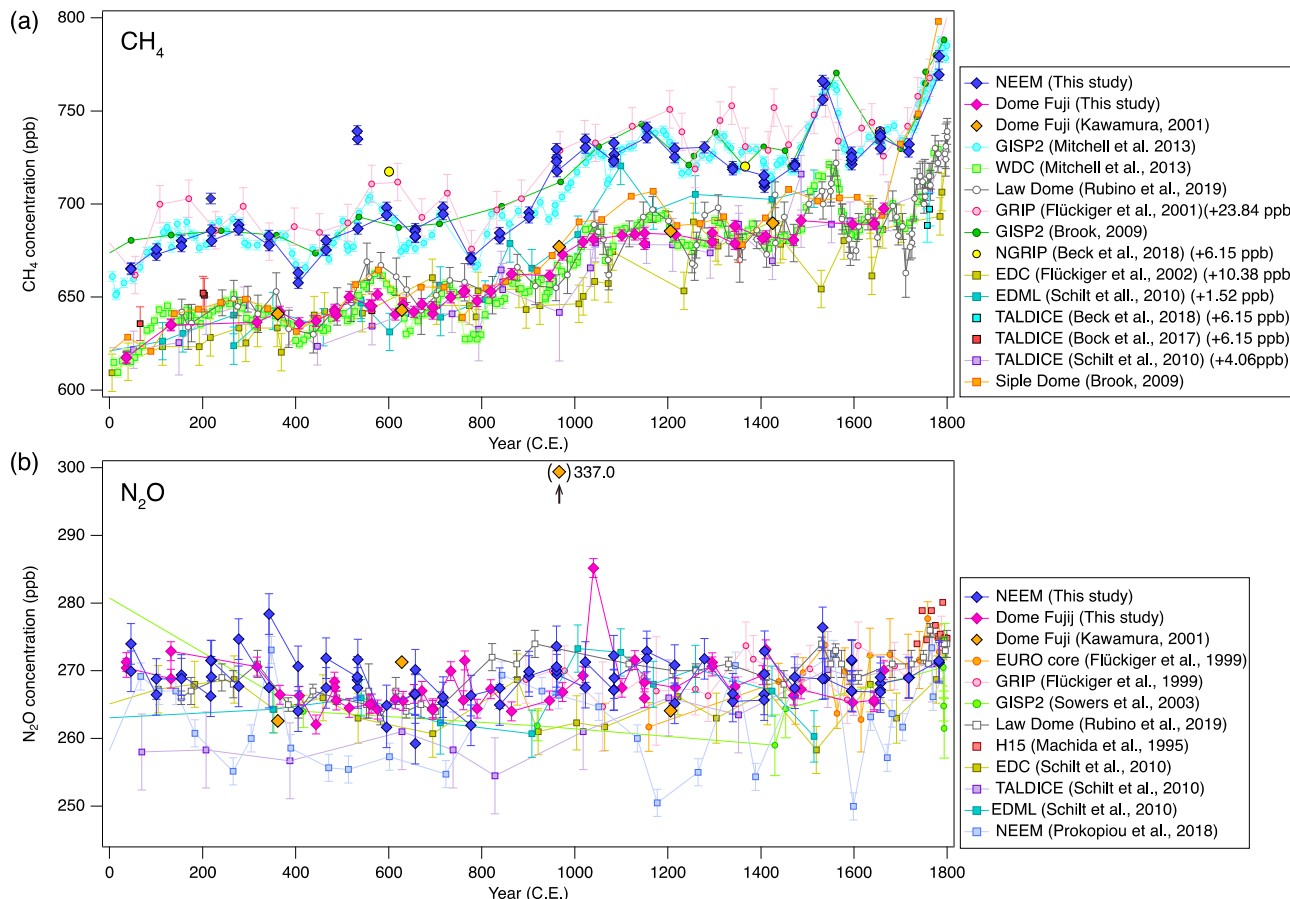

**Figure A1:** (a) CH4 and (b) N2O concentrations for 0 – 1800 C.E. from the DF and NEEM ice cores measured with our new method, and the comparison with published records from other groups (Beck et al., 2018; Bock et al., 2017; Brook, 2009; Flückiger et al., 1999; Flückiger et al., 2002; Kawamura, 2001; Machida et al., 1995; Mitchell et al., 2013; Prokopiou et al., 2018; Rubino et al., 2019; Schilt et al., 2010; Sowers et al., 2003). The DF data is placed on the WDC05A chronology (see Fig. 9), and all the other data are placed on the respective (published) time scales (details are summarized in Table A1). The GRIP, NGRIP, EDC, EDML, and TALDICE data are corrected for systematic offsets relative to the WDC data, as reported by Beck et al. (2018).