# Peer review of "New technique for high-precision, simultaneous measurements of CH4, N2O and CO2 concentrations, isotopic and elemental ratios of N2, O2 and Ar, and total air content in ice cores by wet extraction"

_Atmospheric Measurement Techniques, 2020_

## Referee Comment (RC1) · Anonymous Referee #1 · 23 Jul 2020

Review of "New technique for high-precision, simultaneous measurements of  $CH_4$ ,  $N_2O$  and  $CO_2$  concentrations, isotopic and elemental ratios of  $N_2$ ,  $O_2$  and Ar, and total air content in ice cores by wet extraction" by Oyabu, I. et al. AMTDiss..

**General:**

The manuscript discusses a significantly improved extraction method for sample size without compromising precisions of several important paleo-proxy parameter. The multi-proxy approach is very helpful for improving not only the resolution due to the lower sample amounts necessary, but also regarding the comparison among the different parameters analysed. This again improves the issues with timing, since all the parameters are measured on the same sample, as well as the intercomparison of parameter because only one laboratory and one method is used.

The manuscript is very nicely written with detailed information how the method works and how it is used for standard and sample analyses. Furthermore, the authors show tests that only a very limited number of corrections are necessary which is obviously due to the indepth selection and preconditioning of all materials used in the extraction, split and measurement lines. They further state how the values are calibrated.

It was easy to read the manuscript and I would like to congratulate the authors. I have only a few rather minor comments and suggestions. I suggest to publish it once these comments have been taken into consideration.

Minor points:

You often used subscripts rather than superscripts in the text. This need to be changed, i.e.  $\delta_{15}N$  rather than  $\delta^{15}N$ , or cm3 rather than cm3. Please check any such issues.

Line 137: New header (Description of method and manipulations)

Line 211: Flame Ionization Detector not Frame

Line 293: How are the coefficients d, e and f calculated, how do they relate to a, b and c? Eq. 6, lines 380ff: what about the sample loss during the first evacuation after loading the sample? Is this neglectable?

Eq. 8: Why is the normalization made to direct atmospheric air and not to a standard that is well linked to the outside air at a given time.

Line 548: ... is impossible to be of atmospheric origin...

Line 603ff: you might cite Huber et al., EPSL 2006

---

## Author Comment (AC1) · 8 Aug 2020

*Review of "New technique for high-precision, simultaneous measurements of $CH_4$, $N_2O$ and $CO_2$ concentrations, isotopic and elemental ratios of $N_2$, $O_2$ and Ar, and total air content in ice cores by wet extraction" by Oyabu, I. et al. AMTD.*

*General:*

*The manuscript discusses a significantly improved extraction method for sample size without compromising precisions of several important paleo-proxy parameter. The multi-proxy approach is very helpful for improving not only the resolution due to the lower sample amounts necessary, but also regarding the comparison among the different parameters analysed. This again improves the issues with timing, since all the parameters are measured on the same sample, as well as the intercomparison of parameter because only one laboratory and one method is used.*

*The manuscript is very nicely written with detailed information how the method works and how it is used for standard and sample analyses. Furthermore, the authors show tests that only a very limited number of corrections are necessary which is obviously due to the in depth selection and preconditioning of all materials used in the extraction, split and measurement lines. They further state how the values are calibrated.*

*It was easy to read the manuscript and I would like to congratulate the authors. I have only a few rather minor comments and suggestions. I suggest to publish it once these comments have been taken into consideration.*

Thank you very much for your review. Our replies are in Blue.

*Minor points:*

*You often used subscripts rather than superscripts in the text. This need to be changed, i.e. $\delta^{15}N$ rather than $\delta_{15}N$, or cm3 rather than cm3. Please check any such issues.*

They were due to unexpected errors of MS Word application when the word file was converted to the PDF file. We will use a different conversion software to solve the issue and check all the descriptions.

*Line 137: New header (Description of method and manipulations)*

We will add a new header there, and we will also add two more headers under section 2.1., as the following.

**2.1.1 Extraction line and its pre-treatment**

Line 93 (current manuscript) "A schematic diagram of our extraction system is…….".

**2.1.2 Preparation of apparatus and ice samples**

Line 110 (current manuscript) "For routine air extraction, the sample tubes and extraction line are……….".

**2.1.3 Manipulations for air extraction**

Line 137 (current manuscript) "Up to six vessels, thus prepared, are brought to our laboratory room at……..".

*Line 211: Flame Ionization Detector not Frame*

We will correct it.

*Line 293: How are the coefficients d, e, and f calculated, how do they relate to a, b and c?*

For the FIDs, the relationships between peak area and pressure (for a same standard gas) are found to be almost linear over a wide pressure range, but it is slightly nonlinear towards lowest pressures within the range of the ice core measurements. Also, the response of ECD detector is generally nonlinear. We thus interpolate the three calibration points for each of three standard gas by a quadratic fit (the determination of a, b, and c for each gas). The relationship between the concentration and area at any given pressure is also slightly nonlinear, and thus we interpolate the three calibration points on the area-concentration space (from three standard gases, at the pressure of the sample gas) also with a quadratic fit (the determination of d, e, and f). A three-point calibration with quadratic fit is a common practice in precise atmospheric observations. The a, b, and c should be closely related with d, e, and f (after compensating for different units, for the same molar abundance of a gas of interest) because they both represent the same nonlinear responses of the same detectors. However, the coefficients would not necessarily be identical because of uncertainties unrelated with the detector characteristics, such as those from pressure measurement, standard gas scale, and deviation of actual pressure-area relationship from the quadratic function.

We will add some more description to the text based on the above explanation, together with a figure below.

[Figure]

**Figure:** Example of calibration procedure for $CH_4$ concentration. (a) Black lines are quadratic fits to the peak area versus pressure for the three standard gases. Peak areas of the standard gases at the sample pressure are estimated by substituting the sample pressure on the fitted curves (red points, $A_{St,1,P}$, $A_{St,2,P}$, and $A_{St,3,P}$). (b) Red line is the calibration curve for the sample, which is a quadratic fit through the three red points from (a).

*Eq. 6, lines 380ff: what about the sample loss during the first evacuation after loading the*

*sample? Is this neglectable?*

We indeed take into account the sample loss during the evacuation as follows, although it is small. After all the extractions for a day, the mass of Trap 1 is measured to obtain the total mass of sublimated ice during the first-stage evacuations. The typical amount of ice is 0.5 – 1.5 g for four to six samples (each with ~90-min evacuation). To account for the additional loss during the second evacuation after switching the line to the transfer line (~30 min), the mass of ice in Trap 1 is multiplied by 4/3 (120 min/90 min). The ice loss for each sample is estimated by simply dividing the total ice loss by the number of samples, and it is typically 0.1 – 0.3 g. We will add this explanation to the text.

*Eq. 8: Why is the normalization made to direct atmospheric air and not to a standard that is*

*well linked to the outside air at a given time.*

The normalization of the sample is indeed made to the reference can, which is determined against atmosphere from time to time. Perhaps the current equation gave the wrong impression that the sample is directly normalized against atmosphere. We will re-write the equation in an expanded form, so that the readers can immediately see (without looking for the previous equations) that all the measurements are made against the reference can.

*Line 548: ...is impossible to be of atmospheric origin…*

We will correct it.

*Line 603ff: you might cite Huber et al., EPSL 2006*

We will add Huber et al. (2006, EPSL) and Severinghaus et al. (2009, Science).

---

## Referee Comment (RC2) · Jochen Schmitt (Referee) · 12 Sep 2020

General remarks:

The paper by Ikumi Oyabu and co-authors presents a new technique to extract and subsequently measure a large suite of key parameters on a single ice core sample. Since so many and diverse parameters are obtained from this new device, this paper is quite extensive. Not only the technique is described, but also new results for sev-

eral parameters are presented and compared with previous records. Thanks to the fact that the manuscript is both well-structured and clearly written, the reader is not overwhelmed and can find the necessary information without too much effort.

The introduction provides the reader with sufficient information of the key applications of the individual parameters that are obtained with the new device. This chapter nicely captures the general advantages of multi-species measurement systems, i.e. to save precious ice and, more importantly, to obtain a large suite of parameters from the same piece of ice. Obviously, this feature is advantageous to improve the palaeo-climatic interpretation (no age difference between parameters), it is also very useful for troubleshooting and scrutinizing the parameters. As the ice core community continuously improves the precision and accuracy of these parameters, more and more previously unknown effects (e.g., production of species in the ice or preferential loss of some gas species from the ice) emerge from the noise level that alter the archived information of the ice core. Multi-species methods that provide a wide array of parameters without compromising the individual precisions are the key techniques to help identify and quantify those effect. In that sense, the new method presented by Oyabu and co-authors is very welcome and sets a new standard for these analyses.

Overall, the paper is already in good shape. Below I commented only on minor issues to improve the accessibility of the text and figures.

Minor comments on the text:

Line 77: Beginning of Chapter 2. You immediately start describing your procedure and then you compare/discuss it afterwards with other options to extract air with wet extraction techniques. While OK, you may add a short introductory section to briefly discuss the different wet techniques and point out their advantages and drawbacks. This may help readers not familiar with wet extraction techniques to put your technique into a wider perspective; e.g. . . .three different types of wet extraction techniques have been developed that. . . 1) melt-refreeze techniques (you already mention it with their

drawbacks) 2) melting under vacuum with the vessel closed, with subsequent helium purging (takes long and consumes lots of helium often used for isotope work on greenhouse gases e.g. Bock et al. 2014, AMT); 3) melting under vacuum with immediate removal of the released air (Kawamura et al., Schmitt et al. 2014, AMT, Bereiter et al. 2018). Method 3) is the preferred way to achieve sufficiently high extraction efficiencies for precise values of soluble trace gases (e.g. $N_2O$, Xe) and high-precision ratios of atmospheric main components (e.g. $O_2/N_2$).

Line 83: please clarify: "Also, the pressure over the meltwater is relatively low ($\sim$100 – 200 Pa),..." The reported 100 to 200 Pa is not the air pressure over the meltwater (within the vessel) as this value is measured far away downstream at P1, after the water traps, if I got it right. The air pressure after the first water trap is modulated by the individual flow resistances of the tubing etc. between the vessel, the water trap, and the He cryostat. To my knowledge, there is no simple solution to come up with a robust estimate of the air pressure over the meltwater. One option would be to install a pressure sensor close to the vessel and before the first water trap and measure the total pressure ($pH_2O$ at 0°C around 600 Pa plus the air pressure) and subtract the water vapour pressure. Using bubble-free ice as a comparison allows to get an estimate of the actual $pH_2O$ while the ice melts, which might be warmer than 0°C since you melt the ice quite quickly with the hot water bath. Line 108: you might add here that this $H_2O + O_2$ pre-treatment is sufficient for the (relatively short) extraction step. However, $O_2/N_2$ ratios are not stable in these sample tubes over a longer time, therefore we use a different set of dip tubes after splitting (GEPW tubes).

Line 111: you might remove this extra info "(i.e., on weekends and holidays)" to help keep the paper short.

Line 122: remove "highly". unclear: "the surface of ice samples". Actually, the fractionation happens on the exposed outer parts of the stored ice core sections (for me the term "ice sample" refers to the piece of ice that is prepared prior to the measurement in the lab).

Line 123: "... removed to precisely measure" add: ... and accurately the archived composition for these ratios.

Line 127: "... not important for greenhouse gas ...large molecular size...". This sentence could be better phrased to prevent misunderstanding. For example, $CO_2$ concentrations could indeed be biased by gas loss if $O_2$ (or if $O_2$, $N_2$, Ar, are lost without relative fractionation) is preferentially lost from the ice, while $CO_2$ is preserved due to its larger diameter. Consequently, $CO_2$ conc. would be biased higher as $CO_2$ conc. is just the ratio of $CO_2$ against total amount of (remaining) air. This process might occur but it is not yet visible due to limitations in the measurement precision. Gas loss biases for $CO_2$ were discussed as a potential option to explain the observed offsets among different ice cores of the same age (e.g. Eggleston et al. 2016 doi:10.1002/2015PA002874).

Line 133: remove the word vessel after Dewar.

Line 137: "laboratory room", remove the word room after laboratory;

Line 144: regarding the impact of the ice + gas loss during sublimation on TAC: can you estimate how much ice is lost due to sublimation (e.g. weighing the sample before and after 90 +30 min of pumping?

Line 149: "After the evacuation, all vessels, except for one for the first extraction, are..." please rewrite

Line 156: "The hot water bath is removed after the completion of ice melting." At that point, I wondered how the operator detects when the ice is gone because the vessel where the ice is melted is full metal, thus there is no visual control of that point. I guess it is done via a sudden drop in the pressure reading at P1? Also, all valves of the entire system are manually operated, right?

Line 159: "The time required for melting the ice sample is $<\sim 3$ minutes, and the total time for the air transfer takes $\sim 10$ min (including the ice melting)." Perhaps it is easier for the reader if you write that the ice melting takes ca. 3 min and the pumping step

after melting is completed takes additional ca. 7 min (i.e. the duration of the individual steps). Also: convert the mTorr unit into Pa to use only one pressure unit throughout the paper.

Line 163-165: this sentence could benefit from rephrasing. I am not sure if I got it.

Line 170: "one night" perhaps just write the hours that are physically needed for this equilibration step and not "one night"

Line 173: Does that mean you measure two sets of 6 samples each day?

Line 182: "...depletion of the O2/N2 ratio" just to be on the save side, explicitly say that O2 is consumed at the metal surfaces leading to lower O2/N2 values (to exclude the option that extra N2 causes lower O2/N2 ratios)

Line 184: perhaps add a sentence stating that the stainless steel sample tubes lead to an unacceptable trend in the O2/N2 ratio as O2 is consumed (perhaps provide numbers if available so the reader gets a handle on the size of this O2 depletion).

Line 187: delete "(e.g., during nights and weekends)",

Line 191: replace "(measured at the head of a turbomolecular pump with an ionization gauge) with (P3)

Line 198: perhaps add the reason for the -196°C cold trap: ..to remove CO2 and N2O...

Line 201: is the lowering of the tube by a few cm an essential step or rather precautionary or does it speed up the process?

Line 204: unclear "Sometimes the samples are measured on the following day of the splitting".

Line 209: "the air left in the original sample tube" can you roughly provide the reader with an estimate of how much the rest is (e.g. fraction of the original amount)? You

provide this info (80% vs 20%) only in the conclusion.

Line 211: FID = flame ionization detector

Line 223-: This GC section is nicely written too, yet due to the inherent complexity of the system it still took me a while to jump from one valve to the next and figure out which way the gas goes. It is hard to come up with a better description but you could help the reader by optimizing details, e.g. you write "we use small sample loops (1/16" o.d., 0.5 cm3, Valco)" here, it would help that you use the same unit as in Fig. 3 (rather than 0.5 cm3 in the text and 500 uL in the figure)...write e.g. we use two 500 uL sample loops (Loop 1 and 2 in Fig. 3 [mentioning the names Loop 1 and 2 helps the reader locate the object quicker])

Line 229: 400 – 600 hPa...

Line 241: let? "At 1.74 min, V2 is switched to lead $CO_2$ from column 1 to pass through..."

Line 346 : "... but the first cycle is discarded and only the latter 16 values are used." Delete "and only the latter 16 values are used" as this should be clear already.

Line 351: "...by simply opening the tube valve". Just curiosity: is the valve opened quickly?

Line 359: (at $\Delta P = 0$, +100 and -100 ‰. In line 347 you write "... imbalancing the sample pressure by $\pm$ 10 %". It is already clear that the $\pm$ 10 % refer to the +100 and -100 ‰ but using the same "unit", either % or ‰ could save a few seconds for some readers.

Line 437: "The standard gas thus transferred to the sample tubes is (are) measured ..." check is/are.

Line 465: as you assume that a constant amount of $O_2$ is consumed independent of the sample size you could use a Keeling plot style fit (i.e. 1/V) and work with a linear

system instead of the exp function.

Line 501: please rephrase this sentence

Line 511: "...112.88 – 157.81 m (bubbly ice, 2.0 – 0.2 kyr BP)," please arrange order, the same a line below for the NEEM core

Line 551: "In any case, we exclude these anomalously high concentrations from the calculation of pooled standard deviation and comparison with other ice core records". Perhaps rephrase: I am not so convinced by your statement that these anomalies should not be compared with other records as CH4 anomalies have been observed for NEEM and GISP2, therefore these anomalies are valuable and important observations. You might also cite and have a look in Mitchel et al. 2013, Supplement figure S1: there are several instances where GISP2 shows elevated CH4 that are not an analytical artefact but features of the ice (see Rhodes et al. papers 2013, 2016). Just my curiosity: 2 out of three of anomalous samples from 361 m show also elevated CO2 values. Do the corresponding dO2/N2 values for these samples also deviate from their neighbors (e.g. lighter values due to microbial O2 consumption)?

Line 561: The comparison of your N2O measurements with previous measurements can be improved (especially in Fig. A1) as many records exist and the evolution during the Late Holocene is rather flat with only little fine structure. To make this comparison easier for the reader, you might use the Late Holocene Monte Carlo N2O spline with 2s error envelopes and compare your DF and NEEM records with that spline as in Figure 4 of Fischer et al. 2019, Biogeosciences, doi: 10.5194/bg-16-3997-2019. https://www.ncdc.noaa.gov/paleo-search/study/28890, the direct link to the Late Holocene N2O spline is: https://www1.ncdc.noaa.gov/pub/data/paleo/icecore/fischer2019/fischer2019spline3k-2o.txt

Line 578: to be complete, you might add "... generally measured with mechanic dry extraction techniques (references) and sublimation (Schmitt et al. 2011, doi:10.5194/amt-

4-1445-2011).

Line 585 and the following line: It might be better to mention in an introductory sentence that reliable atm. CO2 reconstructions are only possible in Antarctic ice cores because impurity levels in Greenland are too high and lead to excess CO2 (refs). . . .so it is no wonder that your NEEM data also shows higher values.

Line 595: ". . .outer ice pieces were. . ."

Line 599-601: these two sentences have almost the same meaning, combine.

Line 601: ". . .indicating significant size-dependent fractionation in the outer ice". Without further information, the reader has no clue why you immediately state that these observations are due to a size-fractionating process. May be O2 has a higher permeation rate due to solubility in the ice matrix compared to N2. Either provide more background or just write e.g. gas species-dependent gas loss

Line 603: ". . .but not for N2, in the outer ice." Unclear: because N2 is not lost at all.

Line 609: please better specify what is plotted in Fig. 14: While it is clear that Fig. 13 shows the differences between samples with 5 or 8 mm removed this is not so clear for Fig. 14.

Line 612: ". . .more than 8 mm is sufficient" message not clear: is 8 mm enough?

Line 624: "$\delta$Î§2/ÎÎ2 as originally trapped. . .can be reconstructed (delete measured) from the 20-year old samples. Also, the overall match of the corrected old data and the new data is a good sign that the relatively large gas-loss correction was accurately done at that time.

Line 625: regarding the d18O values in Fig. 15b it seems that the old values that experienced gas loss (thus mass-dependent fractionation of the O2) are consistently heavier and is thus in line with the idea of the mass-dependent fractionation during the O2 loss from the ice (a scaled correction factor that was used to correct the O2/N2

data from gas-loss effects might work as well). Comment regarding the d15N values: The new measurements of the clathrate ice look, on average, ca 0.03‰ higher than the old Kawamura values. This systematic offset is not visible for the young ice, thus calibration is not the reason. Yet for gas loss, the sign is in the wrong direction.

Line 648: technique?

comments on the figures and tables:

Fig. 1 typo in the scheme: "He ditector", should read detector. Note, the schematic is quite large and in the final version it might get squeezed a bit to fit the page. To allow readability of the labels please increase the font size a bit (in most cases the distances allow that).

Fig. 2: you might help the reader find the "small volume (1.4 cm3) between the valves" by adding colors to this section.

Fig. 3: the ":" symbol in the legend seems not necessary. electronforming (without n); explain SuS

Fig. 4: I guess the y-axes are arbitrary units, please add any label to that axes. CH4 panel, please add labels to the O2 and CH4 to make it more intuitive for the reader which peak is what. Since your paper has many figures (15) and tables you might want to save a bit space here and try plotting Fig. 4 more compactly, e.g. plotting all three gases on one x-axis without showing the others (like you did in Fig. 5)

Fig. 5: there seems to be a typo at the x-axis labels (16.7.1), also Fig. 7. To better differentiate the symbols (filled vs open markers), you might increase the symbol size. Panel c. For the O2/N2 ratio is seams that with each new filling of the reference can the trends get smaller, i.e. less O2 is consumed for a certain time interval. Is this an indication that consumption ceases over time as all of the oxygen-reactant is consumed?

Fig. 6: there is a typo in the equation: there is + and - before 0.078802, equation (7)

says -0.078802. you may add the equation into the figure rather than in the caption

Fig. 8: it looks like that the new $CO_2$ measurements are slightly higher between 100 and 150 m and after 1800 m compared to the old ones? perhaps calc. the difference of means for the 100-150 m. Is it due to better extraction efficiency (or lower headspace pressure air reducing $CO_2$ solution in the meltwater) or longer reaction time for the chemical $CO_2$ production pathway in the meltwater?

Fig. 9: Caption: remove "from other groups". Panel c: you don't need the connecting lines for South Pole and EDML and WDC. just keep it for Law Dome.

Fig. 10: Caption: write "The line connects the blue markers..." since there is only one line, therefore, there is no need to specify this line it as "dotted line (blue)" Actually, I am not sure if you need the blue line at all as the next neighbouring points to the right side are quite a bit older and thus outside of the gas age distribution. For me this situation with the artefact samples is convincing already as the age difference of the plotted data is smaller than the NEEM gas age distribution. Additionally, you might indicate these two artefact depth levels already in Fig. 9 panel a) e.g. with a red circle around the artefact samples that you zoom in in Fig. 10.

Fig. 11: caption. unclear statement: "For single (non-duplicate) measurements, "a" piece is not cut.". I am not sure if I got it right. If you do a single measurement, piece "b" is measured only, but you need to do the cut between a and b anyway? Further, how did you cut the curvature between the inner and outer pieces?

Fig. 12: you might consider producing a single figure (as e.g. nicely done in Fig. 15) out of the four panels because you don't need four-times Depth(m) and the depth scales. The same applies to Figs. 13 and 14. Also, you might consider using a Delta symbol (e.g. D d Ar/N2 to indicate the "pair differences" to prevent too long y-axis labels every time and write this in the caption (Pair differences (Delta) between the inner and outer ice piece.

Fig. 14: figure: go for the same design: the dashed line at zero is plotted as a thick line in d) while thin for a, b, c. please chose one thickness, e.g. the thin line style. Caption line 955: add pieces: between the two inner ice pieces: and also specify if it is a-b or b-a.

Fig. 15: figure top x-axis label: replace Air age with Gas age because you use gas age throughout your manuscript. Legend between panel c) and d) the names for the grey and red crosses are switched, the red symbols refer to the values After gas loss correction? caption line 1001: delete dotted lines, because there are none (crosses connected with dotted lines, Kawamura, 2000; Kawamura et al., 2007). Table A1: typo: TALDICE not TALDAICE; the Eurocore ice core and project is mostly written Eurocore (in some cases EUROCORE, but EURO core would be new)

Fig. A1: panel a) this panel is a bit busy. you might consider plotting more records without the connecting line (EDML, TALDICE Schilt, Siple Dome) typo. EDML (Schilt et all. 2010), et al. And for the legend, TALDICE add a blank between 4.06 and ppb.

panel b) this is a busy panel and you might consider plotting only some records (your data and perhaps NEEM from Prokopiou et al) with a connecting line while removing the line for others (Eurocore, GRIP, GISP2, EDC, TALDICE, EDML)

Finally, a few technical things: Most of your subscripts and superscripts in your manuscript have been moved up or down. Please check but the usual style convention for AMT papers is that the units % and ‰ are written without a blank, e.g. 0.23‰ while all others come with a blank. Also, the unit liter is written L, e.g., 25 mL.

Please also note the supplement to this comment:
https://amt.copernicus.org/preprints/amt-2020-171/amt-2020-171-RC2-supplement.pdf

---

## Author Comment (AC2) · 9 Oct 2020

*General remarks:*

*The paper by Ikumi Oyabu and co-authors presents a new technique to extract and subsequently measure a large suite of key parameters on a single ice core sample. Since so many and diverse parameters are obtained from this new device, this paper is quite extensive. Not only the technique is described, but also new results for several parameters are presented and compared with previous records. Thanks to the fact that the manuscript is both well-structured and clearly written, the reader is not overwhelmed and can find the necessary information without too much effort.*

*The introduction provides the reader with sufficient information of the key applications of the individual parameters that are obtained with the new device. This chapter nicely captures the general advantages of multi-species measurement systems, i.e. to save precious ice and, more importantly, to obtain a large suite of parameters from the same piece of ice. Obviously, this feature is advantageous to improve the palaeo-climatic interpretation (no age difference between parameters), it is also very useful for troubleshooting and scrutinizing the parameters. As the ice core community continuously improves the precision and accuracy of these parameters, more and more previously unknown effects (e.g., production of species in the ice or preferential loss of some gas species from the ice) emerge from the noise level that alter the archived information of the ice core. Multi-species methods that provide a wide array of parameters without compromising the individual precisions are the key techniques to help identify and quantify those effect. In that sense, the new method presented by Oyabu and co-authors is very welcome and sets a new standard for these analyses.*

*Overall, the paper is already in good shape and below I commented only on minor issues to improve the accessibility of the text and figures.*

Thank you very much for your very detailed and thorough review, including a number of specific suggestions on the manuscript. We will take them into consideration in revising the manuscript. Below, we reply to your comments in blue.

*Minor comments on the text:*

*Line 77: Beginning of Chapter 2. You immediately start describing your procedure and then you compare/discuss it afterwards with other options to extract air with wet extraction techniques. While OK, you may add a short introductory section to briefly discuss the different wet techniques and point out their advantages and drawbacks. This may help readers not familiar with wet extraction techniques to put your technique into a wider perspective; e.g. …three different types of wet extraction techniques have been developed that…1) melt-refreeze techniques (you already mention it with their drawbacks) 2) melting under vacuum with the vessel closed, with subsequent helium purging (takes long and consumes lots of helium often used for isotope work on greenhouse gases e.g. Bock et al. 2014, AMT); 3) melting under vacuum with immediate removal of the released air (Kawamura et al., Schmitt et al. 2014, AMT, Bereiter et al. 2018). Method 3) is the preferred way to achieve sufficiently high extraction efficiencies for precise values of soluble trace gases (e.g. $N_2O$, $Xe$) and high-precision ratios of atmospheric main components (e.g. $O_2/N_2$).*

We will modify the first paragraph of Chapter 2 as follows.

"Four types of wet extraction techniques have been developed by different laboratories: [1] ice is melted and slowly refrozen in a closed vessel to expel dissolved gas from the meltwater (so-called melt-refreeze technique, e.g., Brook et al., 2005; Chappellaz et al., 1997; Flückiger et al., 1999; Severinghaus et al., 2009; Sowers et al., 1989; Lipenkov et al., 1995 (air content)), [2] ice is melted in a closed vessel with subsequent agitation during transfer to extract dissolved gas (Severinghaus et al., 2003; Kobashi et al., 2008), [3] ice is melted in a closed vessel with subsequent helium purging to extract dissolved gas (e.g., Bock et al. 2014), and [4] ice is melted in a vessel open to a sample tube to transfer the extracted air immediately (e.g., Bereiter et al. 2018; Kawamura et al., 2003; Nakazawa et al., 1993a; Schmitt et al. 2014). Method [1] is most widely used with small ice samples (~10 to 50 g) for measuring basic gas components for paleoclimatic reconstructions such as $CH_4$ and $N_2O$ concentrations, or the isotopic and elemental ratios of $N_2$, $O_2$, and Ar with high precisions. The method requires a relatively long time for refreezing (up to several tens of minutes), and thus possibly elevates trace gas concentrations in the extracted air by degassing from the inner wall of the vessel, as well as alter the air composition by gas-dependent dissolution in meltwater and incomplete degassing during refreezing. Method [2] is used with larger ice samples (50 to 100 g) for $N_2$ and noble gases. Method [3] is used with much larger ice (several hundred grams) for measuring isotopic ratios of greenhouse gases. It takes a long time and consumes a large amount of helium. Method [4] is typically used with samples with intermediate or large size (one to several hundred grams) for measuring multiple gas species. It is also a preferred way to achieve both high extraction efficiencies for soluble trace gases (e.g., $N_2O$ and Xe) and high-precision ratios of $N_2$, $O_2$, and Ar."

*Line 83: please clarify: "Also, the pressure over the meltwater is relatively low (~100 – 200 Pa),…" The reported 100 to 200 Pa is not the air pressure over the meltwater (within the vessel) as this value is measured far away downstream at P1, after the water traps, if I got it right. The air pressure after the first water trap is modulated by the individual flow resistances of the tubing etc. between the vessel, the water trap, and the He cryostat. To my knowledge, there is no simple solution to come up with a robust estimate of the air pressure over the meltwater. One option would be to install a pressure sensor close to the vessel and before the first water trap and measure the total pressure (pH2O at 0°C around 600 Pa plus the air pressure) and subtract the water vapour pressure. Using bubble-free ice as a comparison allows to get an estimate of the actual pH2O while the ice melts, which might be warmer than 0°C since you melt the ice quite quickly with the hot water bath.*

As you point out, the pressure (~100 – 200 Pa) is the reading of P1 and thus probably much lower than the air pressure over the meltwater in the vessel. We will simply remove "(~100 – 200 Pa)" from the text. It would be interesting and informative if we can actually measure the air pressure in the vessel, but we have not done such experiments considering the fact that it would not help to improve the method itself, and that it requires significant cost (a dedicated pressure sensor, some modification to the system, and time and effort to establish the experimental procedures and precision). Together with the prior sentence, we modified the text as the following.

"It requires a relatively short time (< 10 minutes) for melting ice and transferring extracted air, minimizing contaminations due to degassing from the inner walls of the apparatus as well as dissolution of gases in the meltwater. The much lower air pressure over the meltwater than that in Method [1] also helps to lower the gas dissolution in the meltwater (Kawamura et al., 2003)."

*Line 108: you might add here that this $H_2O + O_2$ pre-treatment is sufficient for the (relatively short) extraction step. However, $O_2/N_2$ ratios are not stable in these sample tubes over a longer time, therefore we use a different set of dip tubes after splitting (GEPW tubes).*

We will revise the text as follows.

We note that we use a different set of sample tubes with a better performing treatment after the splitting step for $\delta O_2/N_2$ stability (GEPW tubes, see below), but the $H_2O + O_2$ pre-treatment is sufficient for the extraction step and has the advantage of low cost.

*Line 111: you might remove this extra info "(i.e., on weekends and holidays)" to help keep the paper short.*

We will delete it.

*Line 122: remove "highly". unclear: "the surface of ice samples". Actually, the fractionation happens on the exposed outer parts of the stored ice core sections (for me the term "ice sample" refers to the piece of ice that is prepared prior to the measurement in the lab).*

This part was also commented on by Reviewer1 in a different context. We will move most of the paragraph to the beginning of section 5.2 (discussion of gas-loss fractionation) according to Reviewer1, and modified according to your suggestions (this and the next two suggestions were taken into account).

The text now reads (beginning of Section 5.2):

"Previous studies have indicated that gases can slightly be lost from ice cores during storage, causing size- and mass-dependent fractionations in $\delta O_2/N_2$, $\delta^{18}O$ and $\delta Ar/N_2$ (Bender et al., 1995; Bereiter et al., 2009; Huber et al., 2006a; Ikeda-Fukazawa et al., 2005; Kawamura et al., 2007; Severinghaus et al., 2009). Greenhouse-gas concentrations could also be biased (presumably to higher values, Ikeda-Fukazawa et al., 2004, 2005; Bereiter et al., 2009; Eggleston et al. 2016), but it would not be detected in most cases with the current measurement precisions. As the $\delta O_2/N_2$, $\delta^{18}O$ and $\delta Ar/N_2$ ratios become fractionated especially in the exposed outer part of the core, the surface must sufficiently be removed to precisely measure the air composition and accurately reconstruct the ratios as originally archived in the ice sheet (Bereiter et al., 2009; Ikeda-Fukazawa et al., 2005; Kawamura et al., 2007; Severinghaus et al., 2009). The thickness of sufficient surface removal should depend on the storage period, storage temperature and the form of air in ice (bubbles or clathrate-hydrates). To examine whether those ratios as originally recorded in the ice sheet can be found in the long-stored DF core (at -50 °C for ~20 years), we measured samples from the same depths with different thickness of surface removal (for example, 8 and 5 mm) (Fig. 11). The outer ice pieces were also measured, and the results are compared with those from the inner ice."

*Line 123: "... removed to precisely measure" add: ... and accurately the archived composition for these ratios.*

See above.

*Line 127: "... not important for greenhouse gas ...large molecular size...". This sentence could be better phrased to prevent misunderstanding. For example, $CO_2$ concentrations could indeed be biased by gas loss if $O_2$ (or if $O_2$, $N_2$, Ar, are lost without relative fractionation) is preferentially lost from the ice, while $CO_2$ is preserved due to its larger*

*diameter. Consequently, $CO_2$ conc. would be biased higher as $CO_2$ conc. is just the ratio of $CO_2$ against total amount of (remaining) air. This process might occur but it is not yet visible due to limitations in the measurement precision. Gas loss biases for $CO_2$ were discussed as a potential option to explain the observed offsets among different ice cores of the same age (e.g. Eggleston et al. 2016 doi:10.1002/2015PA002874).*

See above.

*Line 133: remove the word vessel after Dewar.*
We will remove "vessel".

*Line 137: "laboratory room", remove the word room after laboratory;*
We will remove "room".

*Line 144: regarding the impact of the ice + gas loss during sublimation on TAC: can you estimate how much ice is lost due to sublimation (e.g. weighing the sample before and after 90 +30 min of pumping?*
We take into account the sample loss during the evacuation as follows, although it is small. After completing all extractions for a day, the mass of Trap 1 is measured to obtain the total mass of sublimated ice during the first-stage evacuations (all samples combined). The typical amount is 0.5 – 1.5 g for four to six samples (each with ~90-min evacuation). To account for the additional loss during the second evacuation after switching the line to the transfer line (~30 min), the mass of ice in Trap 1 is multiplied by 4/3 (120 min/90 min). Then, the ice loss for each sample is estimated by simply dividing the total ice loss by the number of samples, and it is typically 0.1 – 0.3 g. We will add this explanation to the text (section 3.3).

*Line 149: "After the evacuation, all vessels, except for one for the first extraction, are…" please rewrite*
We will rewrite as:
  "After the evacuation, all but one of the vessels are isolated by closing the valves above them (V2 – V7)."

*Line 156: "The hot water bath is removed after the completion of ice melting." At that point, I wondered how the operator detects when the ice is gone because the vessel where the ice is melted is full metal, thus there is no visual control of that point. I guess it is done via a sudden drop in the pressure reading at P1? Also, all valves of the entire system are manually operated, right?*
We found that the pressure at P1 is not a good guide for knowing the timing of the completion of ice melting, so we rely on the senses of the operators on the noise from the vessel and temperature at the bottom of the vessel. Popping noise can be heard during the ice melting (especially if it is clathrate ice), and it changes to higher frequency noise, which could originate in the boiling of meltwater under vacuum. However, the difference in noise is not always clear, thus we also sense the temperature at the bottom of the vessel by temporarily removing the hot water bath and touch the bottom for a few seconds. The rapidity of temperature decrease sensed by hand can tell if the ice is melted away (if ice still exists, we quickly feel icy coldness). We will add a brief description to the manuscript.
All valves of the extraction and split lines are manually operated.

*Line 159: "The time required for melting the ice sample is <~3 minutes, and the total time for the air transfer takes ~10 min (including the ice melting)." Perhaps it is easier for the reader if you write that the ice melting takes ca. 3 min and the pumping step after melting is completed takes additional ca. 7 min (i.e. the duration of the individual steps).*

We will change the text to:

"The melting of the ice sample takes <~3 minutes, and the remaining air transfer takes ~7 min after the melting."

*Also: convert the mTorr unit into Pa to use only one pressure unit throughout the paper.*

We will unify the units.

*Line 163-165: this sentence could benefit from rephrasing. I am not sure if I got it.*

We change the text to the following.

"The pressure in the next vessel is measured with the Baratron Gauge (P1) by closing V20, V21 and V22 and opening the valve of the vessel (typically <1 Pa for the second vessel and ~4 Pa for the fifth vessel, because of gradual accumulation of air released from ice samples). This ensures the absence of a leak through the valve above the vessel and hence the quality of the sample air in the previous extractions."

*Line 170: "one night" perhaps just write the hours that are physically needed for this equilibration step and not "one night"*

We will change it to "15 – 24 hours".

*Line 173: Does that mean you measure two sets of 6 samples each day?*

No, we can operate only one set of 6 samples in a day. We will change the text to:

"Finally, as the preparation for the next extractions (on the following day or after), another set of sample tubes and traps are connected to the line and evacuated with the turbomolecular pump (to $<10^{-2}$ Pa), and then the line is checked with a helium leak detector ($< 10^{-8}$ Pa L s$^{-1}$). After the leak check, the tubes and line are evacuated until the next extractions."

*Line 182: "...depletion of the $O_2/N_2$ ratio" just to be on the save side, explicitly say that $O_2$ is consumed at the metal surfaces leading to lower $O_2/N_2$ values (to exclude the option that extra $N_2$ causes lower $O_2/N_2$ ratios)*

We will add it as follows:

"To minimize the consumption of $O_2$ at the metal surfaces leading to depletion of the $\delta O_2/N_2$ ratio during the sample storage in the tube,......"

*Line 184: perhaps add a sentence stating that the stainless steel sample tubes lead to an unacceptable trend in the $O_2/N_2$ ratio as $O_2$ is consumed (perhaps provide numbers if available so the reader gets a handle on the size of this $O_2$ depletion).*

We will add the following sentence.

"In our experiences, stainless-steel sample tubes with mechanical polishing or electropolishing may lead to an unacceptable depletion of $\delta O_2/N_2$ (e.g., by -5 ‰) in less than 8 hours because of the $O_2$ consumption. The GEPW tubes do not deplete $\delta O_2/N_2$, and thus the storage correction is not necessary (see 4.2.1)."

*Line 187: delete "(e.g., during nights and weekends)",*
We will delete it.

*Line 191: replace "(measured at the head of a turbomolecular pump with an ionization gauge) with (P3)*
We will correct it.

*Line 198: perhaps add the reason for the -196°C cold trap: ..to remove CO2 and N2O...*
We will add it.

*Line 201: is the lowering of the tube by a few cm an essential step or rather precautionary or does it speed up the process?*
This step is a precautionary procedure, at least for our routine analysis with small samples (~1 mL$_{STP}$).

*Line 204: unclear "Sometimes the samples are measured on the following day of the splitting".*
We will delete the sentence and modify the prior sentence as:
"The GEPW tube is warmed to room temperature and allowed to homogenize the sample air for at least 3 hours before the MS analysis (the longest waiting time is ~20 hours)."

*Line 209: "the air left in the original sample tube" can you roughly provide the reader with an estimate of how much the rest is (e.g. fraction of the original amount)? You provide this info (80% vs 20%) only in the conclusion.*
We will add that the amount left is ~80% of the extracted air.

*Line 211: FID = flame ionization detector*
We will correct it.

*Line 223-: This GC section is nicely written too, yet due to the inherent complexity of the system it still took me a while to jump from one valve to the next and figure out which way the gas goes. It is hard to come up with a better description but you could help the reader by optimizing details, e.g. you write "we use small sample loops (1/16" o.d., 0.5 cm3, Valco)" here, it would help that you use the same unit as in Fig. 3 (rather than 0.5 cm³ in the text and 500 µL in the figure)...write e.g. we use two 500 µL sample loops (Loop 1 and 2 in Fig. 3 [mentioning the names Loop 1 and 2 helps the reader locate the object quicker] )*
We will use the same units (mL) and modified the explanation according to the suggestion.

*Line 229: 400 – 600 hPa…*

We will add "hPa".

*Line 241: let? "At 1.74 min, V2 is switched to lead CO2 from column 1 to pass through…"*
We will correct it.

*Line 346 : "… but the first cycle is discarded and only the latter 16 values are used." Delete "and only the latter 16 values are used" as this should be clear already.*
We will delete it.

*Line 351: "…by simply opening the tube valve". Just curiosity: is the valve opened quickly?*
We open the valve as quickly as possible.

*Line 359: (at ΔP = 0, +100 and -100 ‰). In line 347 you write "… imbalancing the sample pressure by ± 10 %". It is already clear that the ± 10 % refer to the +100 and -100 ‰, but using the same "unit", either % or ‰, could save a few seconds for some readers.*
We will use "10%".

*Line 437: "The standard gas thus transferred to the sample tubes is (are) measured …" check is/are.*
We will keep it to "are" as there may be more than one sample tube (filled with the same standard gas from a cylinder).

*Line 465: as you assume that a constant amount of $O_2$ is consumed independent of the sample size you could use a Keeling plot style fit (i.e. 1/V) and work with a linear system instead of the exp function.*
We tried the Keeling plot and the linear fit, but the goodness of fit turned out to be poor (figure below), and the magnitude of correction seemed to change too rapidly at low volumes. It might imply that $O_2$ consumption may not strictly be independent of sample size, and we would like to keep the exponential function that gives the better fit.

[Figure]

*Line 501: please rephrase this sentence*
We will modify this part to:

"In each month, the atmosphere is sampled in two flasks on the same day, and a total of 10 or more aliquots are measured and averaged. The STD-A is measured every week, and the values in the same month are averaged. The monthly atmospheric or STD-A values thus assigned against the reference can are averaged over two consecutive months, and the ice-core samples are normalized against the nearest two-month-average values (for effectively correcting for the drifts in the reference can)".

*Line 511: "…112.88 – 157.81 m (bubbly ice, 2.0 – 0.2 kyr BP)," please arrange order, the same a line below for the NEEM core*
We will correct them.

*Line 551: "In any case, we exclude these anomalously high concentrations from the calculation of pooled standard deviation and comparison with other ice core records". Perhaps rephrase: I am not so convinced by your statement that these anomalies should not be compared with other records as CH4 anomalies have been observed for NEEM and GISP2, therefore these anomalies are valuable and important observations. You might also cite and have a look in Mitchel et al. 2013, Supplement figure S1: there are several instances where GISP2 shows elevated CH4 that are not an analytical artefact but features of the ice (see Rhodes et al. papers 2013, 2016).*
We agree with the reviewer and change this part to the following.
"Anomalously high $CH_4$ concentrations with similar magnitudes have been reported from GISP2 and NEEM cores measured at other laboratories, and the reasons may be $CH_4$ production after bubble close-off (in the ice sheet) or during air extraction from dusty glacial-period ice (Mitchel et al., 2013; Rhodes et al., 2013, 2016; Lee et al., 2020). For the purpose of evaluating our system, we exclude the anomalous values from the calculation of pooled standard deviations and quantitative comparison with other records because they are extremely inhomogeneous. We speculate that the $CH_4$ anomalies in our data originate in in-situ production because we used the Holocene (low dust) ice, and full investigation of our system on $CH_4$ production (as found by Lee et al., 2020) would require higher-resolution analyses of dusty ice and inter-comparison with other laboratories, which is beyond the scope of this paper."

*Just my curiosity: 2 out of three of anomalous samples from 361 m show also elevated CO2 values. Do the corresponding dO2/N2 values for these samples also deviate from their neighbours (e.g. lighter values due to microbial O2 consumption)?*
We checked $CH_4$, $CO_2$ and $\delta O_2/N_2$ around 361 m and 418 m, where $CH_4$ concentrations are anomalously high. The $\delta O_2/N_2$ are indeed ~3 ‰ lower in the samples with higher $CH_4$. However, $\delta O_2/N_2$ neighboring samples in the vertical direction can vary at a similar degree (for example at ~190 m where $CH_4$ values are normal), so the hypothesis of microbial $O_2$ consumption may not be tested with the current dataset. We attach expanded figures below (not included in the manuscript).

[Figure]

*Line 561: The comparison of your N₂O measurements with previous measurements can be improved (especially in Fig. A1) as many records exist and the evolution during the Late Holocene is rather flat with only little fine structure. To make this comparison easier for the reader, you might use the Late Holocene Monte Carlo N₂O spline with 2s error envelopes and compare your DF and NEEM records with that spline as in Figure 4 of Fischer et al. 2019, Biogeosciences, doi: 10.5194/bg-16-3997-2019.*

*https://www.ncdc.noaa.gov/paleo-search/study/28890, the direct link to the Late Holocene N₂O spline is: https://www1.ncdc.noaa.gov/pub/data/paleo/icecore/fischer2019/fischer2019spline3k-2o.txt*

We will add the spline curve from Fischer et al. 2019 and also newly published N₂O data (NEEM and Styx Glacier cores) (Ryu et al. 2020, Global Biogeochemical Cycles) to the main figure (error bars are summarized in a box for better visibility of the data points). We keep the original data points in Figure A1 but delete the lines connecting data points for better visibility. We will add a discussion based on the new comparison as follows.

"We also compare our data with the Monte Carlo spline fit through the NGRIP, TALDICE, EDML, and EDC data by the University of Bern (Fischer et al., 2019; see Fig. A1 for individual data points) and high-resolution data from the NEEM and Styx Glacier ice cores by SNU (Ryu et al., 2020). Multi-centennial-scale variations (i.e., relatively low concentrations around 600 C. E. and high concentrations around 1100 C.E.) are commonly seen in all the datasets. However, there appear to be some offsets between the data from NIPR, University of Bern, and SNU. The NEEM

and Styx Glacier data by SNU are systematically lower by ~5 ppb than our data. Because the NEEM core is measured by both laboratories, the offset cannot be explained by the difference in the original N₂O concentration in the ice. We examine here the possibility that our method overestimates the N₂O concentration. Based on the standard-gas transfer tests, we do not apply extraction correction for N₂O concentration. This raises the possibility that, if our tests indeed underestimate the N₂O dissolution, then our ice-core data should become lower than the true values. This scenario leads to an upward correction of our dataset and thus does not explain the offset. Therefore, the causes of the systematic offset between the two datasets may be the differences in standard gas scales and calibration methods employed by the two laboratories. The spline curve by the University of Bern shows depth-dependent offset relative to other datasets. The 2σ error band of the spline curve overlaps well with our data between ~1000 and 1800 C.E., but it is systematically lower than our data and agree with the SNU data for ~0 – 1000 C.E. We measure the ice samples in random order to avoid any apparent trends in the data that might originate in the drifts in the standard gases or instruments (on weekly to monthly timescales)."

[Figure]

*Line 578: to be complete, you might add "... generally measured with mechanic dry extraction techniques (references) and sublimation (Schmitt et al. 2011, doi:10.5194/amt-4-1445-2011).*

We will add the phrase and reference according to your suggestion. The sentence now reads:

"The $CO_2$ concentration is generally measured with dry extraction techniques (mechanical crushing) (e.g., Ahn et al., 2009; Barnola et al., 1987; Monnin et al., 2001; Nakazawa et al., 1993a) and sublimation (Schmitt et al. 2011), ..."

*Line 585 and the following line: It might be better to mention in an introductory sentence that reliable atm. $CO_2$ reconstructions are only possible in Antarctic ice cores because impurity levels in Greenland are too high and lead to excess $CO_2$ (refs). ...so it is no wonder that your NEEM data also shows higher values.*

We will add the following sentence at the end of the paragraph.

"This result is not surprising because it is well known that reliable $CO_2$ reconstructions are only possible from Antarctic ice cores owing to in-situ $CO_2$ production in the Greenland ice sheet with high impurity concentrations (Anklin et al., 1995). Anklin et al. (1995) measured Eurocore (Greenland) late Holocene ice with both dry and wet extraction methods, and found that the wet extraction values were higher (by ~30 to 100 ppm) than the dry extraction values, which in turn are higher than the Antarctic records by up to ~20 ppm. Their results thus suggest that the excess $CO_2$ in our NEEM dataset might be partly produced during the extraction by chemical reactions in the meltwater."

*Line 595: "...outer ice pieces were..."*
We will add "pieces" between " ice" and "were".

*Line 599-601: these two sentences have almost the same meaning, combine.*
We modified the second sentences as:

" In addition, $\delta O_2/N_2$ is more depleted than $\delta Ar/N_2$, consistent with the gas-loss fractionations found in earlier studies, which proposed that the gas-loss fractionation is largely size-dependent (molecular diameter of $O_2$ is smaller than Ar) with weak mass dependency (Bender et al., 1995; Huber et al., 2006; Severinghaus et al., 2009). ".

*Line 601: "...indicating significant size-dependent fractionation in the outer ice". Without further information, the reader has no clue why you immediately state that these observations are due to a size-fractionating process. May be $O_2$ has a higher permeation rate due to solubility in the ice matrix compared to $N_2$. Either provide more background or just write e.g. gas species-dependent gas loss*
We will add some information (see above).

*Line 603: "...but not for N2, in the outer ice." Unclear: because N2 is not lost at all.*
We would like to keep the original sentence because it is possible that some $N_2$ is also lost from ice cores, but our analytical precision is not good enough to detect its isotopic fractionation. To clarify, we change "significant" to "detectable".

*Line 609: please better specify what is plotted in Fig. 14: While it is clear that Fig. 13 shows the differences between*

*samples with 5 or 8 mm removed this is not so clear for Fig. 14.*

We will change the sentence as:

"On the other hand, no significant differences are observed between $\delta O_2/N_2$ values from the inner ice (with different outer removal) if the removal is 8 mm or more (we tested with combinations of 8, 9, 11, and 13 mm) (Fig. 14).".

*Line 612: "...more than 8 mm is sufficient" message not clear: is 8 mm enough?*

8 mm was sufficient in our tests, so we will change the text accordingly. We will also add that we decided to cut 9 mm for the routine ice-core measurements, to have a safety margin.

"From these results, we conclude that the removal of 8 mm is sufficient to obtain the gas composition as originally trapped in the DF1 core. For our routine measurements of the DF core, we decided to cut 9 mm to include extra margin."

*Line 624 : "$\delta O_2/N_2$ as originally trapped...can be reconstructed (delete measured) from the 20-year old samples. Also, the overall match of the corrected old data and the new data is a good sign that the relatively large gas-loss correction was accurately done at that time.*

We will change "measured" to "reconstructed" in the sentence, and add a phrase ", and that the relatively large gas-loss correction by Kawamura et al. (2007) was rather accurate" at the end of the sentence.

*Line 625: regarding the $\delta^{18}O$ values in Fig. 15b it seems that the old values that experienced gas loss (thus mass-dependent fractionation of the $O_2$) are consistently heavier and is thus in line with the idea of the mass-dependent fractionation during the $O_2$ loss from the ice (a scaled correction factor that was used to correct the $O_2/N_2$ data from gas-loss effects might work as well).*

We checked the relationship between $\delta^{18}O$ and storage period or $O_2/N_2$ and found no significant correlation. A reason for the lack of the correlation (if they should in fact) maybe that the magnitude of $\delta^{18}O$ fractionation (expected to be ~0.1 per mil) is one order of magnitude smaller than the atmospheric variations (~1 per mil). Because the old data does not have any duplicated measurements, it is also impossible to examine pair differences, as conducted by Severinghaus et al., (2009). Another possible reason for the bias in the data is the underestimation of gravitational correction by $\delta^{15}N$ (old data is generally lower than the new data). The old data also have much larger measurement errors (0.02 per mil for $\delta^{15}N$ and 0.05 per mil for $\delta^{18}O$ as one sigma) due mainly to a small number of measurement cycles (8 cycles x 1 block). Due to these limitations, it is difficult to discuss the nature of bias in the old $\delta^{18}O$ data, so we would like to keep the original text here.

*Comment regarding the $\delta^{15}N$ values: The new measurements of the clathrate ice look, on average, ca 0.03‰ higher than the old Kawamura values. This systematic offset is not visible for the young ice, thus calibration is not the reason. Yet for gas loss, the sign is in the wrong direction.*

We do not think it is safe to say that the offsets only exist in the clathrate hydrate ice because there are actually not many data points in the Holocene. There are two reasons we can think of the offset between the new and old data, although it is difficult to conclude.

1) The old method did not split the sample for the MS measurement. It first measured the sample gas with GCs, and then the air remained in the sample tube was measured with MS. There is a possibility that the sample gas is fractionated during the GC measurements.

2) The old method did not remove $CO_2$ before the MS measurements, thus correction on $\delta^{15}N$ for the interference of fragmented $CO_2$ (by CO) was applied to $\delta^{15}N$ data. The correction factor was determined only once during the measurement campaign and could have had some bias.

*Line 648 : technique?*

We will correct it.

*comments on the figures and tables:*

*Fig. 1 typo in the scheme: "He ditector", should read detector. Note, the schematic is quite large and in the final version it might get squeezed a bit to fit the page. To allow readability of the labels please increase the font size a bit (in most cases the distances allows that).*

We will increase the font size in the figure and correct "He ditector".

[Figure]

*Fig. 2: you might help the reader find the "small volume (1.4 cm3) between the valves" by adding colors to this section.*

We will add color to this section.

[Figure]

*Fig. 3: the ":" symbol in the legend seems not necessary. electronforming (without n); explain SuS*

We will remove ":", and "SUS" is a word from the Japan Industrial Standard for stainless steel, so we will change it to Stainless Steel (SS).

[Figure]

*Fig. 4: I guess the y-axes are arbitrary units, please add any label to that axes. CH₄ panel, please add labels to the O₂ and CH₄ to make it more intuitive for the reader which peak is what. Since your paper has many figures (15) and tables you might want to save a bit space here and try plotting Fig. 4 more compactly, e.g. plotting all three gases on one x-axis without showing the others (like you did in Fig. 5)*

We will modify the figure according to the suggestions.

[Figure]

*Fig. 5: there seems to be a typo at the x axis labels (16.7.1), also Fig. 7. To better differentiate the symbols (filled vs open markers), you might increase the symbol size.*

We will use a standard format for the x-axes and increase the symbol size.

Fig. 5 STD-A

[Figure]

Fig. 7 Atmosphere

[Figure]

*Panel c. For the O2/N2 ratio is seams that with each new filling of the reference can the trends get smaller, i.e. less O2 is consumed for a certain time interval. Is this an indication that consumption ceases over time as all of the oxygen-reactant is consumed?*

We have four reference cans, so what we see in the graph is the differences of drifts between the different cans. It is possible that the $O_2$ consumption ceases over time (in every can) as you speculate, but we need more time (more rotations of the cans) before we can discuss it.

*Fig. 6: there is a typo in the equation: there is + and - before 0.078802, equation (7) says -0.078802.*
*you may add the equation into the figure rather than in the caption*

We will correct it and add the equation in the figure.

[Figure]

*Fig. 8: it looks like that the new CO₂ measurements are slightly higher between 100 and 150 m and after 1800 m compared to the old ones? perhaps calc. the difference of means for the 100-150 m. Is it due to better extraction efficiency (or lower headspace pressure air reducing CO₂ solution in the meltwater) or longer reaction time for the chemical CO₂ production pathway in the meltwater?*

We think both of your suggestions are possible. The final headspace pressure should be lower in the new method (the reading at P1 is <0.1 Pa compared with <1 Pa in the old method). The transfer time is probably longer in the new method (as we wait for lower pressure). We don't discuss it in this paper because we would need dedicated experiments to draw a definite conclusion.

*Fig. 9: Caption: remove "from other groups". Panel c: you don't need the connecting lines for South Pole and EDML and WDC. just keep it for Law Dome.*

We will delete "from other groups" in the caption. Please see above for the figure.

*Fig. 10: Caption: write "The line connects the blue markers..." since there is only one line, therefore, there is no need to specify this line it as "dotted line (blue)" Actually, I am not sure if you need the blue line at all as the next neighbouring points to the right side are quite a bit older and thus outside of the gas age distribution. For me this situation with the artefact samples is convincing already as the age difference of the plotted data is smaller than the NEEM gas age distribution. Additionally, you might indicate these two artefact depth levels already in Fig. 9 panel a) e.g. with a red circle around the artefact samples that you zoom in in Fig. 10.*

We will remove the blue line in the figure and add markers to indicate two artefact depth in Fig. 9 (please see above ).

[Figure]

*Fig. 11: caption. unclear statement: "For single (non-duplicate) measurements, "a" piece is not cut.". I am not sure if I got it right. If you do a single measurement, piece "b" is measured only, but you need to do the cut between a and b anyway? Further, how did you cut the curvature between the inner and outer pieces?*

We only cut out the "b" piece for the single measurement (cut at (2)), so the "a" and its outer part remain as the main ice body. We usually cut the outer part at red lines by a bandsaw and shave off the rest by a ceramic knife.

[Figure]

*Fig. 12: you might consider producing a single figure (as e.g. nicely done in Fig. 15) out of the four panels because you don't need four-times Depth(m) and the depth scales. The same applies to Figs. 13 and 14. Also, you might consider using a Δ symbol (e.g. Δ δAr/N2 to indicate the "pair differences" to prevent too long y-axis labels every time and write this in the caption (Pair differences (Δ) between the inner and outer ice piece.*

We will show the figures as follows (Figs. 12, 13 and 14).

[Figure]

*Fig. 14: figure: go for the same design: the dashed line at zero is plotted as a thick line in d) while thin for a, b, c. please chose one thickness, e.g. the thin line style. Caption line 955: add pieces: between the two inner ice pieces: and also specify if it is a-b or b-a.*

We will correct the figure and add the phrase as your suggestion.

*Fig. 15: figure top x-axis label: replace Air age with Gas age because you use gas age throughout your manuscript. Legend between panel c) and d) the names for the grey and red crosses are switched, the red symbols refer to the values After gas loss correction?*

The figure was corrected as below.

[Figure]

*caption line 1001: delete dotted lines, because there are none (crosses connected with dotted lines, Kawamura, 2000; Kawamura et al., 2007).*

We will delete it.

*Table A1: typo: TALDICE not TALDAICE; the Eurocore ice core and project is mostly written Eurocore (in some cases EUROCORE, but EURO core would be new)*

We will correct them.

*Fig. A1:*

*panel a) this panel is a bit busy. you might consider plotting more records without the connecting line (EDML, TALDICE Schilt, Siple Dome) typo. EDML (Schilt et all. 2010), et al. And for the legend, TALDICE add a blank between 4.06 and ppb.*

*panel b) this is a busy panel and you might consider plotting only some records (your data and perhaps NEEM from Prokopiou et al) with a connecting line while removing the line for others (Eurocore, GRIP, GISP2, EDC, TALDICE, EDML)*

We will correct your points.

[Figure]

*Finally, a few technical things:*

*Most of your subscripts and superscripts in your manuscript have been moved up or down.*

They were due to unexpected errors of MS Word application when the word file was converted to the PDF file. We will use different conversion software to solve the issue and check all the descriptions.

*Please check but the usual style convention for AMT papers is that the units % and ‰ are written without a blank, e.g. 0.23‰, while all others come with a blank. Also, the unit litre is written L, e.g., 25 mL.*

We will use a capital letter for a liter (L).

We found the statement "Spaces must be included between number and unit (e.g., 1 %, 1 m)." in the Figure content guidelines, so we will not remove a blank before % and ‰.

---

## Author Response (AR1)

**Response to the Referees**

Comments from Referees are written in *italic* and our replies are in blue.

**Anonymous Referee #1**

*Review of "New technique for high-precision, simultaneous measurements of $CH_4$, $N_2O$ and*

*$CO_2$ concentrations, isotopic and elemental ratios of $N_2$, $O_2$ and Ar, and total air content in ice*

*cores by wet extraction" by Oyabu, I. et al. AMTD.*

*General:*

*The manuscript discusses a significantly improved extraction method for sample size without compromising precisions of several important paleo-proxy parameter. The multi-proxy approach is very helpful for improving not only the resolution due to the lower sample amounts necessary, but also regarding the comparison among the different parameters analysed. This again improves the issues with timing, since all the parameters are measured on the same sample, as well as the intercomparison of parameter because only one laboratory and one method is used.*

*The manuscript is very nicely written with detailed information how the method works and how it is used for standard and sample analyses. Furthermore, the authors show tests that only a very limited number of corrections are necessary which is obviously due to the in depth selection and preconditioning of all materials used in the extraction, split and measurement lines. They further state how the values are calibrated.*

*It was easy to read the manuscript and I would like to congratulate the authors. I have only a few rather minor comments and suggestions. I suggest to publish it once these comments have been taken into consideration.*

Thank you very much for your review. Our replies are in blue.

*Minor points:*

*You often used subscripts rather than superscripts in the text. This need to be changed, i.e. $\delta^{15}N$ rather than $\delta_{15}N$, or $cm^3$ rather than $cm_3$. Please check any such issues.*

They were due to unexpected errors of the MS Word application when the word file was converted to the PDF file. We use a different conversion software to solve the issue and checked all the descriptions.

*Line 137: New header (Description of method and manipulations)*

We added a new header there, and we also added two more headers under section 2.1., as the following.

Line 120 (in the revised manuscript with track changes): **2.1.1 Extraction line and its pre-treatment**

"A schematic diagram of our extraction system is…….".

Line141 : **2.1.2 Preparation of apparatus and ice samples**

"For routine air extraction, the sample tubes and extraction line are……….".

Line 170: **2.1.3 Manipulations for air extraction**

"Up to six vessels, thus prepared, are brought to our laboratory room at……..".

*Line 211: Flame Ionization Detector not Frame*

We correct it (line 257).

*Line 293: How are the coefficients d, e, and f calculated, how do they relate to a, b and c?*

For the FIDs, the relationship between peak area and pressure (for the same standard gas) is found to be almost linear over a wide pressure range, but it is slightly nonlinear towards lowest pressures within the range of the ice core measurements. Also, the response of ECD detector is generally nonlinear. We thus interpolate the three calibration points for each of three standard gases by a quadratic fit (the determination of a, b, and c for each gas). The relationship between the concentration and area at any given pressure is also slightly nonlinear, and thus we interpolate the three calibration points on the area-concentration space (from three standard gases, at the pressure of the sample gas) also with a quadratic fit (the determination of d, e, and f). A three-point calibration with quadratic fit is a common practice in precise atmospheric observations. The a, b, and c should be closely related with d, e, and f (after compensating for different units, for the same molar abundance of a gas of interest) because they both represent the same nonlinear responses of the same detectors. However, the coefficients would not necessarily be identical because of uncertainties unrelated to the detector characteristics, such as those from pressure measurement, standard gas scale, and deviation of actual pressure-area relationship from the quadratic function.

We added a figure (Fig. 5) based on the above explanation and some more description (line 332 – 343).

*Eq. 6, lines 380ff: what about the sample loss during the first evacuation after loading the*
*sample? Is this neglectable?*

We indeed take into account the sample loss during the evacuation as follows, although it is small. After all the extractions for a day, the mass of Trap 1 is measured to obtain the total mass of sublimated ice during the first-stage evacuations. The typical amount of ice is 0.5 – 1.5 g for four to six samples (each with ~90-min evacuation). To account for the additional loss during the second evacuation after switching the line to the transfer line (~30 min), the mass of ice in Trap 1 is multiplied by 4/3 (120 min/90 min). The ice loss for each sample is estimated by simply dividing the total ice loss by the number of samples, and it is typically 0.1 – 0.3 g. We added this explanation in lines 208 – 210 and 432 – 437 in the revised manuscript.

*Eq. 8: Why is the normalization made to direct atmospheric air and not to a standard that is*
*well linked to the outside air at a given time.*

The normalization of the sample is indeed made to the reference can, which is determined against the atmosphere from time to time. We revised the text as lines 558 – 575.

*Line 548: ...is impossible to be of atmospheric origin…*

We corrected it (line 619).

*Line 603ff: you might cite Huber et al., EPSL 2006*

We added Huber et al. (2006, EPSL) and Severinghaus et al. (2009, Science), and revised the text following the comment by another reviewer (lines 710 – 713).

**Referee #2 (Jochen Schmitt)**

*General remarks:*

*The paper by Ikumi Oyabu and co-authors presents a new technique to extract and subsequently measure a large suite of key parameters on a single ice core sample. Since so many and diverse parameters are obtained from this new device, this paper is quite extensive. Not only the technique is described, but also new results for several parameters are presented and compared with previous records. Thanks to the fact that the manuscript is both well-structured and clearly written, the reader is not overwhelmed and can find the necessary information without too much effort.*

*The introduction provides the reader with sufficient information of the key applications of the individual parameters that are obtained with the new device. This chapter nicely captures the general advantages of multi-species measurement systems, i.e. to save precious ice and, more importantly, to obtain a large suite of parameters from the same piece of ice. Obviously, this feature is advantageous to improve the palaeo-climatic interpretation (no age difference between parameters), it is also very useful for troubleshooting and scrutinizing the parameters. As the ice core community continuously improves the precision and accuracy of these parameters, more and more previously unknown effects (e.g., production of species in the ice or preferential loss of some gas species from the ice) emerge from the noise level that alter the archived information of the ice core. Multi-species methods that provide a wide array of parameters without compromising the individual precisions are the key techniques to help identify and quantify those effect. In that sense, the new method presented by Oyabu and co-authors is very welcome and sets a new standard for these analyses.*

*Overall, the paper is already in good shape and below I commented only on minor issues to improve the accessibility of the text and figures.*

Thank you very much for your very detailed and thorough review, including a number of specific suggestions on the manuscript. We took them into consideration in revising the manuscript. Below, we reply to your comments in blue.

*Minor comments on the text:*

*Line 77: Beginning of Chapter 2. You immediately start describing your procedure and then you compare/discuss it afterwards with other options to extract air with wet extraction techniques. While OK, you may add a short introductory section to briefly discuss the different wet techniques and point out their advantages and drawbacks. This may help readers not familiar with wet extraction techniques to put your technique into a wider perspective; e.g. ...three different types of wet extraction techniques have been developed that...1) melt-refreeze techniques (you already mention it with their drawbacks) 2) melting under vacuum with the vessel closed, with subsequent helium purging (takes long and consumes lots of helium often used for isotope work on greenhouse gases e.g. Bock et al. 2014, AMT); 3) melting under vacuum with immediate removal of the released air (Kawamura et al., Schmitt et al. 2014, AMT, Bereiter et al. 2018). Method 3) is the preferred way to achieve sufficiently high extraction efficiencies for precise values of soluble trace gases (e.g. $N_2O$, $Xe$) and high-precision ratios of atmospheric main components (e.g. $O_2/N_2$).*

We modified the first paragraph of Chapter 2 in the revised manuscript (lines 83 – 98 in the revised manuscript with track changes).

*Line 83: please clarify: "Also, the pressure over the meltwater is relatively low (~100 – 200 Pa),…" The reported 100 to 200 Pa is not the air pressure over the meltwater (within the vessel) as this value is measured far away downstream at P1, after the water traps, if I got it right. The air pressure after the first water trap is modulated by the individual flow resistances of the tubing etc. between the vessel, the water trap, and the He cryostat. To my knowledge, there is no simple solution to come up with a robust estimate of the air pressure over the meltwater. One option would be to install a pressure sensor close to the vessel and before the first water trap and measure the total pressure ($pH_2O$ at 0°C around 600 Pa plus the air pressure) and subtract the water vapour pressure. Using bubble-free ice as a comparison allows to get an estimate of the actual $pH_2O$ while the ice melts, which might be warmer than 0°C since you melt the ice quite quickly with the hot water bath.*

As you point out, the pressure (~100 – 200 Pa) is the reading of P1 and thus probably much lower than the air pressure over the meltwater in the vessel. We simply removed "(~100 – 200 Pa)" from the text. It would be interesting and informative if we can actually measure the air pressure in the vessel, but we have not done such experiments considering the fact that it would not help to improve the method itself, and that it requires significant cost (a dedicated pressure sensor, some modification to the system, and time and effort to establish the experimental procedures and precision). Together with the prior sentence, we modified the text as the following (lines 108 – 113).

*Line 108: you might add here that this $H_2O$ + $O_2$ pre-treatment is sufficient for the (relatively short) extraction step. However, $O_2/N_2$ ratios are not stable in these sample tubes over a longer time, therefore we use a different set of dip tubes after splitting (GEPW tubes).*

We added a sentence (lines 138 – 140).

.

*Line 111: you might remove this extra info "(i.e., on weekends and holidays)" to help keep the paper short.*

We deleted it (line 144).

*Line 122: remove "highly". unclear: "the surface of ice samples". Actually, the fractionation happens on the exposed outer parts of the stored ice core sections (for me the term "ice sample" refers to the piece of ice that is prepared prior to the measurement in the lab).*

We moved most of the paragraph to the beginning of section 5.2 (discussion of gas-loss fractionation), and modified according to your suggestions (this and the next two suggestions were taken into account) (lines 695 – 701).

*Line 123: "… removed to precisely measure" add: … and accurately the archived composition for these ratios.*

See above.

*Line 127: "… not important for greenhouse gas …large molecular size…". This sentence could be better phrased to prevent misunderstanding. For example, $CO_2$ concentrations could indeed be biased by gas loss if $O_2$ (or if $O_2$, $N_2$, Ar, are lost without relative fractionation) is preferentially lost from the ice, while $CO_2$ is preserved due to its larger diameter. Consequently, $CO_2$ conc. would be biased higher as $CO_2$ conc. is just the ratio of $CO_2$ against total amount of (remaining) air. This process might occur but it is not yet visible due to limitations in the measurement precision. Gas loss biases for $CO_2$ were discussed as a potential option to explain the observed offsets among different ice cores*

*of the same age (e.g. Eggleston et al. 2016 doi:10.1002/2015PA002874).*
See above.

*Line 133: remove the word vessel after Dewar.*
We removed "vessel" (line 167).

*Line 137: "laboratory room", remove the word room after laboratory;*
We removed "room" (line 171).

*Line 144: regarding the impact of the ice + gas loss during sublimation on TAC: can you estimate how much ice is lost due to sublimation (e.g. weighing the sample before and after 90 +30 min of pumping?*
We take into account the sample loss during the evacuation as follows, although it is small. After completing all extractions for a day, the mass of Trap 1 is measured to obtain the total mass of sublimated ice during the first-stage evacuations (all samples combined). The typical amount is 0.5 – 1.5 g for four to six samples (each with ~90-min evacuation). To account for the additional loss during the second evacuation after switching the line to the transfer line (~30 min), the mass of ice in Trap 1 is multiplied by 4/3 (120 min/90 min). Then, the ice loss for each sample is estimated by simply dividing the total ice loss by the number of samples, and it is typically 0.1 – 0.3 g. We added this explanation to the text (lines 208 – 210 and 432 – 437).

*Line 149: "After the evacuation, all vessels, except for one for the first extraction, are…" please rewrite*
We rewrote as line 183.

*Line 156: "The hot water bath is removed after the completion of ice melting." At that point, I wondered how the operator detects when the ice is gone because the vessel where the ice is melted is full metal, thus there is no visual control of that point. I guess it is done via a sudden drop in the pressure reading at P1? Also, all valves of the entire system are manually operated, right?*
We found that the pressure at P1 is not a good guide for knowing the timing of the completion of ice melting, so we rely on the senses of the operators on the noise from the vessel and temperature at the bottom of the vessel. Popping noise can be heard during the ice melting (especially if it is clathrate ice), and it changes to higher frequency noise, which could originate in the boiling of meltwater under vacuum. However, the difference in noise is not always clear, thus we also sense the temperature at the bottom of the vessel by temporarily removing the hot water bath and touch the bottom for a few seconds. The rapidity of temperature decrease sensed by hand can tell if the ice is melted away (if ice still exists, we quickly feel icy coldness). We added a brief description to the manuscript (line 191).
All valves of the extraction and split lines are manually operated.

*Line 159: "The time required for melting the ice sample is <~3 minutes, and the total time for the air transfer takes ~10 min (including the ice melting)." Perhaps it is easier for the reader if you write that the ice melting takes ca. 3 min and the pumping step after melting is completed takes additional ca. 7 min (i.e. the duration of the individual steps).*
We changed the text (line 195).

*Also: convert the mTorr unit into Pa to use only one pressure unit throughout the paper.*
We unified the units (line 194).

*Line 163-165: this sentence could benefit from rephrasing. I am not sure if I got it.*
We changed the text (lines 199 – 202).

*Line 170: "one night" perhaps just write the hours that are physically needed for this equilibration step and not "one night"*
We changed it to "15 – 24 hours" (line 207).

*Line 173: Does that mean you measure two sets of 6 samples each day?*
No, we can operate only one set of 6 samples in a day. We changed the text (line 210).

*Line 182: "...depletion of the $O_2/N_2$ ratio" just to be on the save side, explicitly say that $O_2$ is consumed at the metal surfaces leading to lower $O_2/N_2$ values (to exclude the option that extra $N_2$ causes lower $O_2/N_2$ ratios)*
We added it as line 223.

*Line 184: perhaps add a sentence stating that the stainless steel sample tubes lead to an unacceptable trend in the $O_2/N_2$ ratio as $O_2$ is consumed (perhaps provide numbers if available so the reader gets a handle on the size of this $O_2$ depletion).*
We added a sentence (lines 225 – 228)

*Line 187: delete "(e.g., during nights and weekends)",*
We deleted it (line 231).

*Line 191: replace "(measured at the head of a turbomolecular pump with an ionization gauge) with (P3)*
We corrected it (line 236).

*Line 198: perhaps add the reason for the -196°C cold trap: ..to remove CO2 and N2O...*
We added it (line 242).

*Line 201: is the lowering of the tube by a few cm an essential step or rather precautionary or does it speed up the process?*
This step is a precautionary procedure, at least for our routine analysis with small samples (~1 mL$_{STP}$).

*Line 204: unclear "Sometimes the samples are measured on the following day of the splitting".*
We deleted the sentence and modified the prior sentence as line 249.

*Line 209: "the air left in the original sample tube" can you roughly provide the reader with an estimate of how much the rest is (e.g. fraction of the original amount)? You provide this info (80% vs 20%) only in the conclusion.*

We corrected this part as "the remaining air in the original sample tube (~80 % of the extracted air)..." in line 254.

*Line 211: FID = flame ionization detector*

We corrected it (line 257).

*Line 223-: This GC section is nicely written too, yet due to the inherent complexity of the system it still took me a while to jump from one valve to the next and figure out which way the gas goes. It is hard to come up with a better description but you could help the reader by optimizing details, e.g. you write "we use small sample loops (1/16" o.d., 0.5 cm3, Valco)" here, it would help that you use the same unit as in Fig. 3 (rather than 0.5 cm$^3$ in the text and 500 µL in the figure)...write e.g. we use two 500 µL sample loops (Loop 1 and 2 in Fig. 3 [mentioning the names Loop 1 and 2 helps the reader locate the object quicker] )*

We use the same units (mL) and modified the explanation according to the suggestion (line 268).

*Line 229: 400 – 600 hPa...*

We added it (line 274).

*Line 241: let? "At 1.74 min, V2 is switched to lead $CO_2$ from column 1 to pass through..."*

We corrected it (line 286).

*Line 346 : "... but the first cycle is discarded and only the latter 16 values are used." Delete "and only the latter 16 values are used" as this should be clear already.*

We deleted it (line 394).

*Line 351: "...by simply opening the tube valve". Just curiosity: is the valve opened quickly?*

We open the valve as quickly as possible.

*Line 359: (at ΔP = 0, +100 and -100 ‰). In line 347 you write "... imbalancing the sample pressure by ± 10 %". It is already clear that the ± 10 % refer to the +100 and -100 ‰, but using the same "unit", either % or ‰, could save a few seconds for some readers.*

We use "10%" (line 407).

*Line 437: "The standard gas thus transferred to the sample tubes is (are) measured ..." check is/are.*

We corrected from "are" to "is" (line 492).

*Line 465: as you assume that a constant amount of $O_2$ is consumed independent of the sample size you could use a Keeling plot style fit (i.e. 1/V) and work with a linear system instead of the exp function.*

We tried the Keeling plot and the linear fit, but the goodness of fit turned out to be poor (figure below), and the magnitude of correction seemed to change too rapidly at low volumes. It might imply that $O_2$ consumption may not

strictly be independent of sample size, and we would like to keep the exponential function that gives the better fit.

[Figure]

*Line 501: please rephrase this sentence*
We modified this part as lines 558 – 562.

*Line 511: "…112.88 – 157.81 m (bubbly ice, 2.0 – 0.2 kyr BP)," please arrange order, the same a line below for the NEEM core*
We corrected (lines 581 and 583).

*Line 551: "In any case, we exclude these anomalously high concentrations from the calculation of pooled standard deviation and comparison with other ice core records". Perhaps rephrase: I am not so convinced by your statement that these anomalies should not be compared with other records as CH4 anomalies have been observed for NEEM and GISP2, therefore these anomalies are valuable and important observations. You might also cite and have a look in Mitchel et al. 2013, Supplement figure S1: there are several instances where GISP2 shows elevated CH4 that are not an analytical artefact but features of the ice (see Rhodes et al. papers 2013, 2016).*
We agree with the reviewer and changed this part as lines 624 – 631.

*Just my curiosity: 2 out of three of anomalous samples from 361 m show also elevated $CO_2$ values. Do the corresponding $dO_2/N_2$ values for these samples also deviate from their neighbours (e.g. lighter values due to microbial $O_2$ consumption)?*
We checked $CH_4$, $CO_2$ and $\delta O_2/N_2$ around 361 m and 418 m, where $CH_4$ concentrations are anomalously high. The $\delta O_2/N_2$ are indeed ~3 ‰ lower in the samples with higher $CH_4$. However, $\delta O_2/N_2$ neighboring samples in the vertical direction can vary by a similar amount (for example at ~190 m where $CH_4$ values are normal), so the hypothesis of microbial $O_2$ consumption is not easy to test. We attach expanded figures below (not included in the manuscript).

[Figure]

*Line 561: The comparison of your N₂O measurements with previous measurements can be improved (especially in Fig. A1) as many records exist and the evolution during the Late Holocene is rather flat with only little fine structure. To make this comparison easier for the reader, you might use the Late Holocene Monte Carlo N₂O spline with 2s error envelopes and compare your DF and NEEM records with that spline as in Figure 4 of Fischer et al. 2019, Biogeosciences, doi: 10.5194/bg-16-3997-2019.*

*https://www.ncdc.noaa.gov/paleo-search/study/28890, the direct link to the Late Holocene N₂O spline is: https://www1.ncdc.noaa.gov/pub/data/paleo/icecore/fischer2019/fischer2019spline3k-2o.txt*

We added the spline curve from Fischer et al. 2019 and also newly published N₂O data (NEEM and Styx Glacier cores) (Ryu et al. 2020, Global Biogeochemical Cycles) to the main figure (Fig. 10, error bars are summarized in a box for better visibility of the data points). We keep the original data points in Figure A1 but delete the lines connecting data points for better visibility. We added a discussion based on the new comparison as lines 644 – 660.

*Line 578: to be complete, you might add "… generally measured with mechanic dry extraction techniques (references) and sublimation (Schmitt et al. 2011, doi:10.5194/amt-4-1445-2011).*

We added the phrase and reference according to your suggestion (lines 674 – 675).

*Line 585 and the following line: It might be better to mention in an introductory sentence that reliable atm. CO₂ reconstructions are only possible in Antarctic ice cores because impurity levels in Greenland are too high and lead*

*to excess $CO_2$ (refs). …so it is no wonder that your NEEM data also shows higher values.*

We added a sentence at the end of the paragraph (lines 684 – 689).

*Line 595: "…outer ice pieces were…"*

We will add "pieces" between " ice" and "were" (line 704).

*Line 599-601: these two sentences have almost the same meaning, combine.*

We modified the second sentences as lines 710 – 713.

*Line 601: "…indicating significant size-dependent fractionation in the outer ice". Without further information, the reader has no clue why you immediately state that these observations are due to a size-fractionating process. May be $O_2$ has a higher permeation rate due to solubility in the ice matrix compared to $N_2$. Either provide more background or just write e.g. gas species-dependent gas loss*

We added some information (see above).

*Line 603: "…but not for $N_2$, in the outer ice." Unclear: because $N_2$ is not lost at all.*

We would like to keep the original sentence because it is possible that some $N_2$ is also lost from ice cores, but our analytical precision is not good enough to detect its isotopic fractionation. To clarify, we changed "significant" to "detectable" (line 714).

*Line 609: please better specify what is plotted in Fig. 14: While it is clear that Fig. 13 shows the differences between samples with 5 or 8 mm removed this is not so clear for Fig. 14.*

We changed the sentence as line 720.

*Line 612: "…more than 8 mm is sufficient" message not clear: is 8 mm enough?*

8 mm was sufficient in our tests, so we changed the text as line 724. We also added that we decided to cut 9 mm for the routine ice-core measurements, to have a safety margin (line 725).

*Line 624 : "$\delta O_2/N_2$ as originally trapped…can be reconstructed (delete measured) from the 20-year old samples. Also, the overall match of the corrected old data and the new data is a good sign that the relatively large gas-loss correction was accurately done at that time.*

We changed "measured" to "reconstructed" in the sentence (line 736), and add a phrase ", and that the relatively large gas-loss correction by Kawamura et al. (2007) was rather accurate" at the end of the sentence.

*Line 625: regarding the $\delta^{18}O$ values in Fig. 15b it seems that the old values that experienced gas loss (thus mass-dependent fractionation of the $O_2$) are consistently heavier and is thus in line with the idea of the mass-dependent fractionation during the $O_2$ loss from the ice (a scaled correction factor that was used to correct the $O_2/N_2$ data from gas-loss effects might work as well).*

We checked the relationship between $\delta^{18}O$ and storage period or $O_2/N_2$ and found no significant correlation. A reason for the lack of correlation may be that the magnitude of $\delta^{18}O$ fractionation (expected to be ~0.1 per mil) is one order

of magnitude smaller than the atmospheric variations (~1 per mil). Because the old data does not have any duplicated measurements, it is also impossible to examine pair differences, as conducted by Severinghaus et al., (2009). Another possible reason for the bias in the data is the underestimation of the gravitational correction by $\delta^{15}N$ (old data is generally lower than the new data). The old data also have much larger measurement errors (0.02 per mil for $\delta^{15}N$ and 0.05 per mil for $\delta^{18}O$ as one sigma) due mainly to a small number of measurement cycles (8 cycles x 1 block). Due to these limitations, it is difficult to discuss the nature of bias in the old $\delta^{18}O$ data, so we would like to keep the original text here.

*Comment regarding the $\delta^{15}N$ values: The new measurements of the clathrate ice look, on average, ca 0.03‰ higher than the old Kawamura values. This systematic offset is not visible for the young ice, thus calibration is not the reason. Yet for gas loss, the sign is in the wrong direction.*

We do not think it is safe to say that the offsets only exist in the clathrate hydrate ice because there are actually not many data points in the Holocene. There are two reasons we can think of for the offset between the new and old data, although it is difficult to conclude.

1) The old method did not split the sample for the MS measurement. It first measured the sample gas with GCs, and then the air that remained in the sample tube was measured with MS. There is a possibility that the sample gas was fractionated during the GC measurements.

2) The old method did not remove $CO_2$ before the MS measurements, thus a correction on $\delta^{15}N$ for the interference of fragmented $CO_2$ (by CO) was applied to $\delta^{15}N$ data. The correction factor was determined only once during the measurement campaign and could have had some bias.

*Line 648 : technique?*
We corrected it (line 761).

*comments on the figures and tables:*
*Fig. 1 typo in the scheme: "He ditector", should read detector. Note, the schematic is quite large and in the final version it might get squeezed a bit to fit the page. To allow readability of the labels please increase the font size a bit (in most cases the distances allows that).*
We increased the font size in the figure and corrected "He ditector".

*Fig. 2: you might help the reader find the "small volume (1.4 cm3) between the valves" by adding colors to this section.*
We added color to this section.

*Fig. 3: the ":" symbol in the legend seems not necessary. electronforming (without n); explain SuS*
We removed ":". "SUS" is a word from the Japan Industrial Standard for stainless steel, so we changed it to Stainless Steel (SS).

*Fig. 4: I guess the y-axes are arbitrary units, please add any label to that axes. CH₄ panel, please add labels to the O₂ and CH₄ to make it more intuitive for the reader which peak is what. Since your paper has many figures (15) and*

*tables you might want to save a bit space here and try plotting Fig. 4 more compactly, e.g. plotting all three gases on one x-axis without showing the others (like you did in Fig. 5)*

We modified the figure according to the suggestions.

*Fig. 5: there seems to be a typo at the x axis labels (16.7.1), also Fig. 7. To better differentiate the symbols (filled vs open markers), you might increase the symbol size.*

We used a standard format for the x-axes and increase the symbol size (Fig. 6 and 8).

*Panel c. For the $O_2/N_2$ ratio is seams that with each new filling of the reference can the trends get smaller, i.e. less $O_2$ is consumed for a certain time interval. Is this an indication that consumption ceases over time as all of the oxygen-reactant is consumed?*

We have four reference cans, so what we see in the graph is the differences of drifts between the different cans. It is possible that the $O_2$ consumption ceases over time (in every can) as you speculate, but we need more time (more rotations of the cans) before we can discuss it.

*Fig. 6: there is a typo in the equation: there is + and - before 0.078802, equation (7) says -0.078802.*
*you may add the equation into the figure rather than in the caption*

We corrected it and added the equation in the figure (Fig. 7).

*Fig. 8: it looks like that the new $CO_2$ measurements are slightly higher between 100 and 150 m and after 1800 m compared to the old ones? perhaps calc. the difference of means for the 100-150 m. Is it due to better extraction efficiency (or lower headspace pressure air reducing $CO_2$ solution in the meltwater) or longer reaction time for the chemical $CO_2$ production pathway in the meltwater?*

We think both of your suggestions are possible. The final headspace pressure should be lower in the new method (the reading at P1 is <0.1 Pa compared with <1 Pa in the old method). The transfer time is probably longer in the new method (as we wait for lower pressure). We don't discuss it in this paper because we would need dedicated experiments to draw a definite conclusion.

*Fig. 9: Caption: remove "from other groups". Panel c: you don't need the connecting lines for South Pole and EDML and WDC. just keep it for Law Dome.*

We deleted "from other groups" in the caption (line 1132). Please see above for the figure (Fig. 10).

*Fig. 10: Caption: write "The line connects the blue markers..." since there is only one line, therefore, there is no need to specify this line it as "dotted line (blue)" Actually, I am not sure if you need the blue line at all as the next neighbouring points to the right side are quite a bit older and thus outside of the gas age distribution. For me this situation with the artefact samples is convincing already as the age difference of the plotted data is smaller than the NEEM gas age distribution. Additionally, you might indicate these two artefact depth levels already in Fig. 9 panel a) e.g. with a red circle around the artefact samples that you zoom in in Fig. 10.*

We removed the blue line in the figure and add markers to indicate the two artefact depths in Fig. 10.

*Fig. 11: caption. unclear statement: "For single (non-duplicate) measurements, "a" piece is not cut.". I am not sure if I got it right. If you do a single measurement, piece "b" is measured only, but you need to do the cut between a and b anyway? Further, how did you cut the curvature between the inner and outer pieces?*

We only cut out the "b" piece for the single measurement (cut at (2)), so the "a" and its outer part remain as the main ice body. We usually cut the outer part at red lines by a bandsaw and shave off the rest by a ceramic knife.

[Figure]

*Fig. 12: you might consider producing a single figure (as e.g. nicely done in Fig. 15) out of the four panels because you don't need four-times Depth(m) and the depth scales. The same applies to Figs. 13 and 14. Also, you might consider using a Δ symbol (e.g. Δ δAr/N2 to indicate the "pair differences" to prevent too long y-axis labels every time and write this in the caption (Pair differences (Δ) between the inner and outer ice piece.*

We corrected the figures (Figs. 13, 14 and 15).

*Fig. 14: figure: go for the same design: the dashed line at zero is plotted as a thick line in d) while thin for a, b, c. please chose one thickness, e.g. the thin line style. Caption line 955: add pieces: between the two inner ice pieces: and also specify if it is a-b or b-a.*

We corrected the figure and added the phrase as your suggestion (line 1170).

*Fig. 15: figure top x-axis label: replace Air age with Gas age because you use gas age throughout your manuscript. Legend between panel c) and d) the names for the grey and red crosses are switched, the red symbols refer to the values After gas loss correction?*

The figure was corrected (Fig. 16).

*caption line 1001: delete dotted lines, because there are none (crosses connected with dotted lines, Kawamura, 2000; Kawamura et al., 2007).*

We deleted it (line 1176).

*Table A1: typo: TALDICE not TALDAICE; the Eurocore ice core and project is mostly written Eurocore (in some cases EUROCORE, but EURO core would be new)*

We corrected TALDICE and removed the Eurocore data from the Fig. A1 because the Eurocore data covers only the

last 800 years.

*Fig. A1:*

*panel a) this panel is a bit busy. you might consider plotting more records without the connecting line (EDML, TALDICE Schilt, Siple Dome) typo. EDML (Schilt et all. 2010), et al. And for the legend, TALDICE add a blank between 4.06 and ppb.*

*panel b) this is a busy panel and you might consider plotting only some records (your data and perhaps NEEM from Prokopiou et al) with a connecting line while removing the line for others (Eurocore, GRIP, GISP2, EDC, TALDICE, EDML)*

We corrected your points.

*Finally, a few technical things:*

*Most of your subscripts and superscripts in your manuscript have been moved up or down.*

They were due to unexpected errors of the MS Word application when the word file was converted to the PDF file.

We will use a different conversion software to solve the issue and check all the descriptions.

*Please check but the usual style convention for AMT papers is that the units % and ‰ are written without a blank, e.g. 0.23‰, while all others come with a blank. Also, the unit litre is written L, e.g., 25 mL.*

We use a capital letter for a liter (L).

We found the statement "Spaces must be included between number and unit (e.g., 1 %, 1 m)." in the Figure content guidelines, so we do not remove a blank before % and ‰.

[revised manuscript text omitted]